# On Efficient Online Imitation Learning
# via Classification

**Yichen Li**
University of Arizona
yichenl@arizona.edu

**Chicheng Zhang**
University of Arizona
chichengz@cs.arizona.edu

## Abstract

Imitation learning (IL) is a general learning paradigm for tackling sequential decision-making problems. Interactive imitation learning, where learners can interactively query for expert demonstrations, has been shown to achieve provably superior sample efficiency guarantees compared with its offline counterpart or reinforcement learning. In this work, we study classification-based online imitation learning (abbrev. COIL) and the fundamental feasibility to design oracle-efficient regret-minimization algorithms in this setting, with a focus on the general nonrealizable case. We make the following contributions: (1) we show that in the COIL problem, any proper online learning algorithm cannot guarantee a sublinear regret in general; (2) we propose LOGGER, an improper online learning algorithmic framework, that reduces COIL to *online linear optimization*, by utilizing a new definition of mixed policy class; (3) we design two oracle-efficient algorithms within the LOGGER framework that enjoy different sample and interaction round complexity tradeoffs, and conduct finite-sample analyses to show their improvements over naive behavior cloning; (4) we show that under the standard complexity-theoretic assumptions, efficient dynamic regret minimization is infeasible in the LOGGER framework. Our work puts classification-based online imitation learning, an important IL setup, into a firmer foundation.

## 1 Introduction

Imitation learning (IL), also known as learning from expert demonstrations [47, 44], is a general paradigm for training intelligent behavior for sequential decision making tasks. IL has been successfully deployed in many applications, such as autonomous driving [47, 45], robot arm control [69], game playing [60], and sequence prediction [23, 8]. It is now well-known that with the help of a demonstrating expert, an IL agent can bypass the exploration challenges of reinforcement learning, achieving a much lower sample requirement than reinforcement learning agents [63].

Two major IL paradigms have been studied in the literature: offline and interactive. In offline IL [1, 64, 74, 30], the learner receives a set of expert demonstrations ahead of time; in contrast, in interactive IL [23, 56, 55, 32], the learner has the ability to interactively query the expert for demonstrations on states at its disposal, allowing expert feedback to be provided in a targeted manner. In both settings, the goal of the learner is to output a policy $\hat{\pi}$ that competes with the expert's policy $\pi^E$, by consuming as few resources (e.g. expert annotations) as possible. Between these two, interactive IL is known to be able to achieve superior policy performance than its offline counterpart under certain favorable assumptions on the expert policy and the environment, in that learning agents can use interaction to address the compounding error challenge [54, 49].

Despite the recent progress in the fundamental limits of the interactive imitation learning in the realizable setting [63, 50, 49], the statistical and computational limits of the interactive imitation learning in the general nonrealizable setting remain open. One promising and influential algorithmic framework for studying and analyzing interactive IL in the nonrealizable setting is DAGGER (Data

36th Conference on Neural Information Processing Systems (NeurIPS 2022).

Aggregation) [56], whose key insight is to reduce interactive IL to regret minimization in online learning [59]. Specifically, it constructs a $N$-round online learning game, where at every round $n \in [N]$, the learner outputs some policy $\pi_n$ from a policy class $\mathcal{B}$, and incurs a loss $F_n(\pi)$; the loss is carefully constructed so that the learner's instantaneous loss $F_n(\pi_n)$ characterizes current policy $\pi_n$'s competitiveness compared to the expert $\pi^E$. A representative example of $F_n(\pi)$ is the expected disagreement between $\pi$ and $\pi^E$ on the state occupancy distribution induced by $\pi_n$, used in the original DAGGER paper [56], which can be expressed as the expected zero-one loss of $\pi$ on a distribution of classification examples – we call such setting *Classification-based Online Imitation Learning* (abbrev. COIL). The DAGGER reduction framework has spurred an active line of research on IL [e.g. 55, 63, 15, 18, 17, 38]: it enables conversions from stochastic online optimization algorithms with static or dynamic regret guarantees to IL algorithms with different output policy suboptimality guarantees, allowing the research community to directly translate new results in online learning to those in IL.

Perhaps surprisingly, from a fundamental perspective, rigorous design of efficient regret minimization algorithms for COIL has been largely overlooked by the prior literature. Specifically, many works assume a fixed parameterization of policies in $\mathcal{B}$, and assume that $F_n(\pi)$'s are convex in $\pi$'s underlying parameters to allow for no-regret online convex optimization [e.g. 56, 63, 15]. Although natural, this viewpoint has some issues: (1) in DAGGER's reduction, the learner uses finite-sample approximations of $F_n(\pi)$, which are often discontinuous in $\pi$'s underlying parameters (e.g. given a policy $\pi_\theta(s) = \text{sign}(\langle \theta, s \rangle)$ as a linear classifier, its zero-one loss on a state, $I(\pi_\theta(s) \neq \pi^E(s))$ is discontinuous in $\theta$), making stochastic gradient-based methods inapplicable. Convex surrogate loss functions has been proposed as a popular workaround [56], but it is well-known that in the nonrealizable setting, even for the special case of supervised learning, minimizing convex surrogate losses can result in very different models compared to minimizing the original zero-one classification losses [7]; (2) it makes the usage of policy classes with complex parameterization (e.g. rule-based policies such as decision trees) difficult, as convexity is hard to establish for such classes.

**Overview of our results.** In this paper, we bridge the above-mentioned gaps by studying the fundamental feasibility of designing efficient regret minimization algorithms for COIL, putting the study of statistical and computational limits of interactive imitation learning in the general nonrealizable setting into a firmer foundation. Our first result is that, analogous to Cover's impossibility result in online classification [20], in the COIL setting, any proper online learning algorithm (that outputs a sequence of policies $\{\pi_n\}_{n=1}^N$ from the original class $\mathcal{B}$) cannot guarantee sublinear regret in general (§ 3.1).

The above negative result motivates the design of improper learning algorithms for regret minimization. To this end, we propose to choose policies from a mixed policy class $\Pi_\mathcal{B}$, and provide an algorithmic framework, LOGGER, that reduces COIL to *online linear optimization*. In a nutshell, LOGGER uses a natural parameterization on $\Pi_\mathcal{B}$ that allows to express $F_n(\pi)$ as a linear function of the underyling parameters of $\pi \in \Pi_\mathcal{B}$. We show that any online linear optimization algorithm with (high-probability) regret guarantees can be plugged into LOGGER to obtain an algorithm for COIL with policy suboptimality guarantees (§ 3.2).

Next, enabled by the LOGGER framework, we design computationally efficient algorithms for static regret minimization. Assuming access to an offline cost-sensitive classification (CSC) oracle $\mathcal{O}$, and a set of unlabeled *separator examples* for $\mathcal{B}$ [66, 24], we design LOGGER-M, a sample and computationally efficient algorithm. With $O(1/\epsilon^2)$ interaction rounds and $O(1/\epsilon^2)$ expert annotations, LOGGER-M enjoys a per-round static regret of $\epsilon$ (§ 4.1). Underlying LOGGER-M is a delicate utilization of the connection between Follow-the-Perturbed-Leader and Follow-the-Regularized-Leader, two well-known online learning algorithm families, first observed by [2]. Moreover, by exploiting the predictability of the COIL problem [51, 18, 17], we design an efficient algorithm LOGGER-ME, that enjoys a per-round static regret of $\epsilon$, with $O(1/\epsilon)$ interaction rounds and $O(1/\epsilon^2)$ expert annotations (§ 4.2). Its reduced number of interaction rounds can enable a more practical deployment of IL agents, especially when interactive expert annotations come in batches or with delays.

Finally, we study efficient dynamic regret minimization in the LOGGER framework (§ 5). We show that this is unlikely to be feasible: under a standard complexity-theoretic assumption, no oracle-efficient algorithms can output policies in $\Pi_\mathcal{B}$ with sublinear dynamic regret. Due to space constraints, we discuss key related works throughout the paper, and defer additional related works to Appendix A.

## 2 Preliminaries

**Basic definitions.** Define $[n] := \{1, \ldots, n\}$. Define indicator function $I(\cdot)$ such that $I(E) = 1$ if condition $E$ is true, and $= 0$ otherwise. We use $\Delta(W)$ to denote the set of probability distributions over a finite set $W$, and use $\mathrm{Onehot}(w, W) \in \Delta(W)$ to denote the delta mass on $w \in W$. For a finite $W$, we will oftentimes treat $u \in \mathbb{R}^{|W|}$ (e.g. $u \in \Delta(W)$) as a $|W|$-dimensional vector; for $w \in W$, denote by $u[w]$ the $w$-th coordinate of $u$. We abuse the notation of $\{\cdot\}$ to denote multisets.

**Episodic MDPs.** We study imitation learning in episodic Markov decision processes (MDPs). An episodic MDP $\mathcal{M}$ is a tuple $(\mathcal{S}, \mathcal{A}, H, c, \rho, P)$, where $\mathcal{S}$ is a finite state space (that can be exponentially large), $\mathcal{A}$ is a finite action set, $H \in \mathbb{N}^+$ is the episode length, $c : \mathcal{S} \times \mathcal{A} \to [0, 1]$ is the cost function, and $\rho \in \Delta(\mathcal{S})$ is the initial state distribution. Also, $P = \{P_t\}_{t=1}^{H-1}$ denotes $\mathcal{M}$'s transition dynamics, with $P_t : \mathcal{S} \times \mathcal{A} \to \Delta(\mathcal{S})$ being the transition probability at step $t$. Throughout, we use $S$ and $A$ to denote $|\mathcal{S}|$ and $|\mathcal{A}|$, respectively. Without loss of generality, we assume that $\mathcal{M}$ is layered, where $\mathcal{S}$ can be partitioned into $H$ disjoint sets $\{\mathcal{S}_t\}_{t=1}^H$; the initial distribution $\rho$ is supported on $\mathcal{S}_1$, and transition distribution $P_t(\cdot \mid s, a)$ is supported on $\mathcal{S}_{t+1}$ for all $t, s, a$. For state $s \in \mathcal{S}$, define $\mathrm{Step}(s)$ as the step $t$ such that $s \in \mathcal{S}_t$.

A learning agent interacts with $\mathcal{M}$ for one episode using the following protocol: for every step $t \in [H]$: it observes a state $s_t \in \mathcal{S}_t$, takes an action $a_t \in \mathcal{A}$, incurs cost $c(s_t, a_t)$, and transitions to next state $s_{t+1} \sim P_t(\cdot \mid s_t, a_t)$ except for the last step when it stops. Given a stationary policy $\pi : \mathcal{S} \to \Delta(\mathcal{A})$, we use $\pi(\cdot|s)$ to denote the action distribution of $\pi$ on $s$. Denote by $\mathbb{E}_\pi$ and $\mathbb{P}_\pi$ the expectation and probability over executing (i.e. rolling out) policy $\pi$ in $\mathcal{M}$. Given policy $\pi$, its state occupancy distribution at step $t$ is defined as $d_\pi^t(\cdot) := \mathbb{P}_\pi(s_t = \cdot)$; its average state occupancy distribution is denoted as $d_\pi := \frac{1}{H} \sum_{t=1}^H d_\pi^t$. Let $J(\pi) := \mathbb{E}_\pi \left[ \sum_{t=1}^H c(s_t, a_t) \right] = H \cdot \mathbb{E}_{s \sim d_\pi} \mathbb{E}_{a \sim \pi(\cdot|s)} \left[ c(s, a) \right]$ denote the expected cumulative cost of $\pi$ over an episode. For policy $\pi$, we denote its value function $V_\pi(s) := \mathbb{E} \left[ \sum_{t=\mathrm{Step}(s)}^H c(s_t, a_t) \mid s, \pi \right]$ and action-value function $Q_\pi(s, a) := c(s, a) + \mathbb{E} \left[ \sum_{t=\mathrm{Step}(s)+1}^H c(s_t, a_t) \mid s, a, \pi \right]$; in words, they are the expected costs of rolling out $\pi$ starting from $s$ and $(s, a)$, respectively. For policy $\pi$, define its advantage function as $A_\pi(s, a) := Q_\pi(s, a) - V_\pi(s)$, which measures the expected performance difference by one step deviation of $\pi$ by taking action $a$ at state $s$. Also, we define the recoverability constant as the ability of $\pi$ to recover from deviation when rolled out in $\mathcal{M}$:

**Definition 1** ($\mu$-recoverability)**.** *A (MDP, policy) pair $(\mathcal{M}, \pi)$ is said to be $\mu$-recoverable, if $\forall s \in \mathcal{S}, a \in \mathcal{A}, \left| A_\pi(s, a) \right| \leq \mu$.* [1]

**Interactive IL.** We study interactive imitation learning [23, 54], where the learner has access to a stationary deterministic demonstrating expert $\pi^E$ and would like to learn a policy with low expected cost. Throughout, we assume that $(\mathcal{M}, \pi^E)$ is $\mu$-recoverable, for some $\mu \leq H$ that can possibly be $\ll H$. At each interaction round, the learner interacts with $\mathcal{M}$ for a few episodes, obtaining trajectories of the form $\tau = (s_1, a_1, s_2, a_2, \ldots, s_H, a_H)$ and queries the expert for feedback on some of the states. Specifically, given a state $s$, the feedback given by the expert is of the form $(\zeta_E(s, a))_{a \in \mathcal{A}} \in \mathbb{R}^A$. Two notable examples are: (1) direct expert annotation [56], i.e. given state $s$, expert provides demonstration $\pi^E(s)$, and we use it to construct an $A$-dimensional feedback $(\zeta_E(s, a))_{a \in \mathcal{A}} = (\mu \cdot I(a \neq \pi^E(s)))_{a \in \mathcal{A}}$; (2) estimates of value functions based on experts' rollout [55], i.e. $(\zeta_E(s, a))_{a \in \mathcal{A}} = (A^E(s, a))_{a \in \mathcal{A}} := (A_{\pi^E}(s, a))_{a \in \mathcal{A}}$.[2] Throughout, we assume $\zeta_E(s, a)$ satisfies $\forall s \in \mathcal{S}, a \in \mathcal{A}, A^E(s, a) \leq \zeta_E(s, a) \leq \mu \cdot I(a \neq \pi^E(s))$; this is satisfied by the two examples above. For a stationary policy $\pi$, define its *imitation loss* as: $L(\pi) := \mathbb{E}_{s \sim d_\pi} \mathbb{E}_{a \sim \pi(\cdot|s)} \left[ \zeta_E(s, a) \right]$. By the performance difference lemma [34] (see Lemma 55 in Appendix H), $J(\pi) - J(\pi^E) \leq H \cdot L(\pi)$, implying that if $\pi$ has a small imitation loss, it will have expected cost competitive with $\pi^E$. In light of this connection, in interactive IL, the learner would like to obtain policy $\hat{\pi}$ with low $L(\hat{\pi})$. Subject to this, throughout the paper, we consider optimizing two measures of data efficiency:

---

[1] The $\mu$-recoverability definition here is slightly different from the original ones in [56], in that it also requires that $A_\pi(s, a) \geq -\mu$; we can drop this assumption with a slightly worse sample complexity analysis.

[2] Strictly speaking, in AggreVate and its variants [55, 63], the learner requests expert rollout to obtain *unbiased estimators* of $A^E(s, a)$; our sample complexity analysis can also be adapted to this setting.

**Protocol 1** Classification-based Online Imitation Learning (COIL)

---

**for** $n = 1, \ldots, N$ **do**

    Learner outputs policy $\pi_n$.

    Loss function $F_n(\pi) := \mathbb{E}_{s \sim d_{\pi_n}} \mathbb{E}_{a \sim \pi(\cdot|s)} \left[ \zeta_E(s, a) \right]$.

    Learner draws samples from $\mathcal{D}^E_{\pi_n}$ to obtain information about loss $F_n$, via interacting with $\mathcal{M}$
    and querying the expert for annotation $\zeta_E$.

**end for**

**Goal of learner:** minimize $\sum_{n=1}^{N} F_n(\pi_n) = \sum_{n=1}^{N} L(\pi_n)$.

---

- *Sample complexity*: the total number of expert annotations $\zeta_E$ requested. A smaller sample complexity reduces the total cost of expert annotations (which often takes human effort).
- *Interaction round complexity*: total number of adaptive interaction rounds. A small number of interaction rounds enables more parallelized annotations within an interaction round, and mitigates issues of annotation delay [72, 68].

**The DAGGER reduction framework for interactive IL.** The DAGGER framework reduces minimizing $L(\pi)$ to no-regret online learning [56, 55]. It constructs a $N$-round online learning game, where at every round $n \in [N]$, the learner outputs some policy $\pi_n$, which induces a loss function $F_n(\pi) = \mathbb{E}_{s \sim d_{\pi_n}} \mathbb{E}_{a \sim \pi(\cdot|s)} \left[ \zeta_E(s, a) \right]$. Its key insight is that, by the definition of $\{F_n\}_{n=1}^{N}$, minimizing the online learning cumulative loss $\sum_{n=1}^{N} F_n(\pi_n)$ is equivalent to minimizing the cumulative imitation losses of $\pi_n$'s, i.e. $\sum_{n=1}^{N} L(\pi_n)$. Research efforts in online learning [59, 43] have mainly focused on the design of algorithms that can output $\{\pi_n\}_{n=1}^{N}$ with static regret $\text{SReg}_N(\mathcal{B})$ or dynamic regret $\text{DReg}_N(\mathcal{B})$ against some benchmark policy class $\mathcal{B}$, formally:

$$\text{SReg}_N(\mathcal{B}) := \sum_{n=1}^{N} F_n(\pi_n) - \min_{\pi \in \mathcal{B}} \sum_{n=1}^{N} F_n(\pi), \quad \text{DReg}_N(\mathcal{B}) := \sum_{n=1}^{N} \left( F_n(\pi_n) - \min_{\pi \in \mathcal{B}} F_n(\pi) \right).$$
(1)

Assuming that the learner chooses policies $\{\pi_n\}_{n=1}^{N}$ from another stationary policy class $\mathcal{B}_0$ (which may or may not be $\mathcal{B}$), the following proposition shows that static and dynamic regret guarantees in the induced online learning game can be converted to policy suboptimality guarantees:

**Proposition 2** (e.g. [15]). *For any $N \in \mathbb{N}^+$ and online learner that outputs $\{\pi_n\}_{n=1}^{N} \in \mathcal{B}_0^N$, define* $\text{Bias}(\mathcal{B}, \mathcal{B}_0, N) := \max_{\{v_n\}_{n=1}^{N} \in \mathcal{B}_0^N} \min_{\pi \in \mathcal{B}} \mathbb{E}_{s \sim \bar{d}_N} \mathbb{E}_{a \sim \pi(\cdot|s)} \left[ I(a \neq \pi^E(s)) \right]$, *where $\bar{d}_N := \frac{1}{N} \sum_{n=1}^{N} d_{v_n}$.*
*Then, choosing $\hat{\pi}$ uniformly at random from $\{\pi_n\}_{n=1}^{N}$ has guarantee:*

$$\mathbb{E}\left[ J(\hat{\pi}) - J(\pi^E) \right] \leq H \cdot \min \left\{ \mu \cdot \text{Bias}(\mathcal{B}, \mathcal{B}_0, N) + \frac{\mathbb{E}[\text{SReg}_N(\mathcal{B})]}{N}, \ \mu \cdot \text{Bias}(\mathcal{B}, \mathcal{B}_0, 1) + \frac{\mathbb{E}[\text{DReg}_N(\mathcal{B})]}{N} \right\}.$$

In the above proposition, $\text{Bias}(\mathcal{B}, \mathcal{B}_0, N)$ takes the worst-case mixture of state occupancy distributions $\{d_{v_n}\}_{n=1}^{N}$ induced by $N$ policies from $\mathcal{B}_0$, and measures the expected disagreement between $\pi^E$ and its best approximating policy in $\mathcal{B}$. Informally, it measures the "approximation error" of benchmark class $\mathcal{B}$: it is always nonnegative, and in the special case of $\pi^E \in \mathcal{B}$ (which we call the *realizable case*), $\text{Bias}(\mathcal{B}, \mathcal{B}_0, N) = 0$. Proposition 2 gives two ways to obtain a competitive imitation policy: (1) choose $(\mathcal{B}, \mathcal{B}_0)$ with a small $\text{Bias}(\mathcal{B}, \mathcal{B}_0, N)$ and achieve a low static regret; (2) choose $(\mathcal{B}, \mathcal{B}_0)$ with a small $\text{Bias}(\mathcal{B}, \mathcal{B}_0, 1)$ and achieve a low dynamic regret. Although achieving low dynamic regret can be significantly more challenging than achieving low static regret, minimizing dynamic regret has the advantage that its approximation error term $\text{Bias}(\mathcal{B}, \mathcal{B}_0, 1)$ is smaller than the static regret formulation's counterpart $\text{Bias}(\mathcal{B}, \mathcal{B}_0, N)$.

**Classification-based Online Imitation Learning (COIL).** As we consider a finite action space $\mathcal{A}$, a policy $\pi$ can be equivalently viewed as a (possibly randomized) multiclass classifier. A cost-sensitive classification (CSC) example is defined to be a pair $(x, \vec{c})$, where $x \in \mathcal{S}$ is its feature part, and $\vec{c} \in \mathbb{R}^A$ is its cost part. Under the DAGGER reduction framework, the loss at iteration $n$, $F_n(\pi) = \mathbb{E}_{s \sim d_{\pi_n}} \mathbb{E}_{a \sim \pi(\cdot|s)} \left[ \zeta_E(s, a) \right]$, can be viewed as the expected cost of policy $\pi$ on a distribution of cost-sensitive examples $\mathcal{D}^E_{\pi_n}$ (formally, $\mathbb{E}_{(s, \vec{c}) \sim \mathcal{D}^E_{\pi_n}} \left[ \vec{c}(h(s)) \right]$), where a sample $(s, \vec{c})$ is drawn from $\mathcal{D}^E_\pi$ by first rolling out $\pi$ and drawing $s \sim d_\pi$, and query the expert on $s$ to obtain

$(\zeta_E(s, a))_{a \in \mathcal{A}}$ as its associated $\vec{c}$. The learner can obtain a finite-sample approximation to $F_n(\pi)$ by interacting with $\mathcal{M}$ and the expert to draw samples from $\mathcal{D}_{\pi_n}^E$. We will focus on designing efficient regret-minimizing algorithms in the COIL setting; see Protocol 1 for a summary.

In addition to data efficiency, we also consider the design of imitation learning algorithms with computational efficiency guarantees. To this end, following a sequence of empirically and theoretically successful works on oracle-efficient learning [66, 25, 5, 21], we assume access to the benchmark policy class $\mathcal{B}$, a collection of $B$ stationary and deterministic policies $h : \mathcal{S} \to \mathcal{A}$, via the following computational oracle:

**Definition 3** (CSC oracle). *A CSC oracle $\mathcal{O}$ for policy class $\mathcal{B}$ is such that: given any input multiset of cost-sensitive examples $D = \{(x_1, \vec{c}_1), \ldots, (x_K, \vec{c}_K)\} \in (\mathcal{S} \times \mathbb{R}^A)^K$, it outputs the policy in $\mathcal{B}$ that has the smallest empirical cost, formally,*

$$\mathcal{O}(D) := \operatorname*{argmin}_{h \in \mathcal{B}} \mathbb{E}_{(x, \vec{c}) \sim D} \left[ \vec{c}(h(x)) \right],$$

*where we slightly abuse the notation and and use $D$ to also denote the uniform distribution over it.*

We measure an algorithm's computational efficiency by its number of calls to oracle $\mathcal{O}$.

## 3  LOGGER: reducing COIL to online linear optimization

In this section, we introduce our main algorithmic framework, LOGGER (an abbreviation for Linear lOss aGGrEgation) for designing regret-minimizing algorithms in the COIL setting. Section 3.1 motivates our approach by showing that natural proper learning-based approaches fail to achieve sublinear regret in general; Section 3.2 introduces our approach of performing improper learning using a carefully-defined mixture policy class, via a reduction to online linear optimization.

### 3.1  Can we achieve sublinear regret using proper learning?

A natural idea for minimizing regret is *proper learning*: at round $n$, the learner chooses some policy $\pi_n$ (possibly at random) from $\mathcal{B}_0 = \mathcal{B}$, our benchmark policy class, using some online learning algorithm, based on the information collected in the first $n - 1$ rounds; the learner then collects information on $F_n$ via rollouts of $\pi_n$ and expert annotations, and continue to the next iteration.

While this approach has demonstrated sharp online regret guarantees in classical online cost-sensitive classification settings [40, 27], perhaps subtly, we show in the following theorem that, this approach is insufficient to guarantee sublinear regret in the COIL setting.

**Theorem 4.** *Suppose the expert's feedback $\zeta_E(s, a)$ is of the form $\mu \cdot I(a \neq \pi^E(s))$ or $A^E(s, a)$. Then, for any $H \geq 3$, there exists an MDP $\mathcal{M}$ of episode length $H$, a deterministic expert policy $\pi^E$, a benchmark policy class $\mathcal{B}$, such that for any learner that sequentially (and possibly at random) generates a sequence of policies $\{\pi_n\}_{n=1}^N \in \mathcal{B}^N$, its static regret satisfies $\mathrm{SReg}_N(\mathcal{B}) = \Omega(N)$.*

The proof of the theorem can be found at Appendix D.1. Its key insight is that, distinct from the classical online CSC setting, in COIL, the loss at round $n$, $F_n$, *depends on the policy chosen at that round* $\pi_n$, making standard regret minimization results in online classification [40, 27] inapplicable. In more detail, we construct a *fixed* MDP that "act adversarially" to policies in $\mathcal{B}$, such that any $\pi_n \in \mathcal{B}$ has $F_n(\pi_n) \geq \frac{H-1}{H}$, whereas $\min_{\pi \in \mathcal{B}} \sum_{n=1}^N F_n(\pi) \leq \frac{N}{2}$. Our theorem is similar in spirit to Cover's impossibility result in online classification [20], which shows that an adversary that adapts to the randomness of the learner at each round can force the learner to suffer linear regret.

### 3.2  A new hypothesis class and the LOGGER algorithmic framework

To sidestep the impossibility result in Theorem 4, we apply the "convexification by randomization" technique in online convex optimization [59] by improper learning on a mixed policy class, defined below.

**Definition 5** (Mixed policy class). *Given policy class $\mathcal{B}$, define its induced mixed policy class*

$$\Pi_{\mathcal{B}} := \left\{ \pi_u(\cdot|s) := \sum_{h \in \mathcal{B}} u[h] \cdot h(\cdot|s) : u \in \Delta(\mathcal{B}) \right\}.$$

---

**Algorithm 2** LOGGER: reducing COIL to online linear optimization

---

1: **Input:** MDP $\mathcal{M}$, Expert feedback $\zeta_E$, sample size per iteration $K$, Online linear optimization algorithm OLOA with decision set $\Delta(\mathcal{B})$.
2: **for** $n = 1, 2, \ldots, N$ **do**
3:    Choose $u_n \leftarrow \text{OLOA}(\{g_i\}_{i=1}^{n-1})$, which induces policy $\pi_n := \pi_{u_n}$.
4:    Draw $K$ examples $D_n = \{(s, \vec{c})\}$ iid from $\mathcal{D}_{\pi_n}^E$, via interaction with $\mathcal{M}$ and expert $\zeta_E$.
5:    $D_n$ induces $g_n = \left( \mathbb{E}_{(s,\vec{c}) \sim D_n} \mathbb{E}_{a \sim h(\cdot|s)} \left[ \vec{c}(a) \right] \right)_{h \in \mathcal{B}}$, an unbiased estimator of $\theta(u_n)$.
6: **end for**

---

*We slightly abuse notation and use $h(\cdot \mid s) \in \Delta(\mathcal{A})$ to denote the delta mass on $h(s) \in \mathcal{A}$.*[3]

At a cursory glance, choosing a policy from $\Pi_\mathcal{B}$ seems equivalent to choosing some policy at random from $\mathcal{B}$, which also falls into the failure mode of proper learning (Theorem 4). We remark that this is not true: rolling out a policy from $\pi_u \in \Pi_\mathcal{B}$ is equivalent to drawing new policies in an i.i.d. fashion from $\mathcal{B}$ *at every step of the episode* instead. As we will see next, the usage of $\Pi_\mathcal{B}$ enables the design of COIL algorithms with sublinear regret.

Our key observation is that with the learner outputting policies from the mixed policy class $\Pi_\mathcal{B}$, online regret minimization in IL becomes an *online linear optimization* problem. Recall that in online IL, the loss at round $n$ is $F_n(\pi) = \mathbb{E}_{s \sim d_{\pi_n}} \mathbb{E}_{a \sim \pi(\cdot|s)} \left[ \zeta_E(s, a) \right]$. By choosing $\pi_n = \pi_{u_n} \in \Pi_\mathcal{B}$ and $\pi = \pi_u \in \Pi_\mathcal{B}$ for $u_n, u \in \Delta(\mathcal{B})$, $F_n(\pi_u)$ can be viewed as a linear function of $u$:

$$F_n(\pi_u) = \sum_{h \in \mathcal{B}} u[h] \cdot \mathbb{E}_{s \sim d_{\pi_n}} \mathbb{E}_{a \sim h(\cdot|s)} \left[ \zeta_E(s, a) \right] = \langle \theta(u_n), u \rangle,$$

where $\theta(v) := \left( \mathbb{E}_{s \sim d_{\pi_v}} \mathbb{E}_{a \sim h(\cdot|s)} \left[ \zeta_E(s, a) \right] \right)_{h \in \mathcal{B}}$. We have

$$\sum_{n=1}^{N} \langle \theta(u_n), u_n \rangle = \sum_{n=1}^{N} F_n(\pi_n), \quad \min_{u \in \Delta(\mathcal{B})} \sum_{n=1}^{N} \langle \theta(u_n), u \rangle = \min_{u \in \Delta(\mathcal{B})} \sum_{n=1}^{N} F_n(\pi_u) = \min_{\pi \in \mathcal{B}} \sum_{n=1}^{N} F_n(\pi),$$

and therefore, minimizing the static regret $\text{SReg}_N(\mathcal{B})$ is equivalent to minimizing the static regret in the online linear optimization problem with losses $\left\{ u \mapsto \langle \theta(u_n), u \rangle \right\}_{n=1}^{N}$.

This motivates LOGGER (Algorithm 2), our main algorithmic framework. Given input an online linear optimization algorithm OLOA and sample size $K$, LOGGER outputs policy sequence $\{\pi_n\}_{n=1}^{N}$. Specifically, at round $n$, LOGGER calls OLOA to perform online linear optimization with respect to linear losses $\{u \mapsto \langle g_i, u \rangle\}_{i=1}^{n-1}$ and obtains $u_n \in \Delta(\mathcal{B})$, which corresponds to a policy $\pi_n \in \Pi_\mathcal{B}$ (line 3); here for every $i$, $g_i$ is an unbiased estimator of $\theta(u_i)$. It then rolls out $\pi_n$ in $\mathcal{M}$ for $K$ times to obtain $K$ samples iid from $d_{\pi_n}$, queries the expert on each sample $s$ to obtain $(\zeta_E(s, a))_{a \in \mathcal{A}}$ as its associated $\vec{c}$, and constructs dataset $D_n = \{(s, \vec{c})\}$ (line 4). Finally LOGGER computes the empirical loss of policies on $D_n$, i.e. $g_n = \left( \mathbb{E}_{(s,\vec{c}) \sim D_n} \mathbb{E}_{a \sim h(\cdot|s)} \left[ \vec{c}(a) \right] \right)_{h \in \mathcal{B}}$ (line 5).

**Comparison to prior works.** [16] considers a general online convex optimization formulation for online IL, dubbed "continuous online learning"; our loss function $\langle \theta(u), \cdot \rangle$ can be viewed as an instantiation of the loss function $f_u(\cdot)$ therein. However, their regret minimization results assume that $f_u(\cdot)$ is strongly convex, which do not cover our COIL setting where $f_u(\cdot)$ is linear.

Define $\text{LReg}_N := \sum_{n=1}^{N} \langle g_n, u_n \rangle - \min_{u \in \Delta(\mathcal{B})} \sum_{n=1}^{N} \langle g_n, u \rangle$ as OLOA's static regret with respect to $\{\langle g_n, \cdot \rangle\}_{n=1}^{N}$. We have the following proposition that links $\text{LReg}_N$ to $\text{SReg}_N(\mathcal{B})$, the static regret of $\{\pi_n\}_{n=1}^{N}$ in the online IL problem.

**Proposition 6.** *For any $\delta \in (0, 1]$, if LOGGER uses some OLOA that outputs $\{u_n\}_{n=1}^{N} \subset \Delta(\mathcal{B})^N$ such that with probability at least $1 - \delta/3$, $\text{LReg}_N \leq \text{Reg}(N)$. Then, with probability at least $1 - \delta$, its output policies $\{\pi_n\}_{n=1}^{N}$ satisfy $\text{SReg}_N(\mathcal{B}) \leq \text{Reg}(N) + O\left( \mu \sqrt{\frac{N \ln(B/\delta)}{K}} \right)$.*

---

[3][63, Theorem 5.3] proposes to perform no-regret learning using another definition of nonstationary mixed policy class. We identify a technical issue with this approach, and defer a detailed discussion to Appendix D.2.

Proposition 6 shows that Algorithm 2 is a regret-preserving reduction from online IL to online linear optimization over $\Delta(\mathcal{B})$, a $B$-dimensional probability simplex. The latter is well-known as the "prediction with expert advice" problem [27] (abbrev. expert problem), where algorithms with different guarantees abound, such as Follow the Regularized Leader (FTRL), Hedge [27] and its adaptive and optimistic variants [e.g. 61, Section 1], many of which have optimal worst-case regret bounds $\mathrm{Reg}(N) = O\left(\sqrt{N \ln(B)}\right)$. Instantiating Algorithm 2 with OLOA set as these algorithms, we obtain a family of online IL algorithms with expected regret of order $O(\sqrt{N})$.

Although satisfying from a statistical efficiency perspective, such online IL algorithms suffers from computational inefficiency: they require explicit calculation of $g_n$ and maintenance of $u_n$, which are $B$-dimensional (dense) vectors whose entries need to be updated separately. For instance, when Hedge is chosen as OLOA, $u_n[h] \propto \exp(-\eta \sum_{i=1}^{n-1} g_n[h]))$ for all $h \in \mathcal{B}$, which naively requires $O(B)$ time per round to maintain. To address this computational efficiency issue, in the next section, we exploit the cost-sensitive classification nature of the COIL problem to design sublinear-regret algorithms that use implicit representations of $g_n$'s, i.e. $D_n$'s, that enjoy oracle-efficiency guarantees.

# 4 Efficient algorithms with static regret guarantees

Using the LOGGER framework, in this section, we propose two oracle-efficient COIL algorithms that have sublinear static regret guarantees against policy class $\mathcal{B}$, in Subsections 4.1 and 4.2 respectively.

## 4.1 LOGGER-M: an efficient algorithm with $O(\sqrt{N})$ static regret

The LOGGER reduction framework calls for a computationally and statistically efficient OLOA, which, if devised, yields an computationally and statistically efficient online imitation learner. However, when viewed as a general adversarial online learning problem, computational hardness results [29] suggest that, even with access to classification oracle $\mathcal{O}$, a prohibitive $\Omega(\sqrt{B})$ time complexity is necessary for sublinear regret. Therefore, in subsequent sections, we adopt an assumption on $\mathcal{B}$ in [66], which, to the best of our knowledge, is the state-of-the-art weakest assumption that allows the design of oracle-efficient online CSC algorithms in the adversarial setting:

**Assumption 1** (Small separator set). *There exists a set $\mathcal{X} \subset \mathcal{S}$ (called the separator set) such that, for every pair of distinct policies $h, h' \in \mathcal{B}$, $\exists x \in \mathcal{X}$, such that $h(x) \neq h'(x)$. Denote by $X := |\mathcal{X}|$.*

**Technical challenges.** Even under the small separator set assumption, the design of low-regret oracle-efficient algorithms for imitation learning still remains nontrivial. A naive application of existing oracle-efficient online CSC algorithms, such as Contextual Follow the Perturbed Leader (CFTPL) [66, 24], still falls into the failure mode of proper learning (Theorem 4 in Section 3.1), where an $\Omega(N)$ regret lower bound is unavoidable in the worst case. This is in sharp contrast to the classical online CSC setting, where CFTPL enjoys a $O(\sqrt{N})$ regret [66, 24]. To recap, at round $n$, CFTPL first constructs a random set of "hallucinated" cost-sensitive examples $Z$ based on the separator set $\mathcal{X}$; it subsequently calls the CSC oracle $\mathcal{O}$ on the union of $Z$ and the accumulated dataset $\cup_{i=1}^{n-1} D_i$ to obtain policy $\pi_n \in \mathcal{B}$. CFTPL achieves computational efficiency by operating on $D_n$'s, an implicit representation of $g_n$'s, the linear losses of the underlying OLO problem.

**Our approach.** The above difficulty motivates the need of a new algorithmic approach for efficient classification-based online IL. In view of Section 3.2's observation that FTRL approaches enjoy a sublinear regret, we ask the question: is it possible to perform FTRL in an oracle-efficient manner? A positive answer will simultaneously address the computational and statistical challenges of COIL.

We answer this question in the affirmative, by utilizing a connection between FTRL and FTPL first observed in [2]: an in-expectation version of FTPL can be viewed as an FTRL algorithm. Using this observation, we design Algorithm 3, namely Mixed CFTPL (abbrev. MFTPL), which mimics FTRL by approximating the in-expectation version of CFTPL in an oracle-efficient manner. Similar to CFTPL, MFTPL keeps $g_n$ implicitly in $D_n$, and calls the CSC oracle. Different from CFTPL, MFTPL runs the oracle-call step in CFTPL for $T$ times and outputs the uniform mixture of the $T$ policies. We refer to $T$ as Algorithm 3's *sparsification parameter*, due to the algorithm's resemblance to Maurey's sparsification [46].

---

**Algorithm 3** MFTPL: an oracle-efficient approximation of FTRL

---

1: **Input:** Linear losses $\{g_i\}_{i=1}^{n-1}$ represented by datasets $\{D_i\}_{i=1}^{n-1}$ each of size $K$ (s.t. $g_i[h] = \mathbb{E}_{(s,\vec{c})\sim D_i}\left[\vec{c}(h(s))\right]$ for all $h \in \mathcal{B}$) , separator set $\mathcal{X}$, learning rate $\eta$, sparsification parameter $T$.
2: **for** $j = 1, 2, \ldots, T$ **do**
3:    Draw $(\ell_{x,j}(a))_{a\in\mathcal{A}} \sim \mathcal{N}(0, I_A)$ iid for each $x \in \mathcal{X}$.
4:    Define $Z_j = \left\{(x, \frac{K}{\eta}\ell_{x,j}) : x \in \mathcal{X}\right\}$.
5:    Compute $u_{n,j} \leftarrow \text{Onehot}(\mathcal{O}((\cup_{i=1}^{n-1}D_i) \cup Z_j), \mathcal{B})$.
6: **end for**
7: **return** $u_n \leftarrow \frac{1}{T}\sum_{j=1}^{T} u_{n,j}$.

---

Specifically, at each iteration $j \in [T]$, MFTPL first draws $(\ell_{x,j}(a))_{a\in\mathcal{A}} \sim \mathcal{N}(0, I_A)$ iid for each $x$ in the separator set $\mathcal{X}$, where $I_A$ is the identity matrix of dimension $A$ (line 3). It then constructs a perturbation set of cost-sensitive examples $Z_j = \left\{(x, \frac{K}{\eta}\ell_{x,j}) : x \in \mathcal{X}\right\}$ that contains each $x$ within the separator set and $\frac{K}{\eta}\ell_{x,j}$ as associated $\vec{c}$, where $K$ accounts for the adjustment on dataset size and $\eta$ accounts for FTRL's learning rate (line 4). It then calls the oracle $\mathcal{O}$ with datasets $\cup_{i=1}^{n-1}D_i$ accumulated so far, together with perturbation examples $Z_j$ to obtain an empirical cost minimizer $h \in \mathcal{B}$. This $h$ is represented by a one-hot vector $u_{n,j} \in \Delta(\mathcal{B})$ that has weight 1 on the $h$-th coordinate and 0 elsewhere (line 5). Finally, after $T$ iterations, MFTPL returns the mean value $u_n = \frac{1}{T}\sum_{j=1}^{T} u_{n,j}$ (line 7). MFTPL guarantees that:

**Lemma 7.** *There exists some strongly convex function $R : \Delta(\mathcal{B}) \rightarrow \mathbb{R}$, such that the following holds. Suppose* MFTPL *receives datasets $\{D_i\}_{i=1}^{n-1}$, separator set $\mathcal{X}$, learning rate $\eta$, sparsification parameter $T$. Then, $\forall \delta \in (0,1]$, with probability at least $1 - \delta$,* MFTPL *makes $T$ calls to the cost-sensitive oracle $\mathcal{O}$, and outputs $u_n \in \Delta(\mathcal{B})$ such that*

$$\forall s \in \mathcal{S}, \|\pi_{u_n}(\cdot|s) - \pi_{u_n^*}(\cdot|s)\|_1 \leq \sqrt{\frac{2A\left(\ln(S) + \ln(\frac{2}{\delta})\right)}{T}},$$

*with $u_n^* := \text{argmin}_{u\in\Delta(\mathcal{B})}\left(\langle \eta\sum_{i=1}^{n-1} g_i, u\rangle + R(u)\right)$.*

Therefore, by setting $T = \Omega(A\ln(S))$, the policy $\pi_n$ induced by the MFTPL's output $u_n$ closely mimics $\pi_{u_n^*}$, a policy induced by the FTRL output $u_n^*$. The mild $\ln(\cdot)$ dependence on $S$ makes the lemma useful in large-state-space settings. Specifically, a naive attempt to show Lemma 7 is to establish the $\|\cdot\|_1$ closeness of $u_n$ and $u_n^*$, given that $T \cdot u_n \sim \text{Multinomial}(T, u_n^*)$. This unavoidably carries an impractical concentration factor of $O(\sqrt{B/T})$, as the bound requires $T$ to be $\Omega(B)$ to be non-vacuous. We circumvent this challenge by directly showing the closeness of the action distributions $\pi_{u_n}$ and $\pi_{u_n^*}$ for all states. We defer the full version of the lemma, including an explicit form of $R$, to Appendix E.1.

**Lemma 8.** *For any $\delta \in (0,1]$,* MFTPL, *if called for $N$ rounds, with input learning rate $\eta = \frac{1}{\mu\sqrt{NA}}\left(\frac{\ln(B)}{X}\right)^{\frac{1}{4}}$ and sparsification parameter $T = \frac{N\ln(2NS/\delta)}{\sqrt{X^3\ln(B)}}$, outputs a sequence $\{u_n\}_{n=1}^{N}$, such that with probability at least $1 - \delta$:*

$$\text{LReg}_N \leq O\left(\mu\sqrt{NA}\left(X^3\ln(B)\right)^{\frac{1}{4}}\right).$$

Composing LOGGER with MFTPL, we obtain an efficient online IL algorithm, LOGGER-M. Its regret guarantees immediately follow from combining Lemma 8 with Proposition 6:

**Theorem 9.** *For any $\delta \in (0,1]$,* LOGGER-M, *with $K = 1$ and* MFTPL *setting its parameters as in Lemma 8, is such that: (1) with probability at least $1 - \delta$, its output $\{\pi_n\}_{n=1}^{N}$ satisfies: $\text{SReg}_N(\mathcal{B}) \leq O\left(\mu\sqrt{NA\ln(1/\delta)}(X^3\ln(B))^{\frac{1}{4}}\right)$; (2) it queries $N$ annotations from the expert; (3) it calls the CSC oracle $\mathcal{O}$ for $\frac{N^2\ln(6NS/\delta)}{\sqrt{X^3\ln(B)}}$ times.*

## 4.2 LOGGER-ME: an efficient algorithm with $O(1)$ static regret

Although LOGGER-M is oracle-efficient, it is unclear whether its $O(\sqrt{N})$ regret guarantee is optimal. As a lower regret can translate to lower sample and interaction round complexity guarantees, it is desirable to design algorithms with regret as low as possible.

A key observation from prior works [e.g. 15, 16, 38] is that, online IL is a *predictable* online learning problem [19, 51], and is thus not completely adversarial. This opens up possibilities to bypass the $O(\sqrt{N})$ worst-case regret barrier. Specifically, in the LOGGER framework, the coefficient of the linear loss at round $n$, $\theta(u_n)$, depends *continuously* on $u_n$; more concretely, we can show:

**Lemma 10.** *For $u, v \in \Delta(\mathcal{B})$, $\|\theta(u) - \theta(v)\|_\infty \leq \mu H \cdot \max_{s \in \mathcal{S}} \|\pi_u(\cdot|s) - \pi_v(\cdot|s)\|_1 \leq \mu H \|u - v\|_1$.*

This property and its variants, termed *distributional continuity*, has been utilized in many online IL algorithms to achieve sharper regret guarantees. These works additionally exploit the strong convexity on the loss functions $F_n(\pi)$ [15, 16, 38], or use some external predictive model that can predict $\nabla F_n(\pi)$ well [18, 17]. Unfortunately, in our COIL setting, neither is the loss function $F_n(\pi_u) = \langle \theta(u_n), u \rangle$ strongly convex in the policy parameter $u$, nor do we have access to an external predictive model, rendering these approaches inapplicable.

We get around these challenges and design an oracle-efficient algorithm, namely MFTPL-EG (where EG stands for extra-gradient), with $O(1)$ regret for the online linear optimization problem, which, when composed with the LOGGER framework, yields the LOGGER-ME algorithm with $O(1)$ regret in the COIL setting. MFTPL-EG is largely inspired by the predictor-corrector framework for policy optimization [17] and extragradient methods in smooth optimization [42, 33]; its details can be found in Appendix E.2. Its key insight is that, although we do not have a predictive model for $\theta(u_n)$, we can use an extra round of interaction with $\mathcal{M}$ and expert annotations to obtain a good estimate of it. Based on this, we derive an online linear optimization regret guarantee of MFTPL-EG, deferred to Appendix E.2. This immediately implies the following guarantee of LOGGER-ME:

**Theorem 11.** *For any $\delta \in (0, 1]$, LOGGER-ME, with $K$ and MFTPL-EG's parameters set appropriately, is such that: (1) with probability at least $1 - \delta$, its output $\{\pi_n\}_{n=1}^N$ satisfies:* $\mathrm{SReg}_N(\mathcal{B}) \leq O(\mu H A \sqrt{X^3 \ln(B)})$; *(2) it queries* $O\left(\frac{N^2 \ln(NB/\delta)}{H^2 A \sqrt{X^3 \ln(B)}}\right)$ *annotations from the expert; (3) it calls the CSC oracle $\mathcal{O}$ for* $O\left(\frac{N^3 \ln(NS/\delta)}{\mu H A X^3 \ln(B)}\right)$ *times.*

**Discussion and comparison.** We now compare the guarantees of LOGGER-M, LOGGER-ME, and the baseline of behavior cloning, where the learner simply draws iid examples from $\mathcal{D}_{\pi_E}^E$ and perform empirical risk minimization over $\mathcal{B}$ to learn a policy $\hat{\pi}$. All algorithms' output policy suboptimality guarantees have the following decomposition:

$$\mathbb{E}\left[J(\hat{\pi}) - J(\pi^E)\right] \leq \mathrm{ApproxErr} + \mathrm{EstimErr}, \tag{2}$$

where $\mathrm{ApproxErr}$ measures the approximation error of the policy class $\mathcal{B}$ to the expert policy $\pi^E$, and $\mathrm{EstimErr}$ is an estimation error term that vanishes with the number of expert annotation examples and iterations increasing. By Proposition 2, for LOGGER-M and LOGGER-ME, their $\mathrm{ApproxErr}$ terms are both $\mu H \cdot \mathrm{Bias}(\mathcal{B}, \Pi_\mathcal{B}, N)$. For their $\mathrm{EstimErr} = \frac{H \cdot \mathrm{SReg}_N(\mathcal{B})}{N}$, we use Theorems 9 and 11 to calculate the minimum total numbers of interaction rounds $I(\epsilon)$, expert annotations $A(\epsilon)$, and oracle calls $C(\epsilon)$, so that $\mathrm{EstimErr}$ is at most $\epsilon$ with probability at least $1 - \delta$. As presented in Table 1, LOGGER-ME has the same sample complexity order as LOGGER-M, but has a much lower interaction round complexity ($\mu H^2/\epsilon$ vs. $\mu^2 H^2/\epsilon^2$).

On the other hand, by standard ERM analysis [58] and conversion from supervised learning to imitation learning guarantees ([65, 34]), behavior cloning on $\mathcal{B}$ using $K$ samples outputs a policy $\hat{\pi}$, such that Equation (2) holds with $\mathrm{ApproxErr} = H^2 \cdot \mathrm{Bias}(\mathcal{B}, \{\pi^E\}, 1)$, and $\mathrm{EstimErr} = H^2 \sqrt{2 \ln(2B/\delta)/K}$ (see Appendix E.3 for a detailed derivation), where $\mathrm{Bias}(\mathcal{B}, \{\pi^E\}, 1) = \min_{h \in \mathcal{B}} \mathbb{E}_{s \sim d_{\pi_E}} \left[ I(h(s) \neq \pi^E(s)) \right]$. We also summarize behavior cloning's performance guarantees in Table 1. Compared with the two interactive IL algorithms above, behavior cloning requires only one interaction round and one call to the oracle $\mathcal{O}$, however its $\mathrm{ApproxErr}$ has a larger coefficient on the optimal classification loss ($H^2$ vs. $\mu H$), and needs more expert annotations ($H^4/\epsilon^2$ vs. $H^2 \mu^2/\epsilon^2$) to achieve approximation error smaller than $\epsilon$ with probability at least $1 - \delta$.

| Algorithm | ApproxErr | $I(\epsilon)$ | $A(\epsilon)$ | $C(\epsilon)$ |
|-----------|-----------|---------------|---------------|---------------|
| Logger-M | $\mu H \cdot \mathrm{Bias}(\mathcal{B}, \Pi_\mathcal{B}, N)$ | $\tilde{O}(\frac{\mu^2 H^2}{\epsilon^2})$ | $\tilde{O}(\frac{\mu^2 H^2}{\epsilon^2})$ | $\tilde{O}(\frac{\mu^4 H^4}{\epsilon^4})$ |
| Logger-ME | $\mu H \cdot \mathrm{Bias}(\mathcal{B}, \Pi_\mathcal{B}, N)$ | $\tilde{O}(\frac{\mu H^2}{\epsilon})$ | $\tilde{O}(\frac{\mu^2 H^2}{\epsilon^2})$ | $\tilde{O}(\frac{\mu^2 H^5}{\epsilon^3})$ |
| Behavior cloning | $H^2 \cdot \mathrm{Bias}(\mathcal{B}, \{\pi^E\}, 1)$ | $1$ | $\tilde{O}(\frac{H^4}{\epsilon^2})$ | $1$ |

Table 1: A comparison between our algorithms and the behavior cloning baseline, in terms of approximation error, and numbers of interaction rounds $I(\epsilon)$, expert annotations $A(\epsilon)$, oracle calls $C(\epsilon)$ needed for estimation error to be at most $\epsilon$ with probability $1 - \delta$. Here $\tilde{O}(\cdot)$ hides dependences on $\ln(\mu H/\epsilon), X, A, \ln(S), \ln(B), \ln(\mu), \ln(1/\delta)$. See Appendix E.3 for the full version of the table.

## 5 Computational hardness of sublinear dynamic regret guarantees

Finally, we study dynamic regret minimization for COIL in the Logger framework. Although in the abstract continuous online learning setup, dynamic regret minimization has recently been shown to be computationally hard [16], given the peculiar linear loss structure of the Logger framework, the computational tractability of dynamic regret minimization within this framework still remains open.

We fill this gap by showing that, under a standard complexity-theoretic assumption (that PPAD-complete problems do not admit randomized polynomial-time algorithms), there do not exist polynomial-time algorithms that achieve sublinear dynamic regret in COIL. Specifically, we have:

**Theorem 12.** *Fix $\gamma > 0$, if there exist a COIL algorithm such that for any $\mathcal{M}$ and expert $\pi^E$, it interacts with $\mathcal{M}$, CSC oracle $\mathcal{O}$, expert feedback $\zeta_E(s, a) = A^E(s, a)$, and outputs a sequence of $\{\pi_n\}_{n=1}^N \in \Pi_\mathcal{B}^N$ s.t. with probability at least $1/2$,*

$$\mathrm{DReg}_N(\mathcal{B}) \leq O(\mathrm{poly}(S, A, B) \cdot N^{1-\gamma}),$$

*in $\mathrm{poly}(N, S, A, B)$ time, then all problems in PPAD are solvable in randomized polynomial time.*

The key insight behind our proof of Theorem 12 is that, achieving a sublinear dynamic regret in the COIL setup is at least as hard as finding an approximate Nash equilibrium in a two-player general-sum game, a well-known PPAD-complete problem [13]. To establish a reduction from a two-player general-sum game to a COIL problem, we carefully construct a tree-structured MDP whose "leaf states" consist of two major groups: one group has costs encoding the two players' payoffs, and the other group has a large constant cost, ensuring that any policy in $\Pi_\mathcal{B}$ with small dynamic regret encodes near-optimal strategies of both players. We refer the readers to Appendix F for details.

## 6 Conclusion

In this work, we investigate the fundamental statistical and computational limits of classification-based online imitation learning (COIL). On the positive side, we propose the Logger framework that enables the design of oracle and regret efficient COIL algorithms with different sample and interaction round complexity tradeoffs, outperforming the behavior cloning baseline. On the negative side, we establish impossibility results for sublinear static regret using proper learning in the COIL setting. We also show the computational hardness of sublinear dynamic regret guarantees in the Logger framework.

Looking forward, it would be interesting to investigate the optimality of our sample complexity and interaction round complexity guarantees; we also speculate that it is possible to relax the small separator set assumption on $\mathcal{B}$ by utilizing very recent results on smoothed online learning [11, 28]. Finally, we are also interested in empirically evaluating our algorithms.

**Acknowledgments and Disclosure of Funding.** We thank the anonymous reviewers for their constructive comments. We thank Kwang-Sung Jun, Ryn Gray, Jason Pacheco, and members of the University of Arizona machine learning reading group for helpful discussions. We thank Wen Sun for helpful communications regarding [63, Theorem 5.3] and pointing us to an updated version [6, Section 15.5]. We thank Weijing Wang for helping with illustrative figures. This work is supported by a startup funding by the University of Arizona.

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
