# Appendix

## Table of Contents

## A  Additional related works

### A.1  IL via reduction to offline learning

An algorithm is said to reduce IL to offline learning, if it interacts with the MDP and the expert to create a series of offline learning tasks, and outputs a policy whose suboptimality depends on the quality of solving the offline learning tasks. One representative example is the Behavior Cloning algorithm, where the learner learns a policy by performing offline supervised learning on a dataset drawn from the expert's state-action occupancy distribution. By [54, Theorem 2.1] (see also earlier work of [65]), Behavior Cloning's output policy's suboptimality is bounded by $H^2$ times the classification loss with respect to the expert's state-action occupancy distribution. Another example is the Forward Training algorithm of [54], where a non-stationary policy is trained incrementally. For every step of the MDP, it trains a policy by performing offline supervised learning on a dataset drawn from the state occupancy distribution at this step induced by the nonstationary policy trained for all previous steps. The output policy's suboptimality of Forward Training is bounded in terms of the averaged 0-1 loss of intermediate offline classification problems at all steps. The same paper [54] also proposed the SMILe algorithm, where the learned stationary policy is defined as a mixture of policies trained in the past, as well as the expert policy whose weight diminishes in the number of learning

rounds. At each round, the learner trains a new policy component under the state distribution induced by the learned policy and uses it to update the learned policy. In the worst case, the output policy's suboptimality guarantee is bounded in terms of the weighted average of 0-1 losses of intermediate offline classification problems. As made explicit by [54], the SEARN [23] algorithm can be applied to imitation learning and its suboptimality guarantee can be bounded in terms of the averaged 0-1 loss of intermediate offline classification problems. Later, [37] extends SEARN to continuous-action regime under the setting of exogenous input, following the similar reduction as in [23].

**IL via reduction to offline surrogate loss minimization.** A few works study IL via offline learning that performs minimization over surrogate losses of 0-1 loss. [71], [6, Theorem 15.3] show that if the average KL divergence between a policy's action distribution and the expert's action distribution is bounded, the policy's suboptimality can in turn be bounded. In addition, under a realizable setting, maximum likelihood estimation (log loss minimization) ensures that the above KL divergence goes to zero as the training sample size grows to infinity. [71] also shows policy suboptimality bounds of generative adversarial imitation learning [30] that depends on the approximation power of the policy class, and an estimation error term that depends on the sample size and the expressivity of the discriminator class.

## A.2   IL via reduction to online learning

A major line of research [54, 56, 55, 63, 15, 18, 17] reduces interactive IL to an online learning problem, where a sequence of online losses are carefully constructed so that the cumulative online loss of a sequence of policies corresponds to the policy sequence's cumulative imitation losses. In the discrete action setting, where policies can be viewed as classifiers, early works such as DAGGER [54] do not directly provide an explicit algorithm for online cost-sensitive classification loss regret minimization, and instead perform regret minimization over convex surrogates of the classification losses. The convex surrogate minimization approach is well-known to be statistically inconsistent even in supervised learning, a special case of imitation learning [7]. Subsequent works [55] reduces online cost-sensitive classification in imitation learning to online least squares regression.

In contrast to these works, we study the original regret minimization problem (induced by CSC losses) in online IL without relaxations, in the nonrealizable setting. Although Sun et al. [63, Theorem 5.2] implicitly designs COIL algorithms for general policy classes by performing online linear optimization over the convex hull of benchmark policies, we identify a subtle technical issue in their approach; we discuss it in detail in Section D.2.

**Online IL as predictable online learning:** In the above reduction from interactive IL to online learning, a key observation from prior works [e.g. 15, 16, 38] is that, online IL is a *predictable* online learning problem [19, 51]. This observation has enabled the design of more sample efficient [16, 38, 18, 17] and convergent [15] imitation learning algorithms. However, these works either assume access to an external predictive model [18, 17] or assume strong convexity of the losses [16, 38, 15], neither of which is satisfied in the COIL setting. Our MFTPL-EG algorithm achieves $O(1)$ static regret without the strong convexity assumption on the losses, and is largely inspired by the predictor-corrector framework of [17], the Mirror-Prox algorithm and extragradient methods in smooth optimization [42, 33].

**IL with dynamic regret guarantees:** Prior works in imitation learning [38, 16] have designed algorithms that achieve sublinear dynamic regret, under the assumption that the imitation losses are strongly convex in the policy parameters. While strong convexity of the losses naturally occurs in settings such as continuous control (e.g. square losses), in our COIL setting with mixed policies, the learner's loss functions do not have strong convexity.

Cheng et al. [16] show that, under an abstract continuous online learning setup, dynamic regret minimization is PPAD-hard, by a reduction from Brouwer's fixed point problem. Our computational hardness result Theorem 12 can be viewed as a strengthening of theirs, in that we show a concrete dynamic regret minimization problem induced by imitation learning in MDPs is PPAD-hard. Our reduction is also significantly different from [16]'s, in that it reduces from the 2-player mixed Nash equilibrium problem: specifically, the reduction constructs a 3-layer MDP based on the payoff matrices of the two players.

In the general online learning setting, [10, 31] design efficient gradient-based algorithms with sublinear dynamic regret guarantees, under the assumption that the sequence of online loss functions

have bounded variations. While these results appear to be promising for designing efficient sublinear dynamic regret algorithms in COIL settings, our computational hardness result strongly suggests that additional structural assumptions on the COIL problem are necessary for such guarantees.

### A.3 Other important related works

**Fundamental sample complexity limits of IL:** Recent works of [50, 49] study minimax sample complexities of realizable imitation learning in the tabular or linear policy class settings, and shows that in general, allowing the learning agent to interact with the environment does not improve the minimax sample complexity. In contrast, in settings where the (MDP, expert policy) pair has a low recoverability constant, interactivity helps reduce the minimax sample complexity. They also show that knowing the transition probability of the MDP helps reduce the minimax sample complexity. Different from their work, our work focuses on the general function approximation setting without realizability assumptions.

**Oracle-efficient online and imitation learning:** A line of works [25, 5, 66, 24, 52, 67] design oracle-efficient online learning algorithms for online classification and online contextual bandit learning. Most of these works either assume that the context distributions are iid, or the contexts are observed ahead of time (i.e. the transductive setting), which are inapplicable in the COIL setting. The only exceptions we are aware of are [66, 24], which utilize a small separator set assumption of the benchmark policy class. However, as we have seen, a direct application of [66, 24] to the online IL setting results in a linear regret (Theorem 4), which motivates our design of MFTPL and MFTPL-EG algorithms. [62] designs oracle-efficient imitation learning algorithms from experts' state observations alone (without seeing experts' actions). Different from ours, their work makes a (strong) realizability assumption: the learner is given access to a policy class and a value function class, that contain the expert's policy and value function, respectively. Also, their algorithm requires regularized CSC oracle, for running FTRL.

**Connections between FTRL and FTPL:** In online linear optimization, [2] first observe that an in-expectation version of FTPL is equivalent to FTRL, where the regularizer depends on the distribution of the noise perturbation. This viewpoint yields a productive line of work that designs new bandit and online learning algorithms [4, 3]. Our work utilizes this connection to design oracle-efficient online imitation learning algorithms with static regret guarantees.

## B  Recap of notations and additional notations used in the proofs

We provide a brief recap of the notations introduced outside Section 2.

In Section 3, we introduce mixed policy class $\Pi_{\mathcal{B}} := \left\{ \pi_u(\cdot|s) := \sum_{h\in\mathcal{B}} u[h] \cdot h(\cdot|s) : u \in \Delta(\mathcal{B}) \right\}$, and cost vector $\theta(v) := \left( \mathbb{E}_{s\sim d_{\pi_v}} \left[ \zeta_E(s, h(s)) \right] \right)_{h\in\mathcal{B}}$, which is induced by the distribution occupancy of $\pi_v \in \Pi_{\mathcal{B}}$, expert feedback $\zeta_E$, and $\mathcal{B}$. Also, we define cost vector induced by CSC dataset $D$ and $\mathcal{B}$ as $g := \left( \mathbb{E}_{(s,\vec{c})\sim D} \left[ \vec{c}(h(s)) \right] \right)_{h\in\mathcal{B}}$.

In Section 4, we introduce algorithm MFTPL and separator set $\mathcal{X}$. Given a deterministic stationary benchmark policy class $\mathcal{B}$, its separator $\mathcal{X}$ satisfies $\forall h, h' \in \mathcal{B}, \exists x \in \mathcal{X}$, s.t. $h(x) \neq h'(x)$. Define sample based perturbation loss variables $\ell_x \sim \mathcal{N}(0, I_A)$ for each $x \in \mathcal{X}$. Denote $\ell = (\ell_x)_{x\in\mathcal{X}} \sim \mathcal{N}(0, I_{XA})$, and the induced perturbation vector $q(\ell) := (\sum_{x\in\mathcal{X}} \ell_x(h(x)))_{h\in\mathcal{B}}$, where $\ell_x(a)$ denotes the $a$-th term of $\ell_x$. When it is clear from context, we abbreviate $q(\ell)$ as $q$. Define perturbation samples set $Z = \left\{ (x, \frac{K}{\eta} \cdot \ell_x) \right\}_{x\in\mathcal{X}}$, where $K$ is the sample budget per round and $\eta$ is the learning rate. Additionally, we use $Z_j = \left\{ (x, \frac{K}{\eta} \cdot \ell_{x,j}) \right\}_{x\in\mathcal{X}}, j = 1, \ldots, T$ to index $T$ perturbation sets induced by $T$ draws of $\ell_j = (\ell_{x,j})_{x\in\mathcal{X}} \sim \mathcal{N}(0, I_{XA})$. Similarly, we abbreviate $q(\ell_j)$ as $q_j$.

We denote by $\Pr(E)$ the probability of event $E$ happening. For function $f$, we say

1. $f(n) = O(\mathrm{poly}(n))$ if $\exists C > 0$ s.t. $f(n) = O(n^C)$;
2. $f(n_1, \ldots, n_k) = O(\mathrm{poly}(n_1, \ldots, n_k))$ if $f(n_1, \ldots, n_k) = O(\mathrm{poly}(n_1 \times \ldots \times n_k))$;
3. $f(n_1, \ldots, n_k) = O(\mathrm{polylog}(n_1, \ldots, n_k))$ if $f(n_1, \ldots, n_k) = O(\mathrm{poly}(\ln n_1, \ldots, \ln n_k))$.

We summarize frequently-used definitions in the main paper and the appendix in Table 2.

Table 2: A review of notations in this paper.

| Name | Description | Name | Description |
|---|---|---|---|
| $\mathcal{M}$ | Markov decision process | $(x, \vec{c})$ | CSC example |
| $H$ | Episode length | $\mathcal{D}_\pi^E$ | $(x, \vec{c})$ distribution induced by $\pi$, $\mathcal{M}$ and $\pi^E$ |
| $t$ | Time step in $\mathcal{M}$ | $\mathcal{O}$ | CSC oracle |
| $\mathcal{S}$ | State space | $\Pi_\mathcal{B}$ | Mixed policy class |
| $S$ | State space size | $u$ | Mixed policy probability weight |
| $s$ | State | $\pi_u$ | Mixed policy induced by $u$ |
| $\text{Step}(s)$ | Time step of state $s$ | $\theta(u)$ | Linear loss vector induced by $\pi_u$ |
| $\mathcal{A}$ | Action space | $K$ | Sample budget per round |
| $A$ | Action space size | $k$ | Sample iteration index |
| $a$ | Action | $D_n$ | Set of CSC examples at iteration $n$ |
| $\rho$ | Initial distribution | $\mathbb{E}_D$ | Empirical average over set $D$ |
| $P$ | Transition dynamics | $g_n$ | Estimator for $\theta(u_n)$ by $D_n$ |
| $c$ | Cost function | $f_n(\pi)$ | Estimator for $F_n(\pi)$ by $D_n$ |
| $\pi$ | Policy | $\mathcal{X}$ | Separator set for $\mathcal{B}$ |
| $\mathbb{E}_\pi$ | Expectation wrt $\pi$ | $X$ | Separator set size |
| $\mathbb{P}_\pi$ | Probability wrt $\pi$ | $T$ | Sparsification parameter |
| $d_\pi^t$ | State occupancy distribution | $j$ | Sparsification iteration number |
| $d_\pi$ | State occupancy distribution | $Z_j$ | Perturbation example set |
| $\tau$ | Trajectory | $\mathcal{N}$ | Gaussian distribution |
| $J(\pi)$ | Expected cumulative cost | $I_A$ | Identity matrix of dimension $A$ |
| $Q_\pi$ | Action value function | $I_{XA}$ | Identity matrix of dimension $X \cdot A$ |
| $V_\pi$ | State value function | $\ell_x$ | Perturbation vector drawn from $\mathcal{N}(0, I_A)$ |
| $A_\pi$ | Advantage function | $\ell$ | $\{\ell_x\}_{x \in \mathcal{X}} \sim \mathcal{N}(0, I_{XA})$ |
| $\mu$ | Recoverability for $\pi^E$ in $\mathcal{M}$ | $q(\ell)$ | Perturbation vector in $\mathbb{R}^B$ induced by $\ell$ |
| $\pi^E$ | Expert policy | $\eta$ | Learning rate |
| $A^E$ | Expert advantage function | $R$ | Closed and strongly convex function |
| $\zeta_E$ | Expert feedback function | $\text{dom}(R)$ | Effective domain of $R$ |
| $L(\pi)$ | Imitation loss of $\pi$ | $R^*$ | Fenchel conjugate of $R$ |
| $N$ | Number of learning rounds | $D_{R^*}$ | Bregman divergence of $R^*$ |
| $n$ | Learning round number | $\Phi_\mathcal{N}$ | Expected CFTPL objective function |
| $i$ | Learning round index | $\nabla\Phi_\mathcal{N}$ | Gradient of $\Phi_\mathcal{N}$ |
| $F_n(\pi)$ | Online loss function | $R_\mathcal{N}$ | $\Phi_\mathcal{N}^*$ (Fenchel conjugate of $\Phi_\mathcal{N}$) |
| $\mathcal{B}$ | Benchmark policy class | $u_n^*$ | Expectation of output from MFTPL |
| $B$ | Benchmark policy class size | $u_n$ | Output from MFTPL |
| $h$ | Policy in $\mathcal{B}$ | $\hat{g}_n$ | Optimistic estimation for $\theta(u_n)$ |
| $\text{SReg}_N(\mathcal{B})$ | Online static regret | $[N]$ | Set $\{1, 2, \cdots, N\}$ |
| $\text{DReg}_N(\mathcal{B})$ | Online dynamic regret | $I(\cdot)$ | Indicator function |
| $\text{LReg}_N$ | Linear optimization regret | $\Delta(W)$ | All probability distributions over $W$ |
| $\text{Bias}(\mathcal{B}, \mathcal{B}_0, N)$ | Approximation error | $\text{Onehot}(w, W)$ | Delta mass (one-hot vector) on $w \in W$ |
| $\delta$ | Failure probability | $u[w]$ | $w$-th term of $u \in \mathbb{R}^{|W|}$ |
| $\Pr(E)$ | Probability of event $E$ | $\Theta$ | $\mathbb{R}^d$ or $\mathbb{R}^B$ vector |

## C  Deferred materials from Section 2

**Proposition 13** (Restatement of Proposition 2). *For any $N \in \mathbb{N}^+$ and online learner that outputs $\{\pi_n\}_{n=1}^N \in \mathcal{B}_0^N$, define $\text{Bias}(\mathcal{B}, \mathcal{B}_0, N) := \max\limits_{\{v_n\}_{n=1}^N \in \mathcal{B}_0^N} \min\limits_{\pi \in \mathcal{B}} \mathbb{E}_{s \sim \bar{d}_N} \mathbb{E}_{a \sim \pi(\cdot|s)} \left[ I(a \neq \pi^E(s)) \right]$,
where $\bar{d}_N := \frac{1}{N} \sum_{n=1}^N d_{v_n}$. Then, choosing $\hat{\pi}$ uniformly at random from $\{\pi_n\}_{n=1}^N$ has guarantee:*

$$\mathbb{E}\left[ J(\hat{\pi}) - J(\pi^E) \right] \leq H \cdot \min \left\{ \mu \cdot \text{Bias}(\mathcal{B}, \mathcal{B}_0, N) + \frac{\mathbb{E}[\text{SReg}_N(\mathcal{B})]}{N}, \ \mu \cdot \text{Bias}(\mathcal{B}, \mathcal{B}_0, 1) + \frac{\mathbb{E}[\text{DReg}_N(\mathcal{B})]}{N} \right\}.$$

*Proof of Proposition 13.* By the performance difference lemma (Lemma 55), the definitions of $L$ and $F_n$, and the assumption that $A^E(s, a) \le \zeta_E(s, a)$, we have

$$\frac{1}{N} \sum_{n=1}^{N} J(\pi_n) - J(\pi^E) = H \cdot \frac{1}{N} \sum_{n=1}^{N} \mathbb{E}_{s \sim d_{\pi_n}} \mathbb{E}_{a \sim \pi_n(\cdot|s)} \left[ A^E(s, a) \right] \le \frac{H}{N} \sum_{n=1}^{N} L(\pi_n) = \frac{H}{N} \sum_{n=1}^{N} F_n(\pi_n).$$

Following the definition of $\mathrm{SReg}_N(\mathcal{B})$ and $\mathrm{DReg}_N(\mathcal{B})$ in Equation (1),

$$\frac{1}{N} \sum_{n=1}^{N} F_n(\pi_n) = \frac{1}{N} \min_{\pi \in \mathcal{B}} \sum_{n=1}^{N} F_n(\pi) + \frac{\mathrm{SReg}_N(\mathcal{B})}{N} = \frac{1}{N} \sum_{n=1}^{N} \min_{\pi \in \mathcal{B}} F_n(\pi) + \frac{\mathrm{DReg}_N(\mathcal{B})}{N}. \quad (3)$$

Since $\mathrm{Bias}(\mathcal{B}, \mathcal{B}_0, N) = \max_{\{v_n\}_{n=1}^{N} \in \mathcal{B}_0^N} \min_{\pi \in \mathcal{B}} \mathbb{E}_{s \sim \bar{d}_N} \mathbb{E}_{a \sim \pi(\cdot|s)} \left[ I(a \ne \pi^E(s)) \right]$, $\{\pi_n\}_{n=1}^{N} \in \mathcal{B}_0^N$, and our assumption that $\zeta_E(s, a) \le \mu I(a \ne \pi^E(s))$, the static regret benchmark is bounded by:

$$\begin{aligned}
\frac{1}{N} \min_{\pi \in \mathcal{B}} \sum_{n=1}^{N} F_n(\pi) &= \min_{\pi \in \mathcal{B}} \frac{1}{N} \sum_{n=1}^{N} \mathbb{E}_{s \sim d_{\pi_n}} \mathbb{E}_{a \sim \pi(\cdot|s)} \left[ \zeta_E(s, a) \right] \\
&= \min_{\pi \in \mathcal{B}} \mathbb{E}_{s \sim \bar{d}_N} \mathbb{E}_{a \sim \pi(\cdot|s)} \left[ \zeta_E(s, a) \right] \\
&\le \min_{\pi \in \mathcal{B}} \mathbb{E}_{s \sim \bar{d}_N} \mathbb{E}_{a \sim \pi(\cdot|s)} \left[ \mu \cdot I(a \ne \pi^E(s)) \right] \\
&\le \mu \cdot \max_{\{v_n\}_{n=1}^{N} \in \mathcal{B}_0^N} \min_{\pi \in \mathcal{B}} \mathbb{E}_{s \sim \bar{d}_N} \mathbb{E}_{a \sim \pi(\cdot|s)} \left[ I(a \ne \pi^E(s)) \right] \\
&= \mu \cdot \mathrm{Bias}(\mathcal{B}, \mathcal{B}_0, N).
\end{aligned}$$

Similarly, $\forall n$,

$$\begin{aligned}
\min_{\pi \in \mathcal{B}} F_n(\pi) &= \min_{\pi \in \mathcal{B}} \mathbb{E}_{s \sim d_{\pi_n}} \mathbb{E}_{a \sim \pi(\cdot|s)} \left[ \zeta_E(s, a) \right] \\
&\le \min_{\pi \in \mathcal{B}} \mathbb{E}_{s \sim d_{\pi_n}} \mathbb{E}_{a \sim \pi(\cdot|s)} \left[ \mu \cdot I(a \ne \pi^E(s)) \right] \\
&\le \mu \cdot \max_{v \in \mathcal{B}_0} \min_{\pi \in \mathcal{B}} \mathbb{E}_{s \sim d_v} \mathbb{E}_{a \sim \pi(\cdot|s)} \left[ I(a \ne \pi^E(s)) \right] \\
&= \mu \cdot \mathrm{Bias}(\mathcal{B}, \mathcal{B}_0, 1).
\end{aligned}$$

By bringing our observations back to Equation (3), we obtain

$$\frac{1}{N} \sum_{n=1}^{N} J(\pi_n) - J(\pi^E) \le \frac{1}{N} \sum_{n=1}^{N} F_n(\pi_n) \le \mu H \cdot \mathrm{Bias}(\mathcal{B}, \mathcal{B}_0, N) + \frac{H}{N} \mathrm{SReg}_N(\mathcal{B}),$$

$$\frac{1}{N} \sum_{n=1}^{N} J(\pi_n) - J(\pi^E) \le \frac{H}{N} \sum_{n=1}^{N} F_n(\pi_n) \le \mu H \cdot \mathrm{Bias}(\mathcal{B}, \mathcal{B}_0, 1) + \frac{H}{N} \mathrm{DReg}_N(\mathcal{B}).$$

Notice that $J(\pi^E)$, $\mathrm{Bias}(\mathcal{B}, \mathcal{B}_0, N)$, and $\mathrm{Bias}(\mathcal{B}, \mathcal{B}_0, 1)$ are constants, we apply the fact that given fixed sequence $\{\pi_n\}_{n=1}^{N}$, $\mathbb{E}\left[ J(\hat{\pi}) | \{\pi_n\}_{n=1}^{N} \right] = \frac{1}{N} \sum_{n=1}^{N} J(\pi_n)$ and the law of total expectation,

$$\begin{aligned}
\mathbb{E}\left[ J(\hat{\pi}) - J(\pi^E) \right] &= \mathbb{E}_{\{\pi_n\}_{n=1}^{N}} \left[ \mathbb{E}\left[ J(\hat{\pi}) | \{\pi_n\}_{n=1}^{N} \right] \right] - J(\pi^E) \\
&= \mathbb{E}_{\{\pi_n\}_{n=1}^{N}} \left[ \frac{1}{N} \sum_{n=1}^{N} J(\pi_n) \right] - J(\pi^E) \\
&\le H \cdot \min \left\{ \mu \cdot \mathrm{Bias}(\mathcal{B}, \mathcal{B}_0, N) + \frac{\mathbb{E}[\mathrm{SReg}_N(\mathcal{B})]}{N}, \ \mu \cdot \mathrm{Bias}(\mathcal{B}, \mathcal{B}_0, 1) + \frac{\mathbb{E}[\mathrm{DReg}_N(\mathcal{B})]}{N} \right\},
\end{aligned}$$

which concludes the proof. $\qquad \square$

## D  Deferred materials from Section 3

### D.1  Deferred materials from Section 3.1

**Theorem 14** (Restatement of Theorem 4). *Suppose the expert's feedback is either of the form* $\zeta_E(s,a) = \mu \cdot I(a \neq \pi^E(s))$ *or* $\zeta_E(s,a) = A^E(s,a)$. *Then, for any* $H \geq 3$, *there exists an MDP* $\mathcal{M}$ *of episode length* $H$, *a deterministic expert policy* $\pi^E$, *a benchmark policy class* $\mathcal{B}$, *such that for any learner that sequentially and possibly at random generates a sequence of policies* $\{\pi_n\}_{n=1}^N \in \mathcal{B}^N$, *its static regret satisfies* $\mathrm{SReg}_N(\mathcal{B}) = \Omega(N)$.

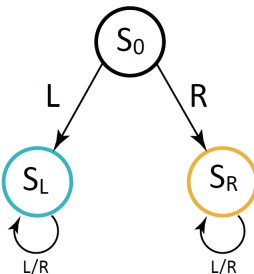

Figure 1: The MDP construction used in the proof of Theorem 14.

*Proof.* Define MDP $\mathcal{M}$ with:

- State space $\mathcal{S} = \{S_0, S_L, S_R\}$ and action space $\mathcal{A} = \{L, R\}$.
- Initial state distribution $\rho(S_0) = 1$
- Transition dynamics: $P_1(S_L|S_0, L) = 1$, $P_1(S_R|S_0, R) = 1$, i.e. playing $L$ at $S_0$ transitions to $S_L$ deterministically, while playing $R$ at $S_0$ transitions to $S_R$. Also, $\forall t \in [H-1], \forall a \in \mathcal{A}$, $P_t(S_L|S_L, a) = 1$, $P_t(S_R|S_R, a) = 1$, i.e. $S_L$ and $S_R$ have transition dynamics that are self-absorbing before termination. See Figure 1 for an illustration.
- Cost function $c(S_0, L) = c(S_0, R) = c(S_L, R) = c(S_R, L) = 0$, $c(S_L, L) = c(S_R, R) = 1$.

Meanwhile, let:

- Benchmark policy class $\mathcal{B} = \{h_L, h_R\}$, where $\forall s$, $h_L(s) = L$ and $h_R(s) = R$.
- Deterministic expert $\pi^E$ such that $\pi^E(S_0) = L$, $\pi^E(S_L) = R$ and $\pi^E(S_R) = L$.

Notice that $\forall s \in \mathcal{S}$, $c(s, \pi^E(s)) = 0$, and therefore $V_{\pi^E}(s) = 0$ for all $s$. Also, by observing $A^E(s,a) = Q_{\pi^E}(s,a) - V_{\pi^E}(s) = c(s,a)$, it can be seen that $A^E(S_0, L) = A^E(S_0, R) = A^E(S_L, R) = A^E(S_R, L) = 0$, $A^E(S_L, L) = A^E(S_R, R) = 1$.

By the transition dynamics, rolling out $h_L$ in $\mathcal{M}$ incurs trajectory $\tau_{h_L} = (S_0, L, S_L, L, \cdots, S_L, L)$ with probability 1, where $A^E(S_0, L) = I(L \neq \pi^E(S_0)) = 0$ and $A^E(S_L, L) = I(L \neq \pi^E(S_L)) = 1$. Similarly, the trajectory induced by $h_R$ is $\tau_{h_R} = (S_0, R, S_R, R, \cdots, S_R, R)$, where $A^E(S_0, R) = 0$, $I(R \neq \pi^E(S_0)) = 1$ and $A^E(S_R, R) = I(R \neq \pi^E(S_R)) = 1$.

**For the direct expert annotation feedback** $\zeta_E(s,a) = \mu \cdot I(a \neq \pi^E(s))$.

To begin with, it can be seen from the advantage function values that $(\mathcal{M}, \pi^E)$ is 1-recoverable (i.e. $\forall s \in \mathcal{S}, a \in \mathcal{A}$, $|A^E(s,a)| \leq 1$). Therefore, the feedback is of the form $\zeta_E(s,a) = I(a \neq \pi^E(s))$. Recall that $F_n(\pi) = \mathbb{E}_{s \sim d_{\pi_n}} \mathbb{E}_{a \sim \pi(\cdot|s)} [\zeta_E(s,a)]$, we follow the trajectories of $h_L, h_R$ and obtain:

- when $\pi_n = h_L$, $F_n(h_L) = \frac{H-1}{H}$, $F_n(h_R) = \frac{1}{H}$.
- when $\pi_n = h_R$, $F_n(h_R) = 1$, $F_n(h_L) = 0$.

With this, we conclude that, $\forall \{\pi_n\}_{n=1}^N \in \{h_L, h_R\}^N$, $\sum_{n=1}^N F_n(\pi_n) \geq \frac{N(H-1)}{H}$.

On the other hand, $\forall \{\pi_n\}_{n=1}^N \in \{h_L, h_R\}^N$, define $P_L := \sum_{n=1}^N \mathbb{E}_{s\sim d_{\pi_n}} I(L = \pi^E(s))$, and $P_R := \sum_{n=1}^N \mathbb{E}_{s\sim d_{\pi_n}} I(R = \pi^E(s))$. For the benchmark term, we have

$$\min_{\pi\in\mathcal{B}} \sum_{n=1}^N F_n(\pi) = \min \left( \sum_{n=1}^N F_n(\pi_L), \sum_{n=1}^N F_n(\pi_R) \right) = \min\left( P_L, P_R \right) \le \frac{N}{2},$$

where the inequality uses the observation that $P_L + P_R = N$ and therefore $\min(P_L, P_R) \le \frac{N}{2}$.

Together we obtain $SReg_N(\mathcal{B}) = \sum_{n=1}^N F_n(\pi_n) - \min_{\pi\in\mathcal{B}} \sum_{n=1}^N F_n(\pi) \ge \frac{N(H-1)}{H} - \frac{N}{2} = \frac{N(H-2)}{2H}$, which is linear in $N$ when $H \ge 3$.

**For the feedback of the form $\zeta_E(s, a) = A^E(s, a)$.**

By bringing in $\zeta_E(s, a) = A^E(s, a)$, we obtain $F_n(\pi) = \mathbb{E}_{s\sim d_{\pi_n}} \mathbb{E}_{a\sim\pi(\cdot|s)} \left[ A^E(s, a) \right]$, following the trajectories of $h_L, h_R$, it can be seen that:

- when $\pi_n = h_L$, $F_n(h_L) = \frac{H-1}{H}$, $F_n(h_R) = 0$.
- when $\pi_n = h_R$, $F_n(h_R) = \frac{H-1}{H}$, $F_n(h_L) = 0$.

This implies $\forall \{\pi_n\}_{n=1}^N \in \{h_L, h_R\}^N$, $\sum_{n=1}^N F_n(\pi_n) = \frac{N(H-1)}{H}$.

On the other hand, $\forall \{\pi_n\}_{n=1}^N \in \{h_L, h_R\}^N$, define $C_L := \sum_{n=1}^N I(\pi_n = h_L)$, and $C_R := \sum_{n=1}^N I(\pi_n = h_R)$, where $C_L + C_R = N$. For the benchmark term, we have

$$\min_{\pi\in\mathcal{B}} \sum_{n=1}^N F_n(\pi) = \min \left( \sum_{n=1}^N F_n(\pi_L), \sum_{n=1}^N F_n(\pi_R) \right) = \min\left( C_L \frac{H-1}{H}, C_R \frac{H-1}{H} \right) \le \frac{N}{2}\cdot\frac{H-1}{H},$$

where the inequality uses the observation that $C_L + C_R = N$ and therefore $\min(C_L, C_R) \le \frac{N}{2}$.

In conclusion, any online proper learning algorithm satisfies $SReg_N(\mathcal{B}) = \sum_{n=1}^N F_n(\pi_n) - \min_{\pi\in\mathcal{B}} \sum_{n=1}^N F_n(\pi) \ge \frac{N(H-1)}{H} - \frac{N(H-1)}{2H} = \frac{N(H-1)}{2H}$, i.e., it suffers regret $SReg_N(\mathcal{B}) = \Omega(N)$ which is linear in $N$ when $H \ge 2$. $\qquad\square$

### D.2 Deferred materials from Section 3.2

**An alternative mixed policy class and its issues.** Prior work [63, Theorem 5.3] propose to use an alternative definition of mixed policy class

$$\tilde{\Pi}_\mathcal{B} = \left\{ \sigma_u : u \in \Delta(B) \right\},$$

where policy $\sigma_u$ is executed in an an episode of an MDP by: draw $h \sim u$ at the beginning of the episode, and execute policy $h$ throughout the episode. Importantly, $\sigma_u$ is not a stationary policy; as a result, $\{a_t\}_{t=1}^H$ are *dependent* conditioned on $\{s_t\}_{t=1}^H$; $\{a_t\}_{t=1}^H$ are only conditionally independent given $\{s_t\}_{t=1}^H$ and $h$.

By the definition of $\sigma_u$, $J(\sigma_u)$ is a weighted combination of $J(h)$ over $h \in \mathcal{B}$, which can be written as $J(\sigma_u) = \sum_{h\in\mathcal{B}} u[h] \cdot J(h)$. [63, Theorem 5.3] propose to perform online optimization over the following losses, $\tilde{F}_n(\sigma_u) := \mathbb{E}_{s\sim d_{\sigma_n}} \left[ \sum_{h\in\mathcal{B}} u[h] \cdot A^E(s, h(s)) \right]$, where $\sigma_n$ denotes $\sigma_{u_n}$; specifically, they output a sequence of $\{u_n\}_{n=1}^N \subset \Delta(\mathcal{B})$,

$$\sum_{n=1}^N \tilde{F}_n(\sigma_n) - \min_{u\in\Delta(\mathcal{B})} \sum_{n=1}^N \tilde{F}_n(\sigma_u) \le \mu\sqrt{N\log(B)}.$$

where $\sup_{s,a} |A^E(s, a)| \le \mu$.

We show by our MDP example in Figure 1 above, in general, $J(\sigma_n) - J(\pi^E) \ne H \cdot \tilde{F}_n(\sigma_n)$, which implies that, an online optimization guarantee for $\{\tilde{F}_n(\sigma_u)\}_{n=1}^N$ cannot be converted to a policy

suboptimality guarantee. In contrast, in our LOGGER framework, with the setting of $\zeta_E = A^E$, we always have that $J(\pi_n) - J(\pi^E) = H \cdot F_n(\pi_n)$, which guarantees the conversion.[4]

Consider the MDP in the proof of Theorem 14 and $u_n = (0.5, 0.5)$. Here, policy $\sigma_n$ is executed by picking $h \in \{h_L, h_R\}$ uniformly at random and executing $h$ through the whole episode. Since $J(h_L) = J(h_R) = H - 1$ and $J(\pi^E) = 0$, we obtain $J(\sigma_n) - J(\pi^E) = \sum_{h \in \{h_L, h_R\}} u_n[h] \cdot \left(J(h) - J(\pi^E)\right) = H - 1$. On the other side, it can be shown that $d_{\sigma_n}$ distributes on $\{S_0, S_L, S_R\}$ with probability weight $(\frac{1}{H}, \frac{H-1}{2H}, \frac{H-1}{2H})$, where

$$\sum_{h \in \{h_L, h_R\}} u_n[h] \cdot A^E(S_0, h(s)) = 0,$$

and

$$\sum_{h \in \{h_L, h_R\}} u_n[h] \cdot A^E(S_L, h(s)) = \sum_{h \in \{h_L, h_R\}} u_n[h] \cdot A^E(S_R, h(s)) = \frac{1}{2}.$$

Thus it can be verified that $\tilde{F}_n(\sigma_n) = \mathbb{E}_{s \sim d_{\sigma_n}} \left[\sum_{h \in \{h_L, h_R\}} u_n[h] \cdot A^E(s, h(s))\right] = 2 \cdot \frac{H-1}{2H} \cdot \frac{1}{2} = \frac{H-1}{2H}$. By this we conclude $J(\sigma_n) - J(\pi^E) \neq H \cdot \tilde{F}(\sigma_n)$.

**Proof of Proposition 6.** We begin by stating a more precise version of Proposition 6.

**Proposition 15** (Restatement of Proposition 6). *For any $\delta \in (0, 1]$, if Algorithm 2 uses online linear optimization algorithm OLOA that outputs $\{u_n\}_{n=1}^N \subset \Delta(\mathcal{B})^N$ s.t. with probability at least $1 - \delta/3$,*

$$\text{LReg}_N = \sum_{n=1}^N \langle g_n, u_n \rangle - \min_{u \in \Delta(\mathcal{B})} \sum_{n=1}^N \langle g_n, u \rangle \leq \text{Reg}(N).$$

*Then, with probability at least $1 - \delta$, its output policies $\{\pi_n\}_{n=1}^N$ satisfy*

$$\text{SReg}_N(\mathcal{B}) \leq \text{Reg}(N) + 2\mu\sqrt{\frac{2N\ln(\frac{6}{\delta})}{K}} + 2\mu\sqrt{2N\frac{\ln(B) + \ln(\frac{6}{\delta})}{K}} = \text{Reg}(N) + O\left(\mu\sqrt{\frac{N\ln(B/\delta)}{K}}\right).$$

*Proof.* Recall that in online IL, the loss at round $n$ is $F_n(\pi) = \mathbb{E}_{s \sim d_{\pi_n}} \mathbb{E}_{a \sim \pi(\cdot|s)} [\zeta_E(s, a)]$; For $\pi_u \in \Pi_{\mathcal{B}}$,

$$F_n(\pi_u) = \sum_{h \in \mathcal{B}} u[h] \cdot \mathbb{E}_{s \sim d_{\pi_v}} \mathbb{E}_{a \sim h(\cdot|s)} [\zeta_E(s, a)] = \langle \theta(u_n), u \rangle,$$

where $\theta(v) := \left(\mathbb{E}_{s \sim d_{\pi_n}} \mathbb{E}_{a \sim h(\cdot|s)} [\zeta_E(s, a)]\right)_{h \in \mathcal{B}}$.

In LOGGER, $g_n = \left(\mathbb{E}_{(s, \vec{c}) \sim D_n} \mathbb{E}_{a \sim h(\cdot|s)} [\vec{c}(a)]\right)_{h \in \mathcal{B}}$ is our unbiased estimator for $\theta(u_n)$. By defining $f_n(\pi) := \mathbb{E}_{(s, \vec{c}) \sim D_n} \mathbb{E}_{a \sim \pi(\cdot|s)} [\vec{c}(a)]$, it can be seen that $f_n(\pi_u) = \sum_{h \in \mathcal{B}} u[h] \cdot \mathbb{E}_{(s, \vec{c}) \sim D_n} \mathbb{E}_{a \sim h(\cdot|s)} [\vec{c}(a)] = \langle g_n, u \rangle$.

Since the static regret is defined as $\text{SReg}_N(\mathcal{B}) = \sum_{n=1}^N F_n(\pi_n) - \min_{h \in \mathcal{B}} \sum_{n=1}^N F_n(h)$, where

$$\min_{h \in \mathcal{B}} \sum_{n=1}^N F_n(h) = \min_{u \in \Delta(\mathcal{B})} \sum_{n=1}^N \langle \theta(u_n), u \rangle = \min_{u \in \Delta(\mathcal{B})} \sum_{n=1}^N F_n(\pi_u) = \min_{\pi \in \Pi_{\mathcal{B}}} \sum_{n=1}^N F_n(\pi).$$

We write the static regret as

$$\text{SReg}_N(\mathcal{B}) = \sum_{n=1}^N F_n(\pi_n) - \min_{\pi \in \Pi_{\mathcal{B}}} \sum_{n=1}^N F_n(\pi)$$

$$= \underbrace{\sum_{n=1}^N (F_n(\pi_n) - f_n(\pi_n))}_{(1)} + \underbrace{\sum_{n=1}^N f_n(\pi_n) - \min_{\pi \in \Pi_{\mathcal{B}}} \sum_{n=1}^N f_n(\pi)}_{(2)} + \underbrace{\min_{\pi \in \Pi_{\mathcal{B}}} \sum_{n=1}^N f_n(\pi) - \min_{\pi \in \Pi_{\mathcal{B}}} \sum_{n=1}^N F_n(\pi)}_{(3)}.$$

---

[4]Note that the performance difference lemma (Lemma 55) requires the two policies in comparison to be stationary.

We will bound each term respectively. First, for (2), we recognize that it equals to $\text{LReg}_N = \sum_{n=1}^{N}\langle g_n, u_n\rangle - \min_{u\in\Delta(\mathcal{B})}\sum_{n=1}^{N}\langle g_n, u\rangle$, which is at most $\text{Reg}(N)$ with probability at least $1-\delta/3$ by the assumptions on OLOA.

We now bound the remaining two terms. Before going into details, we index each cost-sensitive examples as $(s_{n,k}, \vec{c}_{n,k})$ for the $k$-th sample that drawn from the $k$-th rollout trajectory at the $n$-th round, where $k\in[K]$, $n\in[N]$ and $\vec{c}_{n,k} = \big(\zeta_E(s_{n,k}, a)\big)_{a\in\mathcal{A}}$. With this notation, we can write cost-sensitive examples generated at round $n$ as $D_n = \big((s_{n,k}, \vec{c}_{n,k})\big)_{k=1}^{K}$ and write $f_n(\pi) = \mathbb{E}_{(s,\vec{c})\sim D_n}\mathbb{E}_{a\sim\pi(\cdot|s)}\big[\vec{c}(a)\big] = \frac{1}{K}\sum_{k=1}^{K}\mathbb{E}_{a\sim\pi(\cdot|s_{n,k})}\big[\zeta_E(s_{n,k}, a)\big]$.

Also, denote $\mathbb{E}_{n,k}[Y]$ as the conditional expectation of random variable $Y$ on all history before the $k$-th rollout of the $n$-th round. More precisely, denote by $\mathcal{U}_{n,k} = \{s_{n',k'} : (n', k') \preceq (n, k)\}$, where $\preceq$ denotes precedence in dictionary order, i.e., $(n_1, k_1) \preceq (n_2, k_2)$ if and only if $n_1 < n_2$, or $n_1 = n_2$ and $k_1 \le k_2$; and $\mathbb{E}_{n,k}[\cdot] := \mathbb{E}[\cdot \mid \mathcal{U}_{n,k-1}]$. As a convention, denote by $\mathcal{U}_{n,0} := \mathcal{U}_{n-1,K}$. By the assumption of $\forall s\in\mathcal{S}, \forall a\in\mathcal{A}$, $A^E(s,a) \le \zeta_E(s,a) \le \mu\cdot I(a\neq\pi^E(s))$ and $|A^E(s,a)| \le \mu$ (recall Section 2), we have that $|\zeta_E(s,a)| \le \mu$ for all $s, a$.

**Term (1):** $\sum_{n=1}^{N}(F_n(\pi_n)-f_n(\pi_n))$. We define $Y_{n,k} := F_n(\pi_n)-\mathbb{E}_{a\sim\pi_n(\cdot|s_{n,k})}\big[\zeta_E(s_{n,k}, a)\big]$ where $s_{n,k}\sim d_{\pi_n}$. It can be seen from the representation of $f_n(\pi) = \frac{1}{K}\sum_{k=1}^{K}\mathbb{E}_{a\sim\pi(\cdot|s_{n,k})}\big[\zeta_E(s_{n,k}, a)\big]$ that

$$F_n(\pi_n) - f_n(\pi_n) = \frac{1}{K}\sum_{k=1}^{K}\Big(F_n(\pi_n) - \mathbb{E}_{a\sim\pi_n(\cdot|s_{n,k})}\big[\zeta_E(s_{n,k}, a)\big]\Big) = \frac{1}{K}\sum_{k=1}^{K}Y_{n,k}.$$

Since $\pi_n$ only depends on history until $n-1$ round, and $s_{n,k}$ are iid drawn from $d_{\pi_n}$, we have

$$\begin{aligned}
\mathbb{E}_{n,k}\big[Y_{n,k}\big] &= \mathbb{E}\Big[F_n(\pi_n) - \mathbb{E}_{a\sim\pi_n(\cdot|s_{n,k})}\big[\zeta_E(s_{n,k}, a)\big] \mid \mathcal{U}_{n,k-1}\Big] \\
&= \mathbb{E}\Big[F_n(\pi_n) - \mathbb{E}_{s\sim d_{\pi_n}}\mathbb{E}_{a\sim\pi_n(\cdot|s_{n,k})}\big[\zeta_E(s, a)\big] \mid \mathcal{U}_{n-1,K}\Big] \\
&= \mathbb{E}\big[F_n(\pi_n) - F_n(\pi_n) \mid \mathcal{U}_{n-1,K}\big] = 0.
\end{aligned}$$

By applying $\|\zeta_E(s_{n,k}, \cdot)\|_\infty \le \mu$, we have

$$\begin{aligned}
|Y_{n,k}| &= |F_n(\pi_n) - \mathbb{E}_{a\sim\pi_n(\cdot|s_{n,k})}\big[\zeta_E(s_{n,k}, a)\big]| \\
&= |\mathbb{E}_{s\sim d_{\pi_n}}\mathbb{E}_{a\sim\pi_n(\cdot|s)}\big[\zeta_E(s, a)\big] - \langle\zeta_E(s_{n,k}, \cdot), \pi_n(\cdot \mid s_{n,k})\rangle| \\
&\le |\mathbb{E}_{s\sim d_{\pi_n}}\langle\zeta_E(s, \cdot), \pi_n(\cdot \mid s)\rangle| + \|\zeta_E(s_{n,k}, \cdot)\|_\infty \\
&\le 2\|\zeta_E(s_{n,k}, \cdot)\|_\infty \le 2\mu.
\end{aligned}$$

This implies the sequence of random variables $\{Y_{1,1}, Y_{1,2}, \cdots, Y_{1,K}, Y_{2,1}, \cdots, Y_{N,K}\}$ form a martingale difference sequence. Applying Azuma-Hoeffding's inequality, we get with probability at least $1-\delta/3$,

$$\left|\sum_{n=1}^{N}(F_n(\pi_n) - f_n(\pi_n))\right| = \frac{1}{K}\left|\sum_{n=1}^{N}\sum_{k=1}^{K}Y_{n,k}\right| \le 2\mu\sqrt{\frac{2N\ln(\frac{6}{\delta})}{K}}.$$

**Term (3):** $\min_{\pi\in\Pi_\mathcal{B}}\sum_{n=1}^{N}f_n(\pi) - \min_{\pi\in\Pi_\mathcal{B}}\sum_{n=1}^{N}F_n(\pi)$. Similar to term (1), for any $h\in\mathcal{B}$, we define $\hat{Y}_{n,k}(h) := F_n(h) - c_{n,k}(h(s_{n,k}), s_{n,k})$ where $s_{n,k}\sim d_{\pi_n}$. Also, we have that $f_n(\pi) = \frac{1}{K}\sum_{k=1}^{K}\mathbb{E}_{a\sim\pi(\cdot|s_{n,k})}\big[\zeta_E(s_{n,k}, a)\big]$, which implies $F_n(h) - f_n(h) = \frac{1}{K}\sum_{k=1}^{K}\hat{Y}_{n,k}(h)$. Following the same analysis shown in term (1), it can be shown that $\mathbb{E}_{n,k}\big[\hat{Y}_{n,k}(h)\big] = 0$ and $|\hat{Y}_{n,k}(h)| \le 2\mu$. By applying Azuma-Hoeffding's inequality, we get for any given $h\in\mathcal{B}$, with probability at least $1-\frac{\delta}{3B}$ (recall that $B = |\mathcal{B}|$),

$$\left|\sum_{n=1}^{N}(F_n(h) - f_n(h))\right| = \frac{1}{K}\left|\sum_{n=1}^{N}\sum_{k=1}^{K}\hat{Y}_{n,k}(h)\right| \le 2\mu\sqrt{2N\frac{\ln(B) + \ln(\frac{6}{\delta})}{K}}.$$

By applying union bound over all $h \in \mathcal{B}$, we get with probability at least $1 - \frac{\delta}{3}$, $\sum_{n=1}^{N}(F_n(h) - f_n(h)) \leq 2\mu\sqrt{2N\frac{\ln(B) + \ln(\frac{6}{\delta})}{K}}$, $\forall h \in \mathcal{B}$. Also, by the fact that $f_n(\pi) = \langle g_n, u \rangle$, it can be shown that

$$\min_{\pi \in \Pi_{\mathcal{B}}} \sum_{n=1}^{N} f_n(\pi) = \min_{u \in \Delta(\mathcal{B})} \sum_{n=1}^{N} \langle g_n, u \rangle = \min_{h \in \mathcal{B}} \sum_{n=1}^{N} f_n(h).$$

Since $\min_{\pi \in \Pi_{\mathcal{B}}} \sum_{n=1}^{N} F_n(\pi) = \min_{h \in \mathcal{B}} \sum_{n=1}^{N} F_n(h)$, by denoting $h^* \in \operatorname{argmin}_{h \in \mathcal{B}} \sum_{n=1}^{N} F_n(h)$, we conclude with probability at least $1 - \delta/3$,

$$
\begin{aligned}
\min_{\pi \in \Pi_{\mathcal{B}}} \sum_{n=1}^{N} f_n(\pi) - \min_{\pi \in \Pi_{\mathcal{B}}} \sum_{n=1}^{N} F_n(\pi) &= \min_{h \in \mathcal{B}} \sum_{n=1}^{N} f_n(h) - \min_{h \in \mathcal{B}} \sum_{n=1}^{N} F_n(h) \\
&= \min_{h \in \mathcal{B}} \sum_{n=1}^{N} f_n(h) - \sum_{n=1}^{N} f_n(h^*) + \sum_{n=1}^{N} (f_n(h^*) - F_n(h^*)) \\
&\leq 0 + 2\mu\sqrt{2N\frac{\ln(B) + \ln(\frac{6}{\delta})}{K}}.
\end{aligned}
$$

Finally, by combining our high probability bounds on terms (1),(2), and (3), applying union bound, we conclude that with probability at least $1 - \delta$,

$$\mathrm{SReg}_N(\mathcal{B}) \leq \mathrm{Reg}(N) + 2\mu\sqrt{\frac{2N\ln(\frac{6}{\delta})}{K}} + 2\mu\sqrt{2N\frac{\ln(B) + \ln(\frac{6}{\delta})}{K}} = \mathrm{Reg}(N) + O\left(\mu\sqrt{\frac{N\ln(B/\delta)}{K}}\right).$$

$\square$

# E  Deferred materials from Section 4

## E.1  Deferred materials from Section 4.1

**A more precise version of Lemma 7.** Denote by $\ell = (\ell_x)_{x \in \mathcal{X}}$ and $q(\ell) = (\sum_{x \in \mathcal{X}} \ell_x(h(x)))_{h \in \mathcal{B}}$. Define $\Phi_{\mathcal{N}} : \mathbb{R}^B \to \mathbb{R}$ as:

$$\Phi_{\mathcal{N}}(\Theta) = \mathbb{E}_{\ell \sim \mathcal{N}(0, I_{XA})}\left[\max_{u \in \Delta(\mathcal{B})} \langle \Theta + q(\ell), u \rangle\right]. \tag{4}$$

Also, define $R_{\mathcal{N}} : \mathbb{R}^B \to \mathbb{R} \cup \{+\infty\}$ as $\Phi_{\mathcal{N}}$'s Fenchel conjugate:

$$R_{\mathcal{N}}(u) = \Phi_{\mathcal{N}}^*(u) = \sup_{\tilde{\Theta} \in \mathbb{R}^B} \langle \tilde{\Theta}, u \rangle - \Phi_{\mathcal{N}}(\tilde{\Theta}). \tag{5}$$

We will need the following two lemmas that establish properties of $\Phi_{\mathcal{N}}$ and $R$ useful in the proof of Lemma 7; for their proofs, please refer to Section G.3.

**Lemma 16.** $\Phi_{\mathcal{N}}(\Theta)$ is differentiable and $\nabla\Phi_{\mathcal{N}}(\Theta) = \mathbb{E}_{\ell \sim \mathcal{N}(0, I_{XA})}\left[\operatorname{argmax}_{u \in \Delta(\mathcal{B})} \langle \Theta + q(\ell), u \rangle\right]$.

**Lemma 17.** $R_{\mathcal{N}}(u)$ is $\sqrt{\frac{\pi}{8}}\frac{1}{AX}$-strongly convex with respect to $\|\cdot\|_1$.

**Lemma 18.** $\operatorname{argmin}_{u \in \Delta(\mathcal{B})} \left(\langle \Theta, u \rangle + R_{\mathcal{N}}(u)\right) = \nabla\Phi_{\mathcal{N}}(-\Theta)$.

We are now ready to present a more precise version of Lemma 7.

**Lemma 19** (A more precise version of Lemma 7). *Suppose Mixed CFTPL receives datasets $\{D_i\}_{i=1}^{n-1}$, separator set $\mathcal{X}$, learning rate $\eta$, sparsification parameter $T$. Then, $\forall \delta \in (0, 1]$, with probability at least $1 - \delta$, Mixed CFTPL makes $T$ calls to the cost-sensitive oracle $\mathcal{O}$, and outputs $u_n \in \Delta(\mathcal{B})$ s.t.*

$$\|\pi_{u_n}(\cdot|s) - \pi_{u_n^*}(\cdot|s)\|_1 \leq \sqrt{\frac{2A\left(\ln(S) + \ln(\frac{2}{\delta})\right)}{T}}, \quad \forall s \in \mathcal{S}, \tag{6}$$

*with*

$$u_n^* := \underset{u \in \Delta(\mathcal{B})}{\operatorname{argmin}} \left( \left\langle \eta \sum_{i=1}^{n-1} g_i, u \right\rangle + R_{\mathcal{N}}(u) \right) = \nabla \Phi_{\mathcal{N}} \left( -\eta \sum_{i=1}^{n-1} g_i \right), \tag{7}$$

*for $\Phi_{\mathcal{N}}$ and $R$ defined in Equations* (4) *and* (5).

*Proof.* In the proof, we refer to results from online linear optimization, which can be checked in Section G. By Lemma 18 and Lemma 16, for $R_{\mathcal{N}}$ defined in Equation (5) and $\Phi_{\mathcal{N}}$ defined in Equation (4),

$$\begin{aligned}
u_n^* &= \underset{u \in \Delta(\mathcal{B})}{\operatorname{argmin}} \left( \left\langle \eta \sum_{i=1}^{n-1} g_i, u \right\rangle + R_{\mathcal{N}}(u) \right) \\
&= \nabla \Phi_{\mathcal{N}} \left( -\eta \sum_{i=1}^{n-1} g_i \right) = \mathbb{E}_{\ell \sim \mathcal{N}(0, I_{XA})} \left[ \underset{u \in \Delta(\mathcal{B})}{\operatorname{argmax}} \left\langle -\eta \sum_{i=1}^{n-1} g_i + q(\ell), u \right\rangle \right].
\end{aligned} \tag{8}$$

We now turn to proving Equation (6). Recall the definition of $u_n = \frac{1}{T} \sum_{j=1}^{T} u_{n,j}$ in Algorithm 3, where $u_{n,j} = \operatorname{Onehot}(\mathcal{O}((\cup_{i=1}^{n-1} D_i) \cup \cup Z_j), \mathcal{B})$ and $Z_j$ is induced by Gaussian random variables $\ell_{x,j} \sim \mathcal{N}(0, I_A)$ for each $x \in \mathcal{X}$. Denote by $\ell_j = (\ell_{x,j})_{x \in \mathcal{X}}$.

Our proof consists of two steps: first, showing that $\mathbb{E}_{\ell_j \sim \mathcal{N}(0, I_{XA})} [u_{n,j}] = u_n^*$; second, applying the concentration inequality for Multinoulli random variables on $\|\pi_{u_n}(\cdot|s) - \pi_{u_n^*}(\cdot|s)\|_1$ for all $s \in \mathcal{S}$.

**To begin with, we first prove** $\mathbb{E}_{\ell_j \sim \mathcal{N}(0, I_{XA})} [u_{n,j}] = u_n^*$. Since each $D_i$ contains $K$ cost-sensitive examples, and $\mathcal{X}$ has size $X$, we have $(\cup_{i=1}^{n-1} D_i) \cup Z_j$ contains in total $(n-1)K + X$ examples. Since $g_i[h] = \mathbb{E}_{(s,\vec{c}) \sim D_i} [\vec{c}(h(s))]$ and $Z_j = \left\{ (x, \frac{K}{\eta} \ell_{x,j}) : x \in \mathcal{X} \right\}$, by denoting $q_j[h] := \sum_{x \in \mathcal{X}} \ell_{x,j}(h(x))$, it can be seen that

$$\begin{aligned}
&\mathbb{E}_{(x,\vec{c}) \sim (\cup_{i=1}^{n-1} D_i) \cup Z_j} \left[ \vec{c}(h(x)) \right] \\
&= \frac{1}{(n-1)K + X} \left( \sum_{i=1}^{n-1} \sum_{(x,\vec{c}) \in D_i} \vec{c}(h(x)) + \sum_{(x,\vec{c}) \in Z_j} \vec{c}(h(x)) \right) \\
&= \frac{K}{(n-1)K + X} \left( \sum_{i=1}^{n-1} \mathbb{E}_{(x,\vec{c}) \sim D_i} \left[ \vec{c}(h(x)) \right] \right) + \frac{1}{(n-1)K + X} \left( \sum_{x \in \mathcal{X}} \frac{K}{\eta} \ell_{x,j}(h(x)) \right) \\
&= \frac{K}{\eta((n-1)K + X)} \left( \eta \sum_{i=1}^{n-1} g_i[h] + q_j[h] \right).
\end{aligned}$$

By the definition of oracle $\mathcal{O}$ Definition 3 and $u_{n,j} = \operatorname{Onehot}(\mathcal{O}((\cup_{i=1}^{n-1} D_i) \cup Z_j), \mathcal{B})$,

$$\begin{aligned}
u_{n,j} &= \operatorname{Onehot} \left( \mathcal{O}((\cup_{i=1}^{n-1} D_i) \cup Z_j), \mathcal{B} \right) \\
&= \operatorname{Onehot} \left( \underset{h \in \mathcal{B}}{\operatorname{argmin}} \mathbb{E}_{(x,\vec{c}) \sim (\cup_{i=1}^{n-1} D_i) \cup Z_j} \left[ \vec{c}(h(x)) \right], \mathcal{B} \right) \\
&= \operatorname{Onehot} \left( \underset{h \in \mathcal{B}}{\operatorname{argmin}} \frac{K}{\eta((n-1)K + X)} (\eta \sum_{i=1}^{n-1} g_i[h] + q_j[h]), \mathcal{B} \right) \\
&= \operatorname{Onehot} \left( \underset{h \in \mathcal{B}}{\operatorname{argmin}} (\eta \sum_{i=1}^{n-1} g_i[h] + q_j[h]), \mathcal{B} \right) \\
&= \operatorname{Onehot} \left( \underset{h \in \mathcal{B}}{\operatorname{argmax}} (-\eta \sum_{i=1}^{n-1} g_i[h] - q_j[h]), \mathcal{B} \right).
\end{aligned}$$

By this we obtain

$$\mathbb{E}_{\ell_j \sim \mathcal{N}(0, I_{XA})} \left[ u_{n,j} \right] = \mathbb{E}_{\ell_j \sim \mathcal{N}(0, I_{XA})} \left[ \text{Onehot} \left( \underset{h \in \mathcal{B}}{\text{argmax}} (-\eta \sum_{i=1}^{n-1} g_i[h] - q_j[h]), \mathcal{B} \right) \right].$$

On the other hand, we have by Equation (8) that $u_n^* = \mathbb{E}_{\ell \sim \mathcal{N}(0, I_{XA})} \left[ \underset{u \in \Delta(\mathcal{B})}{\text{argmax}} \left\langle -\eta \sum_{i=1}^{n-1} g_i + q(\ell), u \right\rangle \right].$ By lemma 44 in Section G, under the distribution of $\ell \sim \mathcal{N}(0, I_{XA})$, $\underset{u \in \Delta(\mathcal{B})}{\text{argmin}} \left\langle -\eta \sum_{i=1}^{n-1} g_i + q(\ell), u \right\rangle$ is unique with probability 1, which is a one-hot vector. With this observation, we can write

$$u_n^* = \mathbb{E}_{\ell \sim \mathcal{N}(0, I_{XA})} \left[ \underset{u \in \Delta(\mathcal{B})}{\text{argmax}} \left\langle -\eta \sum_{i=1}^{n-1} g_i + q(\ell), u \right\rangle \right]$$

$$= \mathbb{E}_{\ell \sim \mathcal{N}(0, I_{XA})} \left[ \underset{u \in \Delta(\mathcal{B})}{\text{argmax}} \left\langle -\eta \sum_{i=1}^{n-1} g_i - q(\ell), u \right\rangle \right]$$

$$= \mathbb{E}_{\ell \sim \mathcal{N}(0, I_{XA})} \left[ \text{Onehot}(\underset{h \in \mathcal{B}}{\text{argmax}} (-\eta \sum_{i=1}^{n-1} g_i[h] + q[h]), \mathcal{B}) \right],$$

where the first equality uses the following observation: Since $\ell$ is iid from $\mathcal{N}(0, I_{XA})$, $\ell$ and $-\ell$ are equal in distribution. By observing $-q(\ell) = -\left( \sum_{x \in \mathcal{X}} \ell_x(h(x)) \right)_{h \in \mathcal{B}} = q(-\ell)$, we have that $q(\ell)$ and $-q(\ell)$ are equal in distribution. This concludes $\mathbb{E}_{\ell_j} \left[ u_{n,j} \right] = u_n^*$.

**Next, we bound** $\| \pi_{u_n}(\cdot|s) - \pi_{u_n^*}(\cdot|s) \|_1$**.** To this end, we show that for any $s \in \mathcal{S}$, $\pi_{u_{n,j}}(\cdot|s)$ is a Multinoulli random variable with expectation $\pi_{u_n^*}(\cdot|s)$, $\pi_u(\cdot|s) = \frac{1}{T} \sum_{j=1}^{T} \pi_{u_{n,j}}(\cdot|s)$, and applying concentration inequality.

Since $u_{n,j} = \text{Onehot}(\mathcal{O}((\cup_{i=1}^{n-1} D_i) \cup Z_j), \mathcal{B})$, which is a one-hot vector, by denoting $h_{n,j} = \mathcal{O}((\cup_{i=1}^{n-1} D_i) \cup Z_j)$, it can be seen that $\pi_{u_{n,j}}(\cdot|s) = \sum_{h \in \mathcal{B}} u_{n,j}[h] h(\cdot|s) = h_{n,j}(\cdot|s)$ is also a one-hot vector. Also, $\forall s \in \mathcal{S}$,

$$\mathbb{E}_{\ell_j \sim \mathcal{N}(0, I_{XA})} \left[ \pi_{u_{n,j}}(\cdot|s) \right] = \mathbb{E}_{\ell_j \sim \mathcal{N}(0, I_{XA})} \left[ \sum_{h \in \mathcal{B}} u_{n,j}[h] h(\cdot|s) \right] = \sum_{h \in B} u_n^*[h] h(\cdot|s) = \pi_{u_n^*}(\cdot|s).$$

Thus, $\pi_{u_{n,j}}(\cdot|s)$ can also be seen as Multinoulli random variable on $\Delta(\mathcal{A})$ with expectation $\pi_{u_n^*}(\cdot|s)$, and $u_n$, the empirical average of $u_{n,j}$ satisfies

$$\pi_{u_n}(\cdot|s) = \sum_{h \in \mathcal{B}} u_n[h] h(\cdot|s) = \sum_{h \in \mathcal{B}} \frac{1}{T} \sum_{i=1}^{T} u_{n,j}[h] h(\cdot|s) = \frac{1}{T} \sum_{i=1}^{T} \pi_{u_{n,j}}(\cdot|s).$$

Thus, we apply concentration inequality for Multinoulli random variables [48] (originally [70, Theorem 2.1]) on $\pi_{u_n}(\cdot|s)$ and obtain given any $s \in \mathcal{S}$, with probability at least $1 - \delta/S$,

$$\| \pi_{u_n}(\cdot|s) - \pi_{u_n^*}(\cdot|s) \|_1 < \sqrt{\frac{2A \left( \ln(S) + \ln(\frac{2}{\delta}) \right)}{T}}.$$

By applying union bound over all states in $\mathcal{S}$, we conclude that, with probability at least $1 - \delta$,

$$\| \pi_{u_n}(\cdot|s) - \pi_{u_n^*}(\cdot|s) \|_1 \leq \sqrt{\frac{2A \left( \ln(S) + \ln(\frac{2}{\delta}) \right)}{T}}, \forall s \in \mathcal{S}. \qquad \square$$

**Lemma 20** (Restatement of Lemma 8). *For any $\delta \in (0, 1]$, MFTPL, if being called for $N$ rounds, with input learning rate $\eta = \frac{1}{\mu\sqrt{NA}}(\frac{\ln(B)}{X})^{\frac{1}{4}}$ and sparsification parameter $T = \frac{N\ln(2NS/\delta)}{\sqrt{X^3\ln(B)}}$, outputs a sequence of $\{u_n\}_{n=1}^N$, such that with probability $1 - \delta$:*

$$\text{LReg}_N = \sum_{n=1}^N \langle g_n, u_n \rangle - \min_{u \in \Delta(\mathcal{B})} \sum_{n=1}^N \langle g_n, u \rangle \leq O\left(\mu\sqrt{NA}(X^3\ln(B))^{\frac{1}{4}}\right).$$

*Proof.* Denote $u_n^* = \nabla\Phi_\mathcal{N}(-\eta\sum_{i=1}^{n-1} g_i)$ the same as Equation (7), where $g_n = \left(\mathbb{E}_{(s,\vec{c})\sim D_n}\left[\vec{c}(h(s))\right]\right)_{h\in\mathcal{B}}$, we rewrite the regret as

$$\text{LReg}_N = \underbrace{\sum_{n=1}^N \langle g_n, u_n \rangle - \sum_{n=1}^N \langle g_n, u_n^* \rangle}_{(1)} + \underbrace{\sum_{n=1}^N \langle g_n, u_n^* \rangle - \min_{u \in \Delta(\mathcal{B})} \sum_{n=1}^N \langle g_n, u \rangle}_{(2)}.$$

**Term (1):** $\sum_{n=1}^N \langle g_n, u_n \rangle - \sum_{n=1}^N \langle g_n, u_n^* \rangle$. For the first term, instead of bounding it naively by $\|u_n - u_n^*\|_1$, we expand its definition and use Lemma 19 to give a tighter bound. Denote $\pi_n^* = \pi_{u_n^*}$, by Lemma 19, we guarantee that for any round $n$, with probability at least $1 - \delta/N$,

$$
\begin{aligned}
\langle g_n, u_n - u_n^* \rangle &= \mathbb{E}_{(x,\vec{c})\sim D_n}\mathbb{E}_{a\sim\pi_n(\cdot|s)}\left[\vec{c}(a)\right] - \mathbb{E}_{(x,\vec{c})\sim D_n}\mathbb{E}_{a\sim\pi_n^*(\cdot|s)}\left[\vec{c}(a)\right] \\
&= \mathbb{E}_{(x,\vec{c})\sim D_n}\left[\langle \vec{c}, \pi_n(\cdot|x) - \pi_n^*(\cdot|x)\rangle\right] \\
&\leq \mathbb{E}_{(x,\vec{c})\sim D_n}\left[\|\pi_n(\cdot|x) - \pi_n^*(\cdot|x)\|_1\|\vec{c}\|_\infty\right] \\
&\leq \mu\sqrt{2A\frac{\ln(NS) + \ln(\frac{2}{\delta})}{T}},
\end{aligned}
$$

where the last line is form applying $\|\vec{c}\|_\infty = \|\zeta_E(x, \cdot)\|_\infty \leq \mu$, and with probability at least $1 - \delta/N$, for all $s \in \mathcal{S}$, $\|\pi_{u_n}(\cdot|s) - \pi_{u_n^*}(\cdot|s)\|_1 \leq \sqrt{2A\frac{\ln(NS)+\ln(\frac{2}{\delta})}{T}}$. Then, by applying union bound for $N$ rounds, and sum over $n \in [N]$ we obtain that with probability at least $1 - \delta$,

$$\sum_{n=1}^N \langle g_n, u_n \rangle - \sum_{n=1}^N \langle g_n, u_n^* \rangle \leq \mu N\sqrt{\frac{2A\left(\ln(NS) + \ln(\frac{2}{\delta})\right)}{T}}.$$

**Term (2):** $\sum_{n=1}^N \langle g_n, u_n^* \rangle - \min_{u\in\Delta(\mathcal{B})} \sum_{n=1}^N \langle g_n, u \rangle$. By the definition of $u_n^* = \nabla\Phi_\mathcal{N}(-\eta\sum_{i=1}^{n-1} g_i)$, $u_n^*$ follows exactly the same update rule of Algorithm 6 with $\Phi_\mathcal{N}$ defined in Equation (4), on online loss $g_n$ and optimistic estimation $\hat{g}_n$ set to be 0 for all $n$. By Theorem 53, the regret of $\{u_n^*\}_{n=1}^N$ is bounded by

$$
\begin{aligned}
\sum_{n=1}^N \langle g_n, u_n^* \rangle - \min_{u\in\Delta(\mathcal{B})} \sum_{n=1}^N \langle g_n, u \rangle &\leq \frac{1}{\eta}\sqrt{2X\ln(B)} + \sum_{n=1}^N XA\eta\|g_n\|_\infty^2 \\
&\leq \frac{1}{\eta}\sqrt{2X\ln(B)} + \eta\mu^2 NXA,
\end{aligned}
$$

where we use the fact that $g_n = \left(\mathbb{E}_{(s,\vec{c})\sim D_n}\left[\vec{c}(h(s))\right]\right)_{h\in\mathcal{B}}$ has $\|g_n\|_\infty$ at most $\mu$ as $\|\zeta_E(s, \cdot)\|_\infty \leq \mu$ for all $s$.

Finally, by combining the bounds on the first and second terms together, we obtain $\forall\delta \in (0, 1]$, with probability at least $1 - \delta$,

$$\sum_{n=1}^N \langle g_n, u_n \rangle - \min_{u\in\Delta(\mathcal{B})} \sum_{n=1}^N \langle g_n, u \rangle \leq \frac{1}{\eta}\sqrt{2X\ln(B)} + \eta\mu^2 NXA + \mu N\sqrt{2A\frac{\ln(NS) + \ln(\frac{2}{\delta})}{T}}.$$

By setting $\eta = \frac{1}{\mu\sqrt{NA}}\left(\frac{\ln(B)}{X}\right)^{\frac{1}{4}}$ and $T = \frac{N\ln(2NS/\delta)}{\sqrt{X^3\ln(B)}}$, we conclude with probability at least $1 - \delta$,

$$\mathrm{LReg}_N \le \left(1 + \frac{\sqrt{2}}{2} + 1\right)\mu\sqrt{2NA}\left(X^3\ln(B)\right)^{\frac{1}{4}} = O\left(\mu\sqrt{NA}\left(X^3\ln(B)\right)^{\frac{1}{4}}\right). \qquad \square$$

**Theorem 21** (Restatement of Theorem 9). *For any $\delta \in (0, 1]$, LOGGER-M, with MFTPL setting its parameters as in Lemma 8 and $K = 1$, satisfies that: (1) with probability at least $1 - \delta$, its output $\{\pi_n\}_{n=1}^N$ satisfies: $\mathrm{SReg}(N) \le O\left(\mu\sqrt{NA\ln(1/\delta)}(X^3\ln(B))^{\frac{1}{4}}\right)$; (2) it queries $N$ annotations from expert $\pi^E$; (3) it calls the CSC oracle $\mathcal{O}$ for $\frac{N^2\ln(6NS/\delta)}{\sqrt{X^3\ln(B)}}$ times.*

*Specifically, LOGGER-M achieves $\frac{H}{N}\mathrm{SReg}_N(\mathcal{B}) \le \epsilon$ with probability at least $1 - \delta$ in $N = O\left(\frac{\mu^2 H^2 A\ln(1/\delta)\sqrt{X^3\ln(B)}}{\epsilon^2}\right)$ interaction rounds, with $O\left(\frac{\mu^2 H^2 A\ln(1/\delta)\sqrt{X^3\ln(B)}}{\epsilon^2}\right)$ expert annotations and $\tilde{O}\left(\frac{\mu^4 H^4 A^2\left(\ln(1/\delta)\right)^2\ln(S/\delta)\sqrt{X^3\ln(B)}}{\epsilon^4}\right)$ oracle calls.*

*Proof.* Following the results in Lemma 8, MFTPL, if being called for $N$ rounds with the prescribed input learning rate $\eta$ and sparsification parameter $T = \frac{N\ln(6NS/\delta)}{\sqrt{X^3\ln(B)}}$, generates a sequence of $\{u_n\}_{n=1}^N$, such that with probability at least $1 - \frac{\delta}{3}$,

$$\mathrm{LReg}_N \le O\left(\mu\sqrt{NA}(X^3\ln(B))^{\frac{1}{4}}\right).$$

By Proposition 6, LOGGER-M output policies $\{\pi_n\}_{n=1}^N$ such that with probability at least $1 - \delta$,

$$\mathrm{SReg}_N(\mathcal{B}) \le O\left(\mu\sqrt{NA}(X^3\ln(B))^{\frac{1}{4}}\right) + O\left(\mu\sqrt{\frac{N\ln(B/\delta)}{K}}\right).$$

By setting $K = 1$, we obtain

$$\mathrm{SReg}_N(\mathcal{B}) \le O\left(\mu\sqrt{NA}(X^3\ln(B))^{\frac{1}{4}}\right) + O\left(\mu\sqrt{N(\ln(B/\delta))}\right) = O\left(\mu\sqrt{NA\ln(1/\delta)}(X^3\ln(B))^{\frac{1}{4}}\right),$$

where $O\left(\mu\sqrt{N\ln(B/\delta)}\right)$ is of lower order, by $X \ge \log_A(B)$ proven in Lemma 56.

Since at each round, Algorithm 2 queries $K = 1$ annotation from the expert and calls MFTPL once, where MFTPL calls oracle $\mathcal{O}$ $T = \frac{N\ln(6NS/\delta)}{\sqrt{X^3\ln(B)}}$ times, then for a total of $N$ rounds, it calls $N$ annotations and calls oracle $\mathcal{O}$ for $\frac{N^2\ln(6NS/\delta)}{\sqrt{X^3\ln(B)}}$ times.

For the second part of the theorem, to guarantee $\frac{H}{N}\mathrm{SReg}_N(\mathcal{B}) \le \epsilon$, it suffices to let $N = O\left(\frac{\mu^2 H^2 A\ln(1/\delta)\sqrt{X^3\ln(B)}}{\epsilon^2}\right)$. The number of annotations and oracle calls follow from plugging this value of $N$ into their settings in the first part of the theorem. $\qquad \square$

## E.2 Deferred materials from Section 4.2

### E.2.1 The MFTPL-EG algorithm and its guarantees

We present MFTPL-EG (Algorithm 4), an alternative to MFTPL (Algorithm 3) for online linear optimization in COIL. Recall that in the LOGGER framework, for every $n$, the linear loss $g_n$ at round $n$ is induced by $D_n$, in that $g_n[h] = \mathbb{E}_{(s,c)\sim D_n}\left[\vec{c}(h(s))\right]$ for all $h \in \mathcal{B}$.

Specifically, at round $n$, MFTPL-EG first computes $\hat{u}_n$, the output of MFTPL on $\{g_i\}_{i=1}^{n-1}$ (line 1); different from MFTPL, instead of using this as $u_n$, it rather uses this as a *estimator* for $u_n$, which is still to be determined at this point. $\hat{u}_n$ induces policy $\hat{\pi}_n = \pi_{\hat{u}_n}$. After rolling out $\hat{\pi}_n$ in $\mathcal{M}$ and requesting expert annotations (line 2), we obtain a dataset $\hat{D}_n$, whose induced linear loss (denoted by

---

**Algorithm 4** Mixed CFTPL with Extra Gradient (abbrev. MFTPL-EG)

---

**Require:** MDP $\mathcal{M}$, expert feedback $\zeta_E$, Linear losses $\{g_i\}_{i=1}^{n-1}$ represented by datasets $\{D_i\}_{i=1}^{n-1}$ each of size $K$ (s.t. $g_i[h] = \mathbb{E}_{(s,\vec{c}) \sim D_i} \left[\vec{c}(h(s))\right]$ for all $h \in \mathcal{B}$), separator set $\mathcal{X}$, learning rate $\eta$, sparsification parameter $T$.

1: $\hat{u}_n \leftarrow \text{MFTPL}\left(\{D_i\}_{i=1}^{n-1}, \mathcal{X}, \eta, T\right)$, and let $\hat{\pi}_n \leftarrow \pi_{\hat{u}_n}$.
2: Draw $K$ examples $\hat{D}_n = \{(s, \vec{c})\}$ iid from $D_{\hat{\pi}_n, E}$, via interaction with $\mathcal{M}$ and expert $\zeta_E$.
3: **return** $u_n \leftarrow \text{MFTPL}\left(\{D_i\}_{i=1}^{n-1} \cup \hat{D}_n, \mathcal{X}, \eta, T\right)$.

---

$\hat{g}_n$), is an unbiased estimate of $\theta(\hat{u}_n)$, which by the distributional continuity property (Lemma 10), turns out to be a good estimator of $\theta(u_n)$. Finally, MFTPL-EG calls MFTPL on the linear losses $\{g_i\}_{i=1}^{n-1} \cup \{\hat{g}_n\}$ (line 3).

To analyze MFTPL-EG, We first restate and prove a distributional continuity property in COIL problems.

**Lemma 22** (Restatement of Lemma 10). *For any $u, v \in \Delta(\mathcal{B})$,*

$$\|\theta(u) - \theta(v)\|_\infty \overset{(*1)}{\leq} \mu H \max_{s \in \mathcal{S}} \|\pi_u(\cdot|s) - \pi_v(\cdot|s)\|_1 \overset{(*2)}{\leq} \mu H \|u - v\|_1.$$

*Proof.* We show $(*1)$ and $(*2)$ respectively.

For $(*1)$, recall the definition of trajectory as $\tau = \{s_1, a_1, \cdots, s_H, a_H\}$, we abuse $d_\pi(\cdot)$ to denote the distribution of trajectory induced by policy $\pi$. By denoting $\theta_u[h] = \mathbb{E}_{s \sim d_{\pi_u}} \left[\zeta_E(s, h(s))\right]$ and $\theta_v[h] = \mathbb{E}_{s \sim d_{\pi_v}} \left[\zeta_E(s, h(s))\right]$, we have for any $h \in \mathcal{B}$:

$$\left|\theta_u[h] - \theta_v[h]\right| = \left|\mathbb{E}_{s \sim d_{\pi_u}} \left[\zeta_E(s, h(s))\right] - \mathbb{E}_{s \sim d_{\pi_v}} \left[\zeta_E(s, h(s))\right]\right|$$

$$= \left|\mathbb{E}_{\tau \sim d_{\pi_u}(\cdot)} \left[\frac{1}{H} \sum_{s \in \tau} \zeta_E(s, h(s))\right] - \mathbb{E}_{\tau \sim d_{\pi_v}(\cdot)} \left[\frac{1}{H} \sum_{s \in \tau} \zeta_E(s, h(s))\right]\right|$$

$$= \left|\sum_{\tau \in (\mathcal{S} \times \mathcal{A})^H} \left((d_{\pi_u}(\tau) - d_{\pi_v}(\tau)) \cdot \frac{1}{H} \sum_{s \in \tau} \zeta_E(s, h(s))\right)\right|$$

$$\leq \left|\sum_{\tau \in (\mathcal{S} \times \mathcal{A})^H} \mu(d_{\pi_u}(\tau) - d_{\pi_v}(\tau))\right|$$

$$= \mu \cdot \|d_{\pi_u}(\cdot) - d_{\pi_v}(\cdot)\|_1 = 2\mu \cdot D_{\text{TV}}\left(d_{\pi_u}(\cdot), d_{\pi_v}(\cdot)\right).$$

where the last inequality is by $\forall s \in \mathcal{S}, \forall a \in \mathcal{A}, \left|\zeta_E(s, a)\right| \leq \mu$. Here, $D_{\text{TV}}(u, v) := \frac{1}{2}\|u - v\|_1$ denotes the Total Variance (TV) distance between two distributions.

Now, by [36, Theorem 4]:

$$D_{\text{TV}}\left(d_{\pi_u}(\cdot), d_{\pi_v}(\cdot)\right) \leq H \cdot \mathbb{E}_{s \sim d_{\pi_u}} \left[D_{\text{TV}}\left(\pi_u(\cdot \mid s), \pi_v(\cdot \mid s)\right)\right], \tag{9}$$

we utilize Equation (9) and conclude that for any $h \in \mathcal{B}$,

$$\left|\theta_u[h] - \theta_v[h]\right| \leq 2\mu \cdot D_{\text{TV}}\left(d_{\pi_u}(\cdot), d_{\pi_v}(\cdot)\right)$$

$$\leq 2\mu H \cdot \mathbb{E}_{s \sim d_{\pi_u}} \left[D_{\text{TV}}\left(\pi_u(\cdot \mid s), \pi_v(\cdot \mid s)\right)\right]$$

$$\leq 2\mu H \max_{s \in \mathcal{S}} D_{\text{TV}}\left(\pi_u(\cdot \mid s), \pi_v(\cdot \mid s)\right)$$

$$= \mu H \max_{s \in \mathcal{S}} \|\pi_u(\cdot|s) - \pi_v(\cdot|s)\|_1.$$

For (∗2), by the definition of $\pi_{u_n}(\cdot|s) = \sum_{h\in\mathcal{B}} u[h]h(\cdot|s)$, we have that $\forall s \in \mathcal{S}$,

$$\|\pi_u(\cdot|s) - \pi_v(\cdot|s)\|_1 = \|\sum_{h\in\mathcal{B}} u[h]h(\cdot|s) - \sum_{h\in\mathcal{B}} v[h]h(\cdot|s)\|_1$$

$$= \sum_{a\in\mathcal{A}} \left| \sum_{h\in\mathcal{B}} (u[h] - v[h])I(h(s) = a) \right|$$

$$\leq \sum_{a\in\mathcal{A}} \sum_{h\in\mathcal{B}} |u[h] - v[h]| \, I(h(s) = a)$$

$$= \sum_{h\in\mathcal{B}} \sum_{a\in\mathcal{A}} |u[h] - v[h]| \, I(h(s) = a)$$

$$= \sum_{h\in\mathcal{B}} |u[h] - v[h]| = \|u - v\|_1.$$

This lets us conclude

$$\|\theta(u) - \theta(v)\|_\infty = \max_{h\in\mathcal{B}} |\theta_u[h] - \theta_v[h]| \leq \mu H \max_{s\in\mathcal{S}} \|\pi_u(\cdot|s) - \pi_v(\cdot|s)\|_1 \leq \mu H \|u - v\|_1. \quad \square$$

**Lemma 23.** *Let $N \geq \mu H A \sqrt{X^3 \ln(B)}$. For any $\delta \in (0, 1]$, if* MFTPL-EG *is called for $N$ rounds, with input learning rate $\eta = \frac{1}{5\mu HAX}$, sparsification parameter $T = \frac{N^2 \ln(8NS/\delta)}{\mu HAX^3 \ln(B)}$ and sample budget $K = \frac{N \ln(8NB/\delta)}{H^2 A \sqrt{X^3 \ln(B)}}$, outputs a sequence $\{u_n\}_{n=1}^N$, such that with probability at least $1 - \delta$:*

$$\text{LReg}_N \leq O\left(\mu H A \sqrt{X^3 \ln(B)}\right).$$

*Proof.* We will follow a proof outline similar to that of Lemma 20; intuitively, we can view MFTPL-EG as approximating the execution of Algorithm 6.

At the $n$-th round in the execution of MFTPL-EG, the algorithm calls MFTPL with dataset $\{D_i\}_{i=1}^{n-1}$ to get $\hat{u}_n$ and gather extra data set $\hat{D}_n$ by rolling out $\hat{\pi}_n = \pi_{\hat{u}_n}$ in $\mathcal{M}$. Then it outputs $u_n$ by running MFTPL on $\{D_i\}_{i=1}^{n-1} \cup \hat{D}_n$.

In parallel to the definition that $g_n = \left( \mathbb{E}_{(s,\vec{c})\sim D_n} \left[ \vec{c}(h(s)) \right] \right)_{h\in\mathcal{B}}$, we denote the loss vector induced by $\hat{D}_n$ as $\hat{g}_n = \left( \mathbb{E}_{(s,\vec{c})\sim \hat{D}_n} \left[ \vec{c}(h(s)) \right] \right)_{h\in\mathcal{B}}$. Following a similar definition as Equation (7), we denote $\hat{u}_n^* := \nabla\Phi_\mathcal{N}(-\eta \sum_{i=1}^{n-1} g_i)$ and $u_n^* := \nabla\Phi_\mathcal{N}(-\eta(\sum_{i=1}^{n-1} g_i + \hat{g}_n))$. We first rewrite the online linear optimization regret in the same way as the proof of Lemma 20 using $u_n^*$,

$$\text{LReg}_N = \underbrace{\sum_{n=1}^N \langle g_n, u_n \rangle - \sum_{n=1}^N \langle g_n, u_n^* \rangle}_{(1)} + \underbrace{\sum_{n=1}^N \langle g_n, u_n^* \rangle - \min_{u\in\Delta(\mathcal{B})} \sum_{n=1}^N \langle g_n, u \rangle}_{(2)},$$

and bound terms (1) and (2) respectively.

**Term (1):** $\sum_{n=1}^N \langle g_n, u_n \rangle - \sum_{n=1}^N \langle g_n, u_n^* \rangle$. Since $u_n$ is the output from MFTPL with input dataset $\{D_i\}_{i=1}^{n-1} \cup \hat{D}_n$, separator set $\mathcal{X}$, and learning rate $\eta$, while $u_n^* := \nabla\Phi_\mathcal{N}(-\eta(\sum_{i=1}^{n-1} g_i + \hat{g}_n))$ is induced by $\{D_i\}_{i=1}^{n-1}$ and $\hat{g}_n$ is induced by $\hat{D}_n$. We apply Lemma 19 and guarantee for any given round $n$, with probability at least $1 - \frac{\delta}{4N}$, for all $s \in \mathcal{S}$, $\|\pi_{u_n}(\cdot|s) - \pi_{u_n^*}(\cdot|s)\|_1 \leq \sqrt{2A\frac{\ln(NS)+\ln(\frac{8}{\delta})}{T}}$. By applying union bound over $N$ rounds, we obtain that event $E_1$ happens with probability at least $1 - \frac{\delta}{4}$, where $E_1$ is defined as

$$E_1 : \|\pi_{u_n}(\cdot|s) - \pi_{u_n^*}(\cdot|s)\|_1 \leq \sqrt{2A\frac{\ln(NS)+\ln(\frac{8}{\delta})}{T}}, \forall n \in [N], \forall s \in \mathcal{S}. \tag{10}$$

Thus, when $E_1$ happens, $\forall n \in [N]$,

$$\langle g_n, u_n - u_n^* \rangle = \mathbb{E}_{(x,\vec{c}) \sim D_n} \mathbb{E}_{a \sim \pi_n(\cdot|s)} \left[ \vec{c}(a) \right] - \mathbb{E}_{(x,\vec{c}) \sim D_n} \mathbb{E}_{a \sim \pi_n^*(\cdot|s)} \left[ \vec{c}(a) \right]$$

$$= \mathbb{E}_{(x,\vec{c}) \sim D_n} \left[ \left\langle \vec{c}, \pi_n(\cdot|x) - \pi_n^*(\cdot|x) \right\rangle \right]$$

$$\leq \mathbb{E}_{(x,\vec{c}) \sim D_n} \left[ \|\pi_n(\cdot|x) - \pi_n^*(\cdot|x)\|_1 \|\vec{c}\|_\infty \right]$$

$$\leq \mu \sqrt{2A \frac{\ln(NS) + \ln(\frac{8}{\delta})}{T}},$$

which implies,

$$\sum_{n=1}^{N} \langle g_n, u_n \rangle - \sum_{n=1}^{N} \langle g_n, u_n^* \rangle \leq \mu N \sqrt{2A \frac{\ln(NS) + \ln(\frac{8}{\delta})}{T}}.$$

**Term (2):** $\sum_{n=1}^{N} \langle g_n, u_n^* \rangle - \min_{u \in \Delta(\mathcal{B})} \sum_{n=1}^{N} \langle g_n, u \rangle$. By the definition of $u_n^* := \nabla \Phi_{\mathcal{N}}(-\eta(\sum_{i=1}^{n-1} g_i + \hat{g}_n))$, $u_n^*$ follows the update rule of Algorithm 6 with $\Phi_{\mathcal{N}}$ defined in Equation (4), online loss $g_n$ and optimistic estimation $\hat{g}_n$. By Theorem 53, the regret is bounded by

$$\sum_{n=1}^{N} \langle g_n, u_n^* \rangle - \min_{u \in \Delta(\mathcal{B})} \sum_{n=1}^{N} \langle g_n, u \rangle$$

$$\leq \frac{1}{\eta} \sqrt{2X \ln(B)} + \sum_{n=1}^{N} \left( \eta AX \|g_n - \hat{g}_n\|_\infty^2 - \frac{1}{4\eta AX} \|u_n^* - \hat{u}_n^*\|_1^2 \right),$$

where we applied the fact that $\nabla \Phi_{\mathcal{N}}(-\eta(\sum_{i=1}^{n-1} g_i + \hat{g}_n)) - \nabla \Phi_{\mathcal{N}}(-\eta(\sum_{i=1}^{n-1} g_i)) = u_n^* - \hat{u}_n^*$.

We now turn to bound $\|g_n - \hat{g}_n\|_\infty^2$. Intuitively, $g_n$ and $\hat{g}_n$ are approximators of $\theta(u_n^*)$ and $\theta(\hat{u}_n^*)$, while by the inequality $(*2)$ of Lemma 22, $\|\theta(u_n^*) - \theta(\hat{u}_n^*)\|_\infty \leq \mu H \|u_n^* - \hat{u}_n^*\|_1$. By [18, Lemma H.3], we bound $\|g_n - \hat{g}_n\|_\infty^2$ by

$$\|g_n - \hat{g}_n\|_\infty^2 \leq 5 \cdot (\underbrace{\|g_n - \theta(u_n)\|_\infty^2}_{(a)} + \underbrace{\|\theta(u_n) - \theta(u_n^*)\|_\infty^2}_{(b)} + \underbrace{\|\theta(u_n^*) - \theta(\hat{u}_n^*)\|_\infty^2}_{(c)}$$

$$+ \underbrace{\|\theta(\hat{u}_n^*) - \theta(\hat{u}_n)\|_\infty^2}_{(\tilde{b})} + \underbrace{\|\theta(\hat{u}_n) - \hat{g}_n\|_\infty^2}_{(\tilde{a})}).$$

We group the terms in three groups: $(a)(\tilde{a})$, $(b)(\tilde{b})$, and $(c)$, and apply different techniques to bound them. For the easiest $(c)$ term, we apply Lemma 22 and bound it by $\|\theta(u_n^*) - \theta(\hat{u}_n^*)\|_\infty^2 \leq \mu^2 H^2 \|u_n^* - \hat{u}_n^*\|_1^2$.

For $(b)$ and $(\tilde{b})$, we apply inequality $(*1)$ in Lemma 22 and get

$$\|\theta(u_n) - \theta(u_n^*)\|_\infty^2 \leq \mu^2 H^2 \max_{s \in \mathcal{S}} \|\pi_{u_n}(\cdot|s) - \pi_{u_n^*}(\cdot|s)\|_1^2,$$

$$\|\theta(\hat{u}_n^*) - \theta(\hat{u}_n)\|_\infty^2 \leq \mu^2 H^2 \max_{s \in \mathcal{S}} \|\pi_{\hat{u}_n}(\cdot|s) - \pi_{\hat{u}_n^*}(\cdot|s)\|_1^2.$$

For term $(b)$, on event $E_1$, which happens with probability $1 - \frac{\delta}{4}$, we have that $\max_{s \in \mathcal{S}} \|\pi_{u_n}(\cdot|s) - \pi_{u_n^*}(\cdot|s)\|_1^2 \leq 2A \frac{\ln(NS) + \ln(\frac{8}{\delta})}{T}$ for all $n \in [N]$. For term $(\tilde{b})$, the same analysis goes through for $\hat{u}_n$ the output from MFTPL with input dataset $\{D_i\}_{i=1}^{n-1}$, and $\hat{u}_n^* := \mathbb{E}_{\ell \sim \mathcal{N}(0, I_{XA})} \left[ \operatorname{argmax}_u \left\langle \sum_{i=1}^{n-1} -g_i + \frac{1}{\eta} q(\ell), u \right\rangle \right]$. Again, applying Lemma 19 and union bound over $n \in [N]$, we guarantee that the following event $E_2$ happens with probability at least $1 - \frac{\delta}{4}$, where

$$E_2 : \max_{s \in \mathcal{S}} \|\pi_{\hat{u}_n}(\cdot|s) - \pi_{\hat{u}_n^*}(\cdot|s)\|_1^2 \leq 2A \frac{\ln(NS) + \ln(\frac{8}{\delta})}{T}, \forall n \in [N], \forall s \in \mathcal{S}.$$

In summary, for $(b)$ and $(\tilde{b})$, we conclude:

1. With probability at least $1 - \frac{\delta}{4}$, $\forall n \in \mathcal{S}$, $\|\theta(u_n) - \theta(u_n^*)\|_\infty^2 \leq 2\mu^2 H^2 A \frac{\ln(NS) + \ln(\frac{8}{\delta})}{T}$.

2. With probability at least $1 - \frac{\delta}{4}$, $\forall n \in \mathcal{S}$, $\|\theta(\hat{u}_n^*) - \theta(\hat{u}_n)\|_\infty^2 \leq 2\mu^2 H^2 A \frac{\ln(NS) + \ln(\frac{8}{\delta})}{T}$.

For $(a)$ and $(\tilde{a})$, we first introduce notation $\theta_n[h] = \mathbb{E}_{s \sim d_{\pi_n}}\left[\zeta_E(s, h(s))\right]$, $\hat{\theta}_n[h] = \mathbb{E}_{s \sim d_{\hat{\pi}_n}}\left[\zeta_E(s, h(s))\right]$, where we recall that $\pi_n = \pi_{u_n}$, $\hat{\pi}_n = \pi_{\hat{u}_n}$. Also, since $g_n[h] = \mathbb{E}_{(s,\vec{c}) \sim D_n}\left[\vec{c}(h(s))\right]$, $\hat{g}_n[h] = \mathbb{E}_{(s,\vec{c}) \sim \hat{D}_n}\left[\vec{c}(h(s))\right]$. Notice $\forall n \in [N]$ and $\forall h \in \mathcal{B}$, $\mathbb{E}g_n[h] = \theta_n[h]$ and $\mathbb{E}\hat{g}_n[h] = \hat{\theta}_n[h]$. Since $\theta_n[h]$, $\hat{\theta}_n[h]$, $g_n[h]$, $\hat{g}_n[h]$ are all in $[-\mu, \mu]$. We have by Hoeffding's Inequality, given any $n \in [N]$ and $h \in \mathcal{B}$,

1. With probability at least $1 - \frac{\delta}{4NB}$, $|g_n[h] - \theta_n[h]| \leq 2\mu\sqrt{\frac{\ln(NB) + \ln(\frac{8}{\delta})}{2K}}$.

2. With probability at least $1 - \frac{\delta}{4NB}$, $|\hat{g}_n[h] - \hat{\theta}_n[h]| \leq 2\mu\sqrt{\frac{\ln(NB) + \ln(\frac{8}{\delta})}{2K}}$.

With union bound applied over $[N]$ and all $h \in \mathcal{B}$, we obtain

1. Event $E_3$ happens with probability at least $1 - \frac{\delta}{4}$, where $E_3$ : $\|g_n - \theta(u_n)\|_\infty^2 \leq 2\mu^2\frac{\ln(NB) + \ln(\frac{8}{\delta})}{K}$, $\forall n \in [N]$.

2. Event $E_4$ happens with probability at least $1 - \frac{\delta}{4}$, where $E_4$ : $\|\theta(\hat{u}_n) - \hat{g}_n\|_\infty^2 \leq 2\mu^2\frac{\ln(NB) + \ln(\frac{8}{\delta})}{K}$, $\forall n \in [N]$.

Finally, by the union bound, event $E = E_1 \cap E_2 \cap E_3 \cap E_4$ happens with probability at least $1 - \delta$. By combining the bounds on all terms we have, we obtain that when event $E$ happens,

$$
\begin{aligned}
\text{LReg}_N \leq & \mu N \sqrt{2A\frac{\ln(NS) + \ln(\frac{8}{\delta})}{T}} + \frac{1}{\eta}\sqrt{2X\ln(B)} \\
& + \sum_{n=1}^N \left(\eta AX\|g_n - \hat{g}_n\|_\infty^2 - \frac{1}{4\eta AX}\|u_n^* - \hat{u}_n^*\|_1^2\right) \\
\leq & \mu N \sqrt{2A\frac{\ln(NS) + \ln(\frac{8}{\delta})}{T}} + \frac{1}{\eta}\sqrt{2X\ln(B)} \\
& + \sum_{n=1}^N \left(\eta AX \cdot 5\mu^2 H^2\|u_n^* - \hat{u}_n^*\|_1^2 - \frac{1}{4\eta AX}\|u_n^* - \hat{u}_n^*\|_1^2\right) \\
& + \eta AX\left(20NA\mu^2 H^2\frac{\ln(NS) + \ln(\frac{8}{\delta})}{T} + 20\mu^2 N\frac{\ln(NB) + \ln(\frac{8}{\delta})}{K}\right).
\end{aligned}
$$

By setting $\eta = \frac{1}{5\mu HAX}$, $T = \frac{N^2\ln(8NS/\delta)}{\mu HAX^3\ln(B)}$ and $K = \frac{N\ln(8NB/\delta)}{H^2 A\sqrt{X^3\ln(B)}}$, we cancel the terms related to $\|u_n^* - \hat{u}_n^*\|_1^2$ and conclude with probability at least $1 - \delta$,

$$
\text{LReg}_N \leq \mu A\sqrt{2\mu HX^3\ln(B)} + (5 + 4 + 4) \cdot \mu HA\sqrt{2X^3\ln(B)} = O\left(\mu HA\sqrt{X^3\ln(B)}\right),
$$

where $\mu A\sqrt{2\mu HX^3\ln(B)}$ is of lower order since $\mu \leq H$, and $\eta AX \cdot 20NA\mu^2 H^2\frac{\ln(NS)+\ln(\frac{8}{\delta})}{T} = \frac{4\mu^2 H^2 A^2 X^3\ln(B)}{N} \leq 4\mu HA\sqrt{2X^3\ln(B)}$ is from $N \geq \mu HA\sqrt{X^3\ln(B)}$. $\qquad\square$

**Theorem 24** (Restatement of Theorem 11). *Let $N \geq \mu HA\sqrt{X^3\ln(B)}$. For any $\delta \in (0, 1]$,* LOGGER-ME, *with* MFTPL-EG *setting its parameters as in Lemma 23, satisfies that: (1) with probability $1 - \delta$, its output $\{\pi_n\}_{n=1}^N$ satisfies that:* $\text{SReg}_N(\mathcal{B}) \leq O\left(\mu HA\sqrt{X^3\ln(B)}\right)$; *(2) it queries* $O\left(\frac{N^2\ln(NB/\delta)}{H^2 A\sqrt{X^3\ln(B)}}\right)$ *annotations from expert $\pi^E$; (3) it calls the CSC oracle $\mathcal{O}$ for $O\left(\frac{N^3\ln(NS/\delta)}{\mu HAX^3\ln(B)}\right)$*

*times.*

*Specifically,* LOGGER-ME *achieves* $\frac{H}{N}\text{SReg}_N(\mathcal{B}) < \epsilon$ *with probability* $1 - \delta$ *in* $2N =$ $O\left(\frac{\mu H^2 A\sqrt{X^3\ln(B)}}{\epsilon}\right)$ *interaction rounds, with* $\tilde{O}\left(\frac{\mu^2 H^2 A\ln(B/\delta)\sqrt{X^3\ln(B)}}{\epsilon^2}\right)$ *expert annotations and* $\tilde{O}\left(\frac{\mu^2 H^5 A^2\ln(S/\delta)\sqrt{X^3\ln(B)}}{\epsilon^3}\right)$ *oracle calls.*

*Proof.* Following the results in Lemma 23, for any $\delta \in (0, 1]$, MFTPL-EG, with the prescribed input learning rate $\eta$, sparsification parameter $T$, and sample budget $K$, outputs a sequence of $\{u_n\}_{n=1}^N$, such that with probability at least $1 - \frac{\delta}{3}$,

$$\text{LReg}_N \leq O\left(\mu H A\sqrt{X^3\ln(B)}\right).$$

By Proposition 6, LOGGER-ME with the prescribed sample budget $K = O\left(\frac{N\ln(NB/\delta)}{H^2 A\sqrt{X^3\ln(B)}}\right)$ outputs policies $\{\pi_n\}_{n=1}^N$ that satisfy with probability at least $1 - \delta$,

$$\text{SReg}_N(\mathcal{B}) \leq O\left(\mu H A\sqrt{X^3\ln(B)}\right) + O\left(\mu\sqrt{\frac{N\ln(B/\delta)}{K}}\right) = O\left(\mu H A\sqrt{X^3\ln(B)}\right),$$

where $O\left(\mu\sqrt{\frac{N\ln(B/\delta)}{K}}\right) = O\left(\mu H\sqrt{A}(X^3\ln(B))^{\frac{1}{4}}\right)$ is of lower order.

Since at each round LOGGER-ME queries $K = O\left(\frac{N\ln(NB/\delta)}{H^2 A\sqrt{X^3\ln(B)}}\right)$ annotations from the expert and calls MFTPL-EG once, where MFTPL-EG also queries $K$ annotations, together for $N$ rounds LOGGER-ME calls $O\left(\frac{N^2\ln(NB/\delta)}{H^2 A\sqrt{X^3\ln(B)}}\right)$ annotations from the expert. Also, since MFTPL-EG calls MFTPL twice, where MFTPL calls oracle $T = O\left(\frac{N^2\ln(NS/\delta)}{\mu H A X^3\ln(B)}\right)$ times, then $N$ rounds together LOGGER-ME calls oracle $O\left(\frac{N^3\ln(NS/\delta)}{\mu H A X^3\ln(B)}\right)$ times.

For the second part of the theorem, to guarantee $\frac{H}{N}\text{SReg}_N(\mathcal{B}) \leq \epsilon$, it suffices to let $N = O\left(\frac{\mu H^2 A\sqrt{X^3\ln(B)}}{\epsilon}\right)$. The number of annotations and oracle calls follow from plugging this value of $N$ into their settings in the first part of the theorem. $\square$

### E.3 Detailed comparisons between LOGGER-M, LOGGER-ME and behavior cloning

We first present a finite-sample analysis of behavior cloning by ERM on the benchmark policy class in the following proposition.

**Proposition 25** (Application of standard agnostic ERM [58])**.** *For $\mathcal{B}$ that contains finite (e.g. $B$) deterministic policies $h : \mathcal{S} \to \mathcal{A}$ and deterministic expert policy $\pi^E$, recall $\text{Bias}(\mathcal{B}, \{\pi^E\}, 1) = \min_{\pi \in \mathcal{B}} \mathbb{E}_{s \sim d_{\pi^E}}\left[I(h(s) \neq \pi^E(s))\right]$. Consider dataset $\mathcal{D} = \{(s_k, \pi^E(s_k))\}_{k=1}^K$, where $s_k \sim d_{\pi^E}$. The output $\hat{\pi}$ from running ERM on $\mathcal{D}$ satisfy with probability $1 - \delta$,*

$$J(\hat{\pi}) - J(\pi^E) \leq H^2 \cdot \text{Bias}(\mathcal{B}, \{\pi^E\}, 1) + H^2\sqrt{\frac{2(\ln(B) + \ln(\frac{2}{\delta}))}{K}}.$$

*Proof.* By standard analysis of ERM for agnostic learning (e.g. [58]), we have with probability $1 - \delta$,

$$\mathbb{E}_{s \sim d_{\pi^E}}\left[I(h(s) \neq \pi^E(s))\right] \leq \min_{\pi \in \mathcal{B}} \mathbb{E}_{s \sim d_{\pi^E}}\left[I(h(s) \neq \pi^E(s))\right] + \sqrt{\frac{2(\ln(B) + \ln(\frac{2}{\delta}))}{K}}$$

$$= \text{Bias}(\mathcal{B}, \{\pi^E\}, 1) + \sqrt{\frac{2(\ln(B) + \ln(\frac{2}{\delta}))}{K}}.$$

Table 3: A comparison between our algorithms and the behavior cloning. Here $\tilde{O}(N)$ denotes $O(N \ln(N))$.

| Algorithm | Bias Term | # Interaction Rounds $I(\epsilon)$ |
|---|---|---|
| MFTPL | $\mu H \cdot \mathrm{Bias}(\mathcal{B}, \Pi_{\mathcal{B}}, N)$ | $O\left(\frac{\mu^2 H^2 A \ln(1/\delta)\sqrt{X^3 \ln(B)}}{\epsilon^2}\right)$ |
| MFTPL-EG | $\mu H \cdot \mathrm{Bias}(\mathcal{B}, \Pi_{\mathcal{B}}, N)$ | $O\left(\frac{\mu H^2 A \sqrt{X^3 \ln(B)}}{\epsilon}\right)$ |
| Behavior Cloning | $H^2 \cdot \mathrm{Bias}(\mathcal{B}, \{\pi^E\}, 1)$ | $1$ |

| Algorithm | # Expert Annotations $A(\epsilon)$ | # Oracle Calls $C(\epsilon)$ |
|---|---|---|
| MFTPL | $O\left(\frac{\mu^2 H^2 A \ln(1/\delta)\sqrt{X^3 \ln(B)}}{\epsilon^2}\right)$ | $\tilde{O}\left(\frac{\mu^4 H^4 A^2 \ln(S/\delta)\left(\ln(1/\delta)\right)^2 \sqrt{X^3 \ln(B)}}{\epsilon^4}\right)$ |
| MFTPL-EG | $\tilde{O}\left(\frac{\mu^2 H^2 A \ln(B/\delta)\sqrt{X^3 \ln(B)}}{\epsilon^2}\right)$ | $\tilde{O}\left(\frac{\mu^2 H^5 A^2 \ln(S/\delta)\sqrt{X^3 \ln(B)}}{\epsilon^3}\right)$ |
| Behavior Cloning | $O\left(\frac{H^4 \ln(B/\delta)}{\epsilon^2}\right)$ | $1$ |

By the performance difference lemma 55, we have

$$J(\pi^E) - J(\hat{\pi}) = H \cdot \mathbb{E}_{s \sim d_{\pi^E}}\left[A_{\hat{\pi}}(s, \pi^E(s))\right],$$

where $A_{\hat{\pi}}(s, \pi^E(s)) = Q_{\hat{\pi}}(s, \pi^E(s)) - V_{\hat{\pi}}(s)$. By the fact that $A_{\hat{\pi}}$ is bounded in $[-H, H]$ and $A_{\hat{\pi}}(s, \hat{\pi}(s)) = 0$, we have

$$
\begin{aligned}
J(\hat{\pi}) - J(\pi^E) =& H \cdot \mathbb{E}_{s \sim d_{\pi^E}}\left[-A_{\hat{\pi}}(s, \pi^E(s))\right] \\
\leq& H \cdot \mathbb{E}_{s \sim d_{\pi^E}}\left[H \cdot I(h(s) \neq \pi^E(s))\right] \\
\leq& H^2 \cdot \mathrm{Bias}(\mathcal{B}, \{\pi^E\}, 1) + H^2 \sqrt{\frac{2(\ln(B) + \ln(\frac{2}{\delta}))}{K}},
\end{aligned}
$$

which concludes the proof. $\qquad\square$

We now summarize the policy suboptimality guarantees of LOGGER-M, LOGGER-ME and behavior cloning in Table 3, based on Theorems 21, 24, and Proposition 25; this extends Table 1 in the main text. In addition to the main observations made in Section 4, we see that LOGGER-ME has a factor of $O(\ln(B))$ higher sample complexity $A(\epsilon)$ than LOGGER-M. In addition, in terms of the dependence on $X$ and $\ln(B)$, the sample complexity of behavior cloning $A(\epsilon)$ has a $\ln(B)$ dependence, which is better than LOGGER-M's $\sqrt{X^3 \ln(B)}$ and LOGGER-ME's $\sqrt{X^3(\ln(B))^3}$. It would be interesting to design interactive imitation learning algorithms with sample complexity that only has a $O(\ln(B))$ dependence, by relaxing the small separator set assumption on $\mathcal{B}$.

# F    Deferred materials from Section 5

In this section, we present the proof of Theorem 12. To this end, we will show that obtaining a sublinear dynamic regret guarantee in the LOGGER framework is as hard as computing an approximate mixed Nash equilibrium of a two-player general-sum game. This is achieved by using polynomial time reduction, where we use $Y \leq_p X$ to denote that problem $Y$ is polynomial-time reducible to problem $X$. To facilitate our discussions, we start with some problem definitions.

## F.1    Preliminaries for two-player general-sum games

A two-player general-sum game [39], also known as bimatrix game, is a non-cooperative game between two players where they can choose actions (or strategies) from set $\mathcal{A}_x$, $\mathcal{A}_y$ that contain $A_x$, $A_y$ choices and gain reward from payoff matrices $V, W \in \mathbb{R}^{A_x \times A_y}$, respectively. We say a

bimatrix game $(V, W)$ is positivly normalized if $V, W \in [0, 1]^{A_x \times A_y}$. Note that we use $V_{ij}, W_{ij}$ ($i \in [A_x], j \in [A_y]$) to index each element in matrix $V$ and $W$. In the bimatrix game, if the first player plays the $i$-th action and the second player plays the $j$-th action, they receive payoffs $V_{i,j}$ and $W_{i,j}$ respectively. Define mixed strategies probability distribution on action sets. The two players are allowed to play $x \in \Delta(\mathcal{A}_x)$ and $y \in \Delta(\mathcal{A}_y)$ that corresponds to the mixed strategies on set $\mathcal{A}_x$ and $\mathcal{A}_y$, and their payoffs are $x^\top V y$ and $x^\top W y$ respectively. A Nash equilibrium of a bimatrix game $(V, W)$ is a pair $(x, y)$, where $x \in \Delta(\mathcal{A}_x)$, $y \in \Delta(\mathcal{A}_y)$, and no player can gain more payoff by changing $x$ or $y$ alone. A relaxed notion, $\epsilon$-approximate 2-player Nash equilibrium is defined below.

**Definition 26** ($\epsilon$-approximate 2-player Nash equilibrium). *For $\epsilon \geq 0$, an $\epsilon$-approximate 2-player Nash equilibrium $(\hat{x}, \hat{y})$ for a bimatrix game $(V, W)$ satisfies that for any $x \in \Delta(\mathcal{A}_x)$ and $y \in \Delta(\mathcal{A}_y)$,*

$$\begin{cases} \hat{x}^\top V \hat{y} \geq x^\top V \hat{y} - \epsilon, \\ \hat{x}^\top W \hat{y} \geq \hat{x}^\top W y - \epsilon, \end{cases}$$

*where $\hat{x} \in \Delta(\mathcal{A}_x)$, $\hat{y} \in \Delta(\mathcal{A}_y)$, and $V, W \in \mathbb{R}^{A_x \times A_y}$.*

In other words, at $(\hat{x}, \hat{y})$, by changing a players' mixed strategies unilaterally, the increase of her payoff is smaller than $\epsilon$. We consider a search problem of finding an approximate 2-player Nash equilibrium:

---

POLY$^{12}$-BIMATRIX:
Input: A positively normalized bimatrix game $(V, W)$, where $V, W \in [0, 1]^{m \times m}$, $m \in \mathbb{N}$.
Output: An $m^{-12}$-approximate 2-player Nash equilibrium of $(V, W)$.

---

It is well-known that POLY$^{12}$-BIMATRIX is a total search problem (i.e. any POLY$^{12}$-BIMATRIX instance has a solution). Furthermore, it is PPAD-complete [13, 22], which means that for any problem $Y$ in PPAD, $Y \leq_p$ POLY$^{12}$-BIMATRIX. PPAD-complete is a computational complexity class believed to be computationally intractable.

### F.2 A related variational inequality problem

The Variational Inequality (VI) formulation serves as a tool to address equilibrium problems. In this subsection, we intoduce a VI problem, which is shown to bridge POLY$^{12}$-BIMATRIX and achieving sublinear dynamic regret in the LOGGER framework in the following subsections.

**Definition 27** (Variational inequality). *Given $\Omega \subset \mathbb{R}^d$ and a vector field $\mathcal{F} : \Omega \to \mathbb{R}^d$, define $\mathrm{VI}(\Omega, \mathcal{F})$, the variational inequality problem induced by $(\Omega, \mathcal{F})$ as finding $u^* \in \Omega$, such that*

$$\forall u \in \Omega, \langle \mathcal{F}(u^*), u - u^* \rangle \geq 0.$$

**Definition 28** ($\epsilon$-approximate solution of variational inequality). *$u^* \in \Omega$ is said to be an $\epsilon$-approximate solution of $\mathrm{VI}(\Omega, \mathcal{F})$ if*

$$\forall u \in \Omega, \langle \mathcal{F}(u^*), u - u^* \rangle \geq -\epsilon.$$

Given a discrete state-action episodic MDP $\mathcal{M}$, expert feedback $\zeta_E$ and deterministic policy class $\mathcal{B}$, where $|\mathcal{B}| = B$, Section 3 defines a vector field $\theta : \Delta(\mathcal{B}) \to \mathbb{R}^B$, where $\theta(u) := \left( \mathbb{E}_{s \sim d_{\pi_u}} \mathbb{E}_{a \sim h(\cdot|s)} [\zeta_E(s, a)] \right)_{h \in \mathcal{B}}$ and $\pi_u(\cdot|s) := \sum_{h \in \mathcal{B}} u[h] \cdot h(\cdot|s)$. Setting $\Omega = \Delta(\mathcal{B}) \subset \mathbb{R}^B$ and $\mathcal{F} = \theta$, we obtain a variational inequality problem, which, as we see next, is tightly connected with the dynamic regret minimization problem in the LOGGER framework.

Note that in the LOGGER framework, if an algorithm at some round $n$ outputs a $\pi_n := \pi_{u_n} \in \Pi_\mathcal{B}$ that achieves a low instant dynamic regret guarantee: $F_n(\pi_n) - \min_{\pi \in \mathcal{B}} F_n(\pi) \leq \epsilon$, by the fact that $F_n(\pi_u) = \langle \theta(u_n), u \rangle$, we obtain $\langle \theta(u_n), u_n \rangle - \min_{u \in \Delta(\mathcal{B})} \langle \theta(u_n), u \rangle \leq \epsilon$, which is equivalent to $\forall u \in \Delta(\mathcal{B}), \langle \theta(u_n), u - u_n \rangle \geq -\epsilon$. This implies $u_n$ is a $\epsilon$-approximate solution of $\mathrm{VI}(\Delta(\mathcal{B}), \theta)$.

This motivates the following search problem of finding an $\epsilon$-approximate solution of $\mathrm{VI}(\Delta(\mathcal{B}), \theta)$:

POLY$^6$-VI-MDP:
Input: Discrete state-action episodic MDP $\mathcal{M} = (\mathcal{S}, \mathcal{A}, H, c, \rho, P)$, expert feedback $\zeta_E : \mathcal{S} \times \mathcal{A} \to \mathbb{R}$, deterministic policy class $\mathcal{B}$.
Output: A $(S + A + B)^{-6}$-approximate solution of VI$(\Delta(\mathcal{B}), \theta)$, where $S = |\mathcal{S}|, A = |\mathcal{A}|, B = |\mathcal{B}|$.

For the remainder of this section, we first establish a reduction from POLY$^{12}$-BIMATRIX to POLY$^6$-VI-MDP. Then, we show that an efficient algorithm that achieves sublinear dynamic regret in the LOGGER framework yields an efficient procedure for solving POLY$^6$-VI-MDP, and thus, all PPAD problems are solvable in randomized polynomial time.

Before diving into the reduction, we first prove that any POLY$^6$-VI-MDP instance has a solution.

**Lemma 29.** POLY$^6$-VI-MDP *is a total search problem, i.e., any* POLY$^6$-VI-MDP *problem instance has a solution.*

*Proof.* Theorem 3.1 in [26] says that, for any nonempty, convex and compact subset $\Omega \subset \mathbb{R}^n$ and continuous mapping $\mathcal{F} : \Omega \to \mathbb{R}^n$, there exists an exact solution to the VI$(\Omega, \mathcal{F})$. Then, it suffices to show that $\Omega = \Delta(\mathcal{B})$ and $\mathcal{F} = \theta$ satisfy these requirements.

First of all, by lemma 10, for any $u, v \in \Delta(\mathcal{B})$, $\|\theta(u) - \theta(v)\|_\infty \leq \mu H \|u - v\|_1$, which implies $\theta(\cdot)$ is a continuous mapping. Secondly, since it is easy to verify that $\Delta(\mathcal{B})$ is a convex set, it remains to show the compactness of $\Delta(\mathcal{B})$. By the Heine–Borel theorem, a subset in $\mathbb{R}^B$ is compact if and only if it is closed and bounded. Then, it suffices to show $\Delta(\mathcal{B})$ is closed and bounded. It can be seen that, $\Delta(\mathcal{B}) := \left\{ u \in \mathbb{R}^B \mid u \succeq 0, \sum_{h \in \mathcal{B}} u[h] = 1 \right\}$ is closed, being the intersection of closed sets, namely the orthant $\mathbb{R}^B_+$ and the hyperplane $\left\{ u \in \mathbb{R}^B \mid \sum_{h \in \mathcal{B}} u[h] = 1 \right\}$. Also, $\Delta(\mathcal{B})$ is a subset of the hypercube $[0, 1]^B$, and is therefore bounded. Combining the above, we conclude the proof. $\square$

### F.3 POLY$^6$-VI-MDP is PPAD-hard

**Theorem 30.** POLY$^6$-VI-MDP *is* PPAD-*hard.*

*Proof.* Since POLY$^{12}$-BIMATRIX is PPAD-complete by [14], we show POLY$^6$-VI-MDP is PPAD-hard by proving POLY$^{12}$-BIMATRIX $\leq_p$ POLY$^6$-VI-MDP. Our proof is organized as follows: First, we describe map f that maps an instance of POLY$^{12}$-BIMATRIX to an instance of POLY$^6$-VI-MDP, and map g that maps a solution of POLY$^6$-VI-MDP to a solution of POLY$^{12}$-BIMATRIX. Then, we prove that f and g run in polynomial time and satisfy:

1. If $(V, W)$ is an input of POLY$^{12}$-BIMATRIX, then f$(V, W)$ is an input of POLY$^6$-VI-MDP.
2. If $u^*$ is a solution of POLY$^6$-VI-MDP instance f$(V, W)$, then g$(u^*)$ is also a solution of POLY$^{12}$-BIMATRIX instance $(V, W)$.
3. If no $u$ is a solution of POLY$^6$-VI-MDP instance f$(V, W)$, then no $(x, y)$ a solution of POLY$^{12}$-BIMATRIX instance $(V, W)$.

**Map f.** Given any POLY$^{12}$-BIMATRIX instance $(V, W)$ where $V, W \in [0, 1]^{m \times m}$ we construct a POLY$^6$-VI-MDP instance where the MDP $\mathcal{M}$ can be viewed as a three-layer tree (whose details will be given shortly), and every non-leaf node in the $\mathcal{M}$ has $A = 2m + 1$ children. See Figure 2.

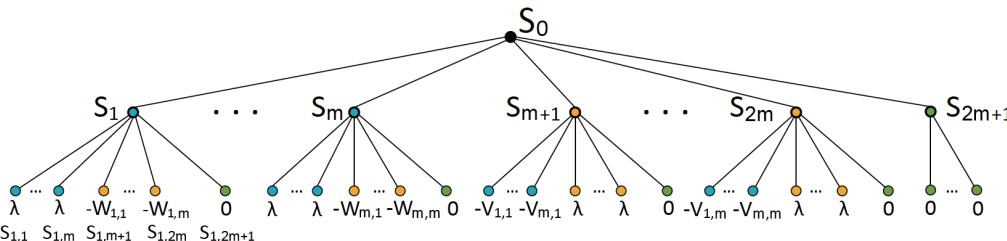

Figure 2: The MDP $\mathcal{M}$ constructed in our reduction.

Formally, layered MDP $\mathcal{M}$ has episode length $H = 3$ with initial state $\mathcal{S}_1 = \{S_0\}$, $A$ states at the second time step denoted as $\mathcal{S}_2 = \{S_1, S_2, \cdots S_A\}$, and $A^2$ states at the third time step denoted as $\mathcal{S}_3 = \{S_{1,1}, S_{1,2}, \cdots S_{1,A}, S_{2,1}, \cdots, S_{A,A}\}$. We define state space $\mathcal{S} = \mathcal{S}_1 \cup \mathcal{S}_2 \cup \mathcal{S}_3$. Define action space $\mathcal{A} = \{a_1, a_2, \cdots, a_A\}$, initial distribution $\rho(S_0) = 1$ and deterministic transition dynamics $P_1(S_i|S_0, a_i) = 1$, $P_2(S_{i,j}|S_i, a_j) = 1$ for all $i, j \in [A]$. Define cost function $c(s, a) := \bar{c}(s)$, where[5]

$$\bar{c}(s) := \begin{cases} 0, & s \in \mathcal{S}_1 \cup \mathcal{S}_2 \\ 0, & s = S_{i,j}, \text{ where } i = A \text{ or } j = A \\ -V_{j,i}, & s = S_{i,j}, \text{ where } i \in [m] + m, j \in [m] \\ -W_{i,j}, & s = S_{i,j}, \text{ where } i \in [m], j \in [m] + m \\ \lambda, & \text{otherwise} \end{cases}$$

where we denote $[m] + m := \{m + 1, \cdots, 2m\}$ and set $\lambda = 54$.

On the policy side, we define benchmark policy class $\mathcal{B}$ that contains $A - 1 = 2m$ deterministic policies $h_j$ ($j \in [A-1]$), such that $\forall s \in \mathcal{S}$, $h_j(s) = a_j$. Define deterministic expert policy $\pi^E$ as: $\pi^E(s) = a_A$, $\forall s \in \mathcal{S}$. Note that action $a_A$ is never a choice for policies in $\mathcal{B}$ but always chosen by the expert. Recall the mixed policy set $\Pi_{\mathcal{B}}$ is defined as

$$\Pi_{\mathcal{B}} := \left\{ \pi_u(\cdot|s) := \sum_{h \in \mathcal{B}} u[h] \cdot h(\cdot|s) : u \in \Delta(\mathcal{B}) \right\}.$$

For the expert feedback function $\zeta_E$, we use advantage function $A^E(s, a) = Q_{\pi^E}(s, a) - V_{\pi^E}(s)$. The values of $A^E(s, a)$ are calculated as follows:

- For $s$ in $\mathcal{S}_3$: since for every $a$, $Q_{\pi^E}(s, a) = c(s, a) = \bar{c}(s) = 0$, we have $A^E(s, a) = 0$.
- For $s$ in $\mathcal{S}_2$:
  - First, suppose $s = S_i$ for $i \in \{1, \ldots, A - 1\}$. Following $\pi^E$ directs the agent to $S_{i,A}$, which encounters zero subsequent cost. This implies that $V_{\pi^E}(S_i) = 0$. On the other hand, taking action $a_j$ transitions to $S_{i,j}$ which encounters cost $\bar{c}(S_{i,j})$ subsequently. This means that $A^E(S_i, a_j) = \bar{c}(S_{i,j})$. Recall that $\forall i \in [A]$ $\bar{c}(S_{i,A}) = 0$, by this we have that $A^E(S_i, a_A) = 0$ for all $S \in \mathcal{S}_2$.
  - Next, suppose s $= S_A$. Taking any action $a$ and following policy $\pi^E$ afterwards encounters zero subsequent cost, which implies that $Q_{\pi^E}(S_A, a) = 0$. This implies that $V_{\pi^E}(S_A) = 0$ and $A^E(S_A, a) = 0$.
- For $s$ in $\mathcal{S}_1$: at the initial state $S_0$, taking any action $a$ and following $\pi^E$ afterwards takes the agents to state $S_{i,A}$ for some $i$, which has zero cost. This implies that $Q_{\pi^E}(S_A, a) = 0$, $V_{\pi^E}(S_A) = 0$, and $A^E(S_A, a) = 0$.

In summary, we have

$$\zeta_E(s, a) := A^E(s, a) = \begin{cases} 0, & s \in \mathcal{S}_1 \cup \mathcal{S}_3 \text{ or } s = S_A \text{ or } a = a_A \\ -V_{j,i}, & s = S_i, a = a_j, \text{ where } i \in [m] + m, j \in [m] \\ -W_{i,j}, & s = S_i, a = a_j, \text{ where } i \in [m], j \in [m] + m \\ \lambda, & \text{otherwise} \end{cases}$$

In summary, given any POLY$^{12}$-BIMATRIX instance $(U, V)$, f returns a POLY$^6$-VI-MDP instance $(\mathcal{M}, \zeta_E, \mathcal{B})$.

**Map g.** The map g is defined as: given $u^* = (u_x^*, u_y^*) \in \Delta(\mathcal{B})$, return

$$\hat{x} = \frac{u_x^*}{\|u_x^*\|_1}, \quad \hat{y} = \frac{u_y^*}{\|u_y^*\|_1}.$$

**Polynomial-time computability of the reduction.** For map f, given any POLY$^{12}$-BIMATRIX instance $(V, W)$ with $V, W \in [0, 1]^{m \times m}$, map $f(V, W)$ returns $(\mathcal{M}, \zeta_E, \mathcal{B})$, where $\mathcal{M} =$

---

[5]Strictly speaking, the cost should be within $[0, 1]$; this can be achieved by shifting the cost function by 1 and dividing by $\lambda + 1$. Without affecting the correctness of the proof, we set the cost this way for the simplicity of presentation.

$(\mathcal{S}, \mathcal{A}, H, c, \rho, P)$ has $|\mathcal{S}| = (2m + 1)^2 + (2m + 1) + 1 = O(m^2)$ states, $|\mathcal{A}| = 2m + 1 = O(m)$ actions, $H = 3$, $|\mathcal{S}| \cdot |\mathcal{A}| = O(m^3)$ cost function values, $(2m + 2)(2m + 1) = O(m^2)$ values for the deterministic transition probability and one fixed initial distribution. Meanwhile, map $\mathsf{f}$ returns $2m$ deterministic benchmark policies and $\zeta_E$ function with $|\mathcal{S}| \cdot |\mathcal{A}| = O(m^3)$ values. Combining the above, we conclude that $\mathsf{f}$ runs in $O(m^3)$ time.

For map $\mathsf{g}$, by its definition, it can be computed in $O(m)$ time. In all, $\mathsf{f}$ and $\mathsf{g}$ are computable in polynomial-time with respect to $m$.

**Correctness of the reduction.**

1. If $(V, W)$ is a valid input of $\text{POLY}^{12}$-BIMATRIX, then $V, W \in [0, 1]^{m \times m}$. Given any $V, W \in [0, 1]^{m \times m}$, by the definition of $\mathsf{f}$, it is straightforward to see $\mathsf{f}$ constructs an discrete state-action episodic MDP with expert feedback $\zeta_E : \mathcal{S} \times \mathcal{A} \to \mathbb{R}$ and deterministic policy class $\mathcal{B}$. Thus, $\mathsf{f}(V, W)$ is a valid input of $\text{POLY}^6$-VI-MDP.

2. By Lemma 31 (given below), if $u^*$ is a solution of $\text{POLY}^6$-VI-MDP instance $\mathsf{f}(V, W)$, then $\mathsf{g}(u^*)$ is also a solution of $\text{POLY}^{12}$-BIMATRIX instance $(V, W)$.

3. By Lemma 29, any $\text{POLY}^6$-VI-MDP instance has a solution.

In conclusion, $\text{POLY}^{12}$-BIMATRIX $\leq_p \text{POLY}^6$-VI-MDP, thus $\text{POLY}^6$-VI-MDP is PPAD-hard. $\quad\square$

**Lemma 31.** $\mathsf{g}(u^*)$ *is a solution of* $\text{POLY}^{12}$-BIMATRIX *instance* $(V, W)$ *if* $u^*$ *is a solution of* $\text{POLY}^6$-VI-MDP *instance* $\mathsf{f}(V, W)$.

*Proof.* Recall that $u^*$ the solution of a $\text{POLY}^6$-VI-MDP instance $(\mathcal{M}, \zeta_E, \mathcal{B})$ satisfies $\forall u \in \Delta(\mathcal{B})$, $\langle \theta(u^*), u - u^* \rangle \geq -(S + A + B)^{-6}$, where $S = |\mathcal{S}|, A = |\mathcal{A}|, B = |\mathcal{B}|$. The proof follows by first calculating vector field $\theta(u)$ induced by $(\mathcal{M}, \zeta_E, \mathcal{B})$, and then showing that $\mathsf{g}(u^*)$ is a solution of the original $\text{POLY}^{12}$-BIMATRIX instance $(V, W)$ if $u^*$ is a solution of $\text{POLY}^6$-VI-MDP instance $\mathsf{f}(V, W)$.

**To begin with, given $V, W \in [0, 1]^{m \times m}$ we calculate $\theta$ in the $\text{POLY}^6$-VI-MDP instance $\mathsf{f}(V, W)$.** We have by definition in Section 3, $\forall u \in \Delta(\mathcal{B})$,

$$\theta(u) = \left( \mathbb{E}_{s \sim d_{\pi_u}} \mathbb{E}_{a \sim h_j(\cdot|s)} \left[ \zeta_E(s, a) \right] \right)_{h_j \in \mathcal{B}} = \frac{1}{3} \left( \mathbb{E}_{s \sim d_{\pi_u}^2} \left[ \zeta_E(s, h_j(s)) \right] \right)_{h_j \in \mathcal{B}},$$

where in the second equality we recall that $d_{\pi_u}^2$ denotes the state occupancy distribution at the second step and $d_{\pi_u} = \frac{1}{3}(d_{\pi_u}^1 + d_{\pi_u}^2 + d_{\pi_u}^3)$. The second equality is by the fact that $\zeta_E(s, a)$ is always $0$ when $s \in \mathcal{S}_1 \cup \mathcal{S}_3$.

Recall that $h_j(s) = a_j$ and $B = 2m$ as defined in the proof of Theorem 30, it can be verified that $\pi_u(\cdot|s) = \sum_{j \in [2m]} u[j] \cdot h_j(\cdot|s) = (u[1], u[2], \cdots, u[2m], 0)^\top$, where the last entry is $0$ since the last action is never chosen by any $h_j \in \mathcal{B}$. We can calculate its state occupancy distribution at step 2 as: $\Pr_{s \sim d_{\pi_u}^2}(s = S_i) = u[i]$ for $i \in [2m]$ and $\Pr_{s \sim d_{\pi_u}^2}(s = S_A) = 0$. Therefore, $\forall j \in [2m]$,

$$\mathbb{E}_{s \sim d_{\pi_u}^2} \left[ \zeta_E(s, a_j) \right] = \sum_{i=1}^{2m} u[i] \zeta_E(S_i, a_j).$$

For any $u \in \Delta(\mathcal{B})$, we use $u_x, u_y$ to denote the vector that consists of the first $m$ elements and the last $m$ elements of $u$ respectively. Given $V, W \in [0, 1]^{m \times m}$, define matrix

$$C := \begin{pmatrix} \lambda \mathbf{1}_{m \times m} & -V \\ -W^\top & \lambda \mathbf{1}_{m \times m} \end{pmatrix} \in \mathbb{R}^{2m \times 2m},$$

where $\mathbf{1}_{m \times m}$ denotes the matrix whose entries are all $1$'s. Notice that by the value of $\zeta_E$ calculated in the proof of Theorem 30, it can be verified that $\forall i \in [2m], \forall j \in [2m], \zeta_E(S_i, a_j) = C_{i,j}$. With this, $\theta(u)$ can be written in matrix form:

$$\theta(u) = \frac{1}{3} \left( \sum_{i=1}^{2m} u[i] \zeta_E(S_i, a_j) \right)_{j \in [2m]} = \frac{1}{3} \begin{pmatrix} \lambda \mathbf{1}_{m \times m} & -V \\ -W^\top & \lambda \mathbf{1}_{m \times m} \end{pmatrix} \cdot \begin{pmatrix} u_x \\ u_y \end{pmatrix} = \frac{1}{3} C u. \quad (11)$$

Therefore, the constructed discrete state-action episodic MDP $\mathcal{M}$, expert feedback $\zeta_E$ and benchmark policy class $\mathcal{B}$ induces the following instance of POLY$^6$-VI-MDP: find $u^* \in \Delta(\mathcal{B})$ such that $\forall u \in \Delta(\mathcal{B})$,

$$\langle \theta(u^*), u - u^* \rangle = \left\langle \frac{1}{3} C u^*, u - u^* \right\rangle \geq -(S + A + B)^{-6} = -(4m^2 + 10m + 4)^{-6}, \quad (12)$$

where we recall that $S = (2m + 1)^2 + (2m + 1) + 1$, $A = 2m + 1$, and $B = 2m$.

**Next, we show that $\mathsf{g}(u^*)$ is a solution of POLY$^{12}$-BIMATRIX instance $(V, W)$.** Recall that in Section F.1, given $V, W \in [0, 1]^{m \times m}$, the mixed strategies on set $\mathcal{A}_x$ and sets $\mathcal{A}_y$ and $|\mathcal{A}_x| = |\mathcal{A}_y| = m$ are represented by $x \in \Delta(\mathcal{A}_x)$ and $y \in \Delta(\mathcal{A}_y)$, respectively. If $(\hat{x}, \hat{y}) \in \Delta(\mathcal{A}_x) \times \Delta(\mathcal{A}_y)$ is a solution of POLY$^{12}$-BIMATRIX instance $(V, W)$, then $(\hat{x}, \hat{y})$ satisfies for any $x \in \Delta(\mathcal{A}_x)$ and $y \in \Delta(\mathcal{A}_y)$,

$$\begin{cases} \hat{x}^\top V \hat{y} \geq x^\top V \hat{y} - m^{-12}, \\ \hat{x}^\top W \hat{y} \geq \hat{x}^\top W y - m^{-12}. \end{cases}$$

Now, consider $u^* = (u_x^*, u_y^*) \in \Delta(\mathcal{B})$, a solution for POLY$^6$-VI-MDP instance $\mathsf{f}(V, W)$, such that $\forall u \in \Delta(\mathcal{B})$,

$$\langle \theta(u^*), u - u^* \rangle \geq -(4m^2 + 10m + 4)^{-6} = -\epsilon,$$

where we use $\epsilon$ to denote $(4m^2 + 10m + 4)^{-6}$.

We will show that $(\hat{x}, \hat{y}) = \mathsf{g}(u^*) = \left( \frac{u_x^*}{\|u_x^*\|_1}, \frac{u_y^*}{\|u_y^*\|_1} \right)$ is a solution of POLY$^{12}$-BIMATRIX instance $(V, W)$. To see this, we first prove that $\forall x \in \Delta(\mathcal{A}_x)$,

$$\hat{x}^\top V \hat{y} \geq x^\top V \hat{y} - m^{-12}. \quad (13)$$

To this end, $\forall x \in \Delta(\mathcal{A}_x)$, by setting $u = (\|u_x^*\|_1 \cdot x, u_y^*) \in \Delta(\mathcal{B})$ and plugging this choice of $u$ into Equation (12), we have

$$\begin{aligned} -\epsilon &\leq \left\langle \frac{1}{3} C u^*, u - u^* \right\rangle \\ &= \frac{\|u_x^*\|_1}{3} \left( (x - \hat{x})^\top, \mathbf{0}^\top \right) \begin{pmatrix} \lambda \mathbf{1}_{m \times m} & -V \\ -W^\top & \lambda \mathbf{1}_{m \times m} \end{pmatrix} \cdot \begin{pmatrix} u_x^* \\ u_y^* \end{pmatrix} \\ &= \frac{\|u_x^*\|_1}{3} \left( \lambda (x - \hat{x})^\top \mathbf{1}_{m \times m} u_x^* - (x - \hat{x})^\top V u_y^* \right) \\ &= \frac{\lambda \|u_x^*\|_1}{3} \mathbf{0}^\top u_x^* - \frac{\|u_x^*\|_1}{3} (x - \hat{x})^\top V u_y^* \\ &= -\frac{\|u_x^*\|_1 \|u_y^*\|_1}{3} (x - \hat{x})^\top V \hat{y}, \end{aligned}$$

where we use $\mathbf{0}$ to denote the all 0 vector in $\mathbb{R}^m$. Combining $\|u_x^*\|_1 \cdot \|u_y^*\|_1 \geq \frac{2}{9}$ as shown later by Lemma 32, we obtain

$$x^\top V \hat{y} - \hat{x}^\top V \hat{y} \leq \frac{3}{\|u_x^*\|_1 \|u_y^*\|_1} \epsilon \leq \frac{27}{2} \epsilon = \frac{27}{2(4m^2 + 10m + 4)^6} \leq m^{-12}.$$

This establishes Equation (13). Using a symmetrical argument, for any $y \in \Delta(\mathcal{A}_y)$, by taking $u = (u_x^*, \|u_y^*\|_1 \cdot y)$, we can also show that $\forall y \in \Delta(\mathcal{A}_y)$,

$$\hat{x}^\top W \hat{y} \geq \hat{x}^\top W y - m^{-12}. \quad (14)$$

Combining Equations (13) and (14), we conclude that $(\hat{x}, \hat{y})$ is a solution of POLY$^{12}$-BIMATRIX instance $(V, W)$. $\qquad \square$

**Lemma 32.** $\forall V, W \in [0, 1]^{m \times m}$, if $u^* = (u_x^*, u_y^*)$ is a solution of POLY$^6$-VI-MDP instance $\mathsf{f}(V, W)$, then

$$\|u_x^*\|_1 \cdot \|u_y^*\|_1 \geq \frac{2}{9}.$$

*Proof.* The lemma is proved by showing $\forall\, V, W \in [0,1]^{m\times m}$, $\forall u = (u_x, u_y) \in \Delta(\mathcal{B})$ such that $\|u_x\|_1 \notin (\frac{1}{3}, \frac{2}{3})$, for any $v = (v_x, v_y) \in \Delta(\mathcal{B})$ that satisfies $\|v_x\|_1 = \|v_y\|_1 = \frac{1}{2}$,

$$\langle \theta(u), v - u \rangle \leq -\frac{1}{3} < -(4m^2 + 10m + 4)^{-6},$$

where we recall the definition of $\theta(\cdot)$ from Equation (11). This implies $u^* = (u_x^*, u_y^*)$ the solution of POLY$^6$-VI-MDP instance $f(V, W)$ satisfies $\|u_x^*\|_1, \|u_x^*\|_1 \in [\frac{1}{3}, \frac{2}{3}]$, $\forall\, V, W \in [0,1]^{m\times m}$. Finally, it can be easily verified that $\forall\, a, b \in [\frac{1}{3}, \frac{2}{3}]$, $ab \geq \frac{2}{9}$.

By the alternative expression of $\theta(\cdot)$ shown in Equation (11), we can write $\langle \theta(u), u \rangle$ and $\langle \theta(u), v \rangle$ with $u = (u_x, u_y)$ and $v = (v_x, v_y)$ as:

$$\langle \theta(u), u \rangle = \frac{1}{3}(-u_x^\top V u_y - u_x^\top W u_y + \lambda(\|u_x\|_1^2 + \|u_y\|_1^2)),$$

$$\langle \theta(u), v \rangle = \frac{1}{3}(-v_x^\top V u_y - u_x^\top W v_y + \lambda(\|u_x\|_1\|v_x\|_1 + \|u_y\|_1\|v_y\|_1)).$$

By algebra, $\forall\, x \in \Delta(\mathcal{A}_x)$, $\forall\, y \in \Delta(\mathcal{A}_y)$ for $|\mathcal{A}_x| = |\mathcal{A}_y| = m$, $\forall\, V, W \in [0,1]^{m\times m}$, $0 \leq x^\top V y$, $x^\top W y \leq 1$, then

$$\langle \theta(u), v - u \rangle = \frac{1}{3}\left( u_x^\top V u_y + u_x^\top W u_y - v_x^\top V u_y - u_x^\top W v_y \right)$$

$$+ \frac{1}{3}\left( \lambda\|u_x\|_1(\|v_x\|_1 - \|u_x\|_1) + \lambda\|u_y\|_1(\|v_y\|_1 - \|u_y\|_1) \right)$$

$$\leq \frac{2}{3} + 18\left( \|u_x\|_1(\|v_x\|_1 - \|u_x\|_1) + \|u_y\|_1(\|v_y\|_1 - \|u_y\|_1) \right),$$

where we recall that $\lambda = 54$. Therefore, consider $\|u_x\| \notin (\frac{1}{3}, \frac{2}{3})$ and any $v$ such that $\|v_x\| = \|v_y\| = \frac{1}{2}$, we have:

$$\|u_x\|_1(\|v_x\|_1 - \|u_x\|_1) + \|u_y\|_1(\|v_y\|_1 - \|u_y\|_1)$$

$$= \|u_x\|_1(\frac{1}{2} - \|u_x\|_1) + \|u_y\|_1(\frac{1}{2} - \|u_y\|_1)$$

$$= -2(\|u_x\|_1 - \frac{1}{2})^2 < -\frac{1}{18}.$$

Plugging this back, we conclude that $\forall u = (u_x, u_y) \in \Delta(\mathcal{B})$ such that $\|u_x\| \notin (\frac{1}{3}, \frac{2}{3})$, for any $v = (v_x, v_y) \in \Delta(\mathcal{B})$ such that $\|v_x\| = \|v_y\| = \frac{1}{2}$,

$$\langle \theta(u), v - u \rangle \leq \frac{2}{3} - 1 = -\frac{1}{3}. \qquad \square$$

### F.4 Computational hardness of achieving sublinear dynamic regret in LOGGER

**Theorem 33** (Restatement of Theorem 12). *Fix $\gamma > 0$, if there exist a COIL algorithm such that for any $\mathcal{M}$ and expert $\pi^E$, it interacts with $\mathcal{M}$, CSC oracle $\mathcal{O}$, expert feedback $\zeta_E(s, a) = A^E(s, a)$, and outputs a sequence of $\{\pi_{u_n}\}_{n=1}^N \in \Pi_\mathcal{B}$ such that with probability at least $\frac{1}{2}$,*

$$\mathrm{DReg}_N(\mathcal{B}) \leq O(\mathrm{poly}(S, A, B) \cdot N^{1-\gamma}),$$

*in $\mathrm{poly}(N, S, A, B)$ time, then all problems in PPAD are solvable in randomized polynomial time.*

*Proof.* We start the proof by showing if there exists a COIL algorithm Alg1 that achieves $p(S, A, B)N^{1-\gamma}$ dynamic regret with probability $\frac{1}{2}$ in time $q(N, S, A, B)$, where $p$ and $q$ are polynomial functions, then, Alg1 yields an algorithm Alg1′ (Algorithm 5) that solves POLY$^6$-VI-MDP with expected polynomial time $\mathrm{poly}(S, A, B)$.

**Correctness.** As Alg1′ return only if $u_n$ is an $(S + A + B)^{-6}$-approximate solution of $\mathrm{VI}(\Delta(\mathcal{B}), \theta)$, it solves the POLY$^6$-VI-MDP problem.

---

**Algorithm 5** Alg1$'$

---
1: **while true do**
2:     Run Alg1 on $\mathcal{M}$, $\mathcal{B}$, $\zeta_E$, $\mathcal{O}$ for $N = (p(S, A, B) \cdot (S + A + B)^6)^{\frac{1}{\gamma}}$ rounds, obtaining a sequence of policies $\{\pi_n\}_{n=1}^N$ parameterized by $\{u_n\}_{n=1}^N$.
3:     If any of $u_n$ is a $(S + A + B)^{-6}$-approximate solution of $\mathrm{VI}(\Delta(\mathcal{B}), \theta)$, return $u_n$.
4: **end while**

---

**Time complexity.** We now bound the time complexity of Alg1$'$. Note that Alg1$'$ sets Alg1 with $N = (p(S, A, B) \cdot (S + A + B)^6)^{\frac{1}{\gamma}} = \mathrm{poly}(S + A + B)$, and Alg1 has a running time of $\mathrm{poly}(N, S, A, B) = \mathrm{poly}(S, A, B)$, together, each iteration of Alg1$'$ takes $\mathrm{poly}(S, A, B)$ time.

We now show that Alg1$'$ runs for an expected number of iterations at most a constant. Specifically, the guarantees of Alg1 implies that for each iteration, with probability at least $\frac{1}{2}$,

$$
\begin{aligned}
\mathrm{DReg}_N(\mathcal{B})(\{\pi_{u_n}\}_{n=1}^N) &= \sum_{n=1}^N F_n(\pi_{u_n}) - \min_{\pi \in \mathcal{B}} F_n(\pi) \\
&= \sum_{n=1}^N \langle \theta(u_n), u_n \rangle - \min_{u \in \Delta(\mathcal{B})} \langle \theta(u_n), u \rangle \\
&= \sum_{n=1}^N \max_{u \in \Delta(\mathcal{B})} \langle \theta(u_n), u_n - u \rangle \\
&\leq f(S, A, B) \cdot N^{1-\gamma},
\end{aligned}
$$

where the first equality is from the definition of dynamic regret in Section 2, and the second is by $F_n(\pi_u) = \langle \theta(u_n), u \rangle$. In this event, since $N = (p(S, A, B) \cdot (S + A + B)^6)^{\frac{1}{\gamma}}$, we have that $\exists n \in [N]$ s.t.

$$
\max_{u \in \Delta(\mathcal{B})} \langle \theta(u_n), u_n - u \rangle \leq p(S, A, B) \cdot N^{-\gamma} = (S + A + B)^{-6},
$$

which means $u_n$ is a solution of POLY$^6$-VI-MDP instance $(\mathcal{M}, \zeta_E, \mathcal{B})$. Hence, the expected number of iterations of running Alg1 before a valid solution being returned is smaller or equal to $\sum_{t=0}^\infty \left(\frac{1}{2}\right)^t = 2$. Thus, the expected running time of Alg1$'$ is $O(\mathrm{poly}(S + A + B))$.

By Theorem 30, all problems in PPAD are polynomial-time reducible to POLY$^6$-VI-MDP. If there exist a COIL algorithm Alg1 that achieves sublinear dynamic regret in LOGGER, then Alg1$'$ can be constructed to solve POLY$^6$-VI-MDP in randomized polynomial time, which means all problems in PPAD are solvable in randomized polynomial time. $\qquad\square$

## G   Online linear optimization results

In this section, we first provide a recap on online linear optimization, the well-known Optimistic Follow the Regularized Leader (FTRL) algorithm (Algorithm 6) [51], and its regret guarantees (Theorem 42). Section G.3 instantiates this general result with the regularizer $R_{\mathcal{N}}$ (Equation (5)) defined in Section E.1.

### G.1   Basic facts on convex analysis

Before we delve into the optimistic FTRL algorithm, we state some useful definitions and facts from convex analysis.

**Definition 34.** *For a differentiable convex function $f : \mathbb{R}^d \to \mathbb{R}$, define $D_f(v, w) = f(v) - f(w) - \langle v - w, \nabla f(w) \rangle$ to be the Bregman divergence induced by $f$.*

**Definition 35.** *Given a convex function $f : \mathbb{R}^d \to \mathbb{R} \cup \{+\infty\}$, where $\Omega \subseteq \mathbb{R}^d$, define $f^* : \mathbb{R}^d \to \mathbb{R} \cup \{+\infty\}$ as $f^*(\theta) := \sup_{w \in \mathbb{R}^d} (\langle \theta, w \rangle - f(w))$ to be its Fechel conjugate.*

**Definition 36.** *A convex function $f : \mathbb{R}^d \to \mathbb{R} \cup \{+\infty\}$ is proper if it is not identically equal to $+\infty$.*

**Definition 37.** *A function $f : \mathbb{R}^d \to \mathbb{R} \cup \{+\infty\}$ is $\alpha$-strongly convex w.r.t. a norm $\|\cdot\|$ if for all $v, w$ in the relative interior of the effective domain of $f$ and $\lambda \in (0, 1)$ we have*

$$f(\lambda v + (1-\lambda)w) \le \lambda f(v) + (1-\lambda)f(w) - \frac{1}{2}\alpha\lambda(1-\lambda)\|v - w\|^2.$$

**Definition 38.** *A function $f : \mathbb{R}^d \to \mathbb{R}$ is $\beta$-strongly smooth w.r.t. a norm $\|\cdot\|$ if $f$ is everywhere differentiable and if for all $v, w \in \mathbb{R}^d$ we have*

$$f(v + w) \le f(v) + \langle \nabla f(v), w \rangle + \frac{1}{2}\beta\|w\|^2.$$

**Definition 39.** *For a convex function $f : \mathbb{R}^d \to \mathbb{R} \cup \{+\infty\}$, define its effective domain $\mathrm{dom}(f) := \{w \in \mathbb{R}^d : f(w) < +\infty\}$.*

Note that the effective domain of a strongly smooth function $f$ satisfies $\mathrm{dom}(f) = \mathbb{R}^d$.

**Fact 40.** *Let $f : \mathbb{R}^d \to \mathbb{R} \cup \{+\infty\}$ be a proper, closed and convex function, then:*

1. *$f^*$ is closed and convex ([53, Theorem 12.2]);*
2. *$f^{**} = f$ ([53, Corollary 12.2.1]);*
3. *$\theta \in \partial f(w) \Leftrightarrow f(w) + f^*(\theta) = \langle \theta, w \rangle \Leftrightarrow w \in \partial f^*(\theta)$, where $\partial g(w)$ denotes $g$'s subdifferential set at $w$ ([73, Theorem 2.4.2]);*
4. *$f$ is $\alpha$-strongly convex with respect to a norm $\|\cdot\|$ if and only if $f^\star$ is $\frac{1}{\alpha}$-strongly smooth with respect to the dual norm $\|\cdot\|_\star$ ([35, Theorem 3]).*

**Proposition 41.** *Let $f : \mathbb{R}^d \to \mathbb{R} \cup \{+\infty\}$ be proper, closed and $\alpha$-strongly convex with respect to $\|\cdot\|$, then $f^*$ is differentiable, and $\nabla f^*(\theta) = \mathrm{argmax}_{w \in \mathrm{dom}(f)} \left( \langle \theta, w \rangle - f(w) \right).$*

*Proof sketch.* Given a function $f$ that is proper, closed and strongly convex, define $f_1 : \mathrm{dom}(f) \to \mathbb{R}$, where $f_1(w) := f(w)$ on $\mathrm{dom}(f)$. It can be seen that $f_1 : \Omega \to \mathbb{R}$ is closed and strongly convex. It can be checked that $f^*(\theta) = \sup_{w \in \mathbb{R}^d} \left( \langle \theta, w \rangle - f(w) \right) = \sup_{w \in \mathrm{dom}(f)} \left( \langle \theta, w \rangle - f(w) \right) = f_1^*(\theta)$ where $f_1^*$ is defined using the notations in [57]. The proposition follows from [57, Lemma 15]. $\square$

### G.2 General results on FTRL and Optimistic FTRL

Online linear optimization refers to the following $N$-round protocol: the learner is given a convex decision set $\Omega \subset \mathbb{R}^d$. At every round $n \in [N]$, the learner chooses some decision $u_n \in \Omega$, and then receives a linear loss $\langle g_n, \cdot \rangle$, where $g_n \in \mathbb{R}^d$. The goal of the learner is to minimize its regret on this sequence of linear losses:

$$\mathrm{LReg}_N := \sum_{n=1}^{N} \langle g_n, u_n \rangle - \min_{u \in \Omega} \sum_{n=1}^{N} \langle g_n, u \rangle.$$

Optimistic FTRL (Algorithm 6), works for online linear optimization with a general decision set $\Omega$. It takes into input a strongly convex regularizer $R$ with effective domain $\Omega$ and a learning rate $\eta > 0$. It maintains the cumulative linear loss $\Theta_n = \sum_{i=1}^{n} g_i$ over time (line 6); at round $n$, it first uses a predicted instantaneous loss $\hat{g}_n$ to construct a guess on the cumulative loss $\hat{\Theta}_n$ (line 3), then chooses the decision $u_n$ that minimizes the regularized guessed cumulative linear loss $\eta\langle\hat{\Theta}_n, u\rangle + R(u)$ (line 4), which by Proposition 41, has an equivalent form of $\nabla R^*(-\eta\hat{\Theta}_n)$. We have the following guarantee on the regret of Optimistic FTRL; it is largely inspired by and slightly generalizes the results of [59, 2, 51].

**Theorem 42.** *Let $R : \mathbb{R}^d \to \mathbb{R} \cup \{+\infty\}$ be a closed and $\alpha$-strongly convex function with respect to $\|\cdot\|$, such that $\Omega = \mathrm{dom}(R)$. The linear regret of Optimistic FTRL with $R$ and learning rate $\eta > 0$,*

---

**Algorithm 6** Optimistic FTRL

---

**Require:** Convex decision set $\Omega \subseteq \mathbb{R}^d$, regularizer $R$ that is closed and $\alpha$-strongly convex with bounded $\mathrm{dom}(R) = \Omega$, learning rate $\eta > 0$.

1: Initialize $\Theta_0 = 0$.
2: **for** $n = 1, 2, \ldots, N$ **do**
3:     The learner makes prediction $\hat{g}_n$, $\hat{\Theta}_n = \Theta_{n-1} + \hat{g}_n$ .
4:     The learner plays $u_n = \mathrm{argmin}_{u \in \Omega} \left( \eta \langle \hat{\Theta}_n, u \rangle + R(u) \right) = \nabla R^*(-\eta \hat{\Theta}_n)$ .
5:     The learner receives real loss $g_n$.
6:     Update $\Theta_n = \Theta_{n-1} + g_n$ .
7: **end for**

---

*satisfies the following:*

$$\mathrm{LReg}_N \leq \frac{\sup_{w \in \Omega} R(w) - \inf_{w \in \Omega} R(w)}{\eta} + \frac{1}{\eta} \sum_{n=1}^{N} \left( \underbrace{D_{R^*}\left(-\eta\Theta_n, -\eta\hat{\Theta}_n\right)}_{\text{divergence penalty}} - \underbrace{D_{R^*}\left(-\eta\Theta_{n-1}, -\eta\hat{\Theta}_n\right)}_{\text{prediction gain}} \right)$$

(15)

$$\leq \frac{\sup_{w \in \Omega} R(w) - \inf_{w \in \Omega} R(w)}{\eta} + \sum_{n=1}^{N} \frac{\eta \|\hat{g}_n - g_n\|_*^2}{2\alpha} - \frac{\alpha}{2\eta} \|\nabla R^*(-\eta\Theta_{n-1}) - \nabla R^*(-\eta\hat{\Theta}_n)\|^2.$$

(16)

*Specifically, if $\hat{g}_n = 0$ for all $n$,*

$$\mathrm{LReg}_N \leq \frac{\sup_{w \in \Omega} R(w) - \inf_{w \in \Omega} R(w)}{\eta} + \sum_{n=1}^{N} \frac{\eta \|g_n\|_*^2}{2\alpha}.$$

*Proof.* By the definition of Bregman divergence, we have

$$\begin{cases} D_{R^*}(-\eta\Theta_n, -\eta\hat{\Theta}_n) = R^*(-\eta\Theta_n) - R^*(-\eta\hat{\Theta}_n) - \left\langle -\eta\Theta_n + \eta\hat{\Theta}_n, \nabla R^*(-\eta\hat{\Theta}_n) \right\rangle, \\ D_{R^*}(-\eta\Theta_{n-1}, -\eta\hat{\Theta}_n) = R^*(-\eta\Theta_{n-1}) - R^*(-\eta\hat{\Theta}_n) - \left\langle -\eta\Theta_{n-1} + \eta\hat{\Theta}_n, \nabla R^*(-\eta\hat{\Theta}_n) \right\rangle. \end{cases}$$

By rearranging the terms, the two equations can be rewritten as

$$\begin{cases} \left\langle \eta\Theta_n - \eta\hat{\Theta}_n, \nabla R^*(-\eta\hat{\Theta}_n) \right\rangle = -R^*(-\eta\Theta_n) + R^*(-\eta\hat{\Theta}_n) + D_{R^*}(-\eta\Theta_n, -\eta\hat{\Theta}_n), \\ -\left\langle \eta\Theta_{n-1} - \eta\hat{\Theta}_n, \nabla R^*(-\eta\hat{\Theta}_n) \right\rangle = R^*(-\eta\Theta_{n-1}) - R^*(-\eta\hat{\Theta}_n) - D_{R^*}(-\eta\Theta_{n-1}, -\eta\hat{\Theta}_n). \end{cases}$$

By adding the two equations and recall that $\Theta_n = \Theta_{n-1} + g_n$ and $u_n = \nabla R^*(-\hat{\Theta}_n)$, we can write

$$\eta \langle g_n, u_n \rangle = \left\langle \eta\Theta_n - \eta\Theta_{n-1}, \nabla R^*(-\eta\hat{\Theta}_n) \right\rangle$$
$$= R^*(-\eta\Theta_{n-1}) - R^*(-\eta\Theta_n) + D_{R^*}(-\eta\Theta_n, -\eta\hat{\Theta}_n) - D_{R^*}(-\eta\Theta_{n-1}, -\eta\hat{\Theta}_n).$$

Summing over $n = 1, \ldots, N$, we have

$$\eta \sum_{n=1}^{N} \langle g_n, u_n \rangle = R^*(0) - R^*(-\eta\Theta_N) + \sum_{n=1}^{N} \left( D_{R^*}(-\eta\Theta_n, -\eta\hat{\Theta}_n) - D_{R^*}(-\eta\Theta_{n-1}, -\eta\hat{\Theta}_n) \right).$$

Therefore, we can bound $\eta\text{LReg}_N$ as:

$$\eta\text{LReg}_N = \sum_{n=1}^{N} \eta \langle g_n, u_n \rangle - \min_{u\in\Omega} \sum_{n=1}^{N} \langle \eta g_n, u \rangle$$

$$= -\min_{u\in\Omega} \langle \eta\Theta_N, u \rangle + R^*(0) - R^*(-\eta\Theta_N) + \sum_{n=1}^{N}\left( D_{R^*}(-\eta\Theta_n, -\eta\hat\Theta_n) - D_{R^*}(-\eta\Theta_{n-1}, -\eta\hat\Theta_n)\right)$$

$$= \max_{u\in\Omega} \langle -\eta\Theta_N, u \rangle - R^*(-\eta\Theta_N) + R^*(0) + \sum_{n=1}^{N}\left( D_{R^*}(-\eta\Theta_n, -\eta\hat\Theta_n) - D_{R^*}(-\eta\Theta_{n-1}, -\eta\hat\Theta_n)\right)$$

$$\leq \sup_{u\in\Omega} R(u) - \inf_{u\in\Omega} R(u) + \sum_{n=1}^{N}( D_{R^*}(-\eta\Theta_n, -\eta\hat\Theta_n) - D_{R^*}(-\eta\Theta_{n-1}, -\eta\hat\Theta_n)),$$

where the last inequality is by applying the definition of Fenchel conjugate and item 2 of Fact40:

1. $\max_{u\in\Omega} \langle -\eta\Theta_N, u \rangle - R^*(-\eta\Theta_N) \leq \sup_{u\in\Omega}\left(\sup_{\tilde\Theta\in\mathbb{R}^d}\left\langle -\tilde\Theta, u\right\rangle - R^*(\tilde\Theta)\right) = \sup_{u\in\Omega} R^{**}(u) = \sup_{u\in\Omega} R(u)$.
2. $R^*(0) = \sup_{u\in\mathbb{R}^d} \langle 0, u \rangle - R(u) = \sup_{u\in\Omega} \langle 0, u \rangle - R(u) = -\inf_{u\in\Omega} R(u)$.

This concludes the proof of Equation (15). We next prove Equation (16).

**Upper bounding the divergence penalty terms $D_{R^*}(-\eta\Theta_n, -\eta\hat\Theta_n)$.** Since $R$ is closed and $\alpha$-strongly convex, by item 4 of Fact 40, $R^*$ is $\frac{1}{\alpha}$-strongly smooth and $\forall \Theta, \Theta' \in \mathbb{R}^d$, $D_{R^*}(\Theta, \Theta') \leq \frac{1}{2\alpha}\|\Theta - \Theta'\|^2$, which implies $\forall n \in [N]$,

$$D_{R^*}(-\eta\Theta_n, -\eta\hat\Theta_n) \leq \frac{1}{2\alpha}\|\eta\Theta_n - \eta\hat\Theta_n\|_*^2 = \frac{\eta^2}{2\alpha}\|g_n - \hat g_n\|_*^2,$$

where the last equality is from the definition of $\Theta_n = g_n + \Theta_{n-1}$ and $\hat\Theta_n = \hat g_n + \Theta_{n-1}$.

**Lower bounding the prediction gain terms $D_{R^*}(-\eta\Theta_{n-1}, -\eta\hat\Theta_n)$.** Since $R$ is closed and $\alpha$-strongly convex, by Lemma 54, we have

$$D_{R^*}(-\eta\Theta_{n-1}, -\eta\hat\Theta_n)) \geq \frac{\alpha}{2}\|\nabla R^*(-\eta\Theta_{n-1}) - \nabla R^*(-\eta\hat\Theta_n)\|^2.$$

Finally, combining Equation (15) with the bounds on divergence penalty and prediction gain terms, Equation (16) is proved. □

**Remark 43.** *An alternative proof of this theorem can be done using the Stronger Follow the Leader Lemma [18, 41].*

### G.3 Regularizer induced by example-based perturbations

In this section, we instantiate the Optimistic FTRL algorithm and regret guarantee in the previous section with a specific $R$ that appears in our MFTPL, MFTPL-EG algorithms. Recall that in these algorithms, we use samples in the separator set $X$ and assign cost vector $\ell_x \sim \mathcal{N}(0, I_A)$ independently on each of them. As a result, we define $R$ in Equation (5) as the Fenchel conjugate of $\Phi_{\mathcal{N}}(\Theta) = \mathbb{E}_{\ell\sim\mathcal{N}(0,I_{XA})}\left[\max_{u\in\Delta(\mathcal{B})} \langle \Theta + q(\ell), u \rangle\right]$; recall that $q(\ell) = (\sum_{x\in\mathcal{X}} \ell_x(h(x))_{h\in\mathcal{B}}$ and $\ell = (\ell_x)_{x\in\mathcal{X}}$.

First, we prove several useful properties of $\Phi_{\mathcal{N}}$.

**Lemma 44** (Restatement of Lemma 16). *$\Phi_{\mathcal{N}}(\Theta)$ is differentiable for any $\Theta \in \mathbb{R}^B$ and $\nabla\Phi_{\mathcal{N}}(\Theta) = \mathbb{E}_{\ell\sim\mathcal{N}(0,I_{XA})}\left[\underset{u\in\Delta(\mathcal{B})}{\text{argmax}} \langle \Theta + q(\ell), u \rangle\right]$.*

*Proof.* To prove the lemma, by [9, Propositions 2.2, 2.3], it suffices to prove that for any $\Theta$, $\text{argmax}_{u\in\Delta(\mathcal{B})} \langle \Theta + q, u \rangle$ is unique with probability 1, over the draw of $\ell \sim \mathcal{N}(0, I_{XA})$.

By the definition of $\operatorname{argmax}$, it can be seen that the solution of $\operatorname*{argmax}_{u \in \Delta(\mathcal{B})} \langle \Theta + q, u \rangle$ is not unique if and only if there exist $h, h' \in \mathcal{B}$ and $h \neq h'$, such that $\Theta[h] + q[h] = \Theta[h'] + q[h'] = \max_h(\Theta[h] + q[h])$.

Define event $E = \left\{ \operatorname{argmax}_{u \in \Delta(\mathcal{B})} \langle \Theta + q, u \rangle \text{ is not unique} \right\}$ and event $E_{hh'}$: $\Theta[h] + q[h] = \Theta[h'] + q[h']$, for all $h, h' \in \mathcal{B}$ and $h \neq h'$. For $E$ to happen, it is necessary that one of $E_{h,h'}$ happens. Formally, $E \subseteq \cup_{h \neq h'} E_{hh'}$. By applying the union bound, we obtain

$$\Pr(E) \leq \Pr(\cup_{h \neq h'} E_{hh'}) \leq \sum_{h \neq h'} \Pr(E_{hh'}).$$

We will now show that for any $h, h' \in \mathcal{B}$ and $h \neq h'$, $\Pr(E_{hh'}) = 0$. By its definition , $E_{hh'}$ happens if and only if $\Theta[h] + q[h] = \Theta[h'] + q[h']$. We can rearrange the terms and get

$$q[h] - q[h'] = \eta \cdot \left( \Theta[h] - \Theta[h'] \right).$$

The following proof shows given any constant $C \in \mathbb{R}$, $q[h] - q[h'] = C$ happens with probability 0. Here we first recall the definition of separator set $\mathcal{X}$: $\exists x \in \mathcal{X}$ s.t. $h(x) \neq h'(x)$ for any $h \neq h'$ and define $\mathcal{X}_{h,h'} := \left\{ x \in \mathcal{X} | h(x) \neq h'(x) \right\}$. It can be seen that $\mathcal{X}_{h,h'}$ is nonempty for $h \neq h'$ by the definition of $\mathcal{X}$. Now we can rewrite $q[h] - q[h']$ with the help of $\mathcal{X}_{h,h'}$,

$$q[h] - q[h'] = \sum_{x \in \mathcal{X}} \ell_x(h(x)) - \ell_x(h'(x)) = \sum_{x \in \mathcal{X}_{h,h'}} \ell_x(h(x)) - \ell_x(h'(x)).$$

Since $\ell = (\ell_x)_{x \in \mathcal{X}} \sim \mathcal{N}(0, I_{XA})$, $q[h] - q[h']$ can be viewed as a sum of $2|\mathcal{X}_{h,h'}|$ independent Gaussian variables following distribution $\mathcal{N}(0, 1)$. By this observation, we have that $q[h] - q[h'] \sim \mathcal{N}(0, 4|\mathcal{X}_{h,h'}|^2)$, which implies $\forall C \in \mathbb{R}$, $\Pr(q[h] - q[h'] = C) = 0$. This in turn shows that $\Pr(E) \leq \sum_{h \neq h'} \Pr(E_{hh'}) = 0$, which concludes the proof of the lemma. $\qquad\square$

**Lemma 45.** $\Phi_{\mathcal{N}}$ is closed and convex on $\mathbb{R}^B$.

*Proof.* To begin with, we show $\Phi_{\mathcal{N}}$ is convex. Recall that $\Phi_{\mathcal{N}}(\Theta) = \mathbb{E}_{\ell \sim \mathcal{N}(0, I_{XA})} \left[ \max_{u \in \Delta(\mathcal{B})} \langle \Theta + q(\ell), u \rangle \right]$ where $q(\ell) = (\sum_{x \in \mathcal{X}} \ell_x(h(x)))_{h \in B}$. To check the convexity of $\Phi_{\mathcal{N}}$, given any $\Theta, \Theta' \in \mathbb{R}^B$ and any $\gamma \in [0, 1]$, we have

$$\begin{aligned}
\Phi_{\mathcal{N}}(\gamma \Theta + (1 - \gamma)\Theta') &= \mathbb{E}_{\ell \sim \mathcal{N}(0, I_{XA})} \max_{u \in \Delta(\mathcal{B})} \langle \gamma \Theta + (1 - \gamma)\Theta' + q(\ell), u \rangle \\
&= \mathbb{E}_{\ell \sim \mathcal{N}(0, I_{XA})} \max_{u \in \Delta(\mathcal{B})} \langle \gamma(\Theta + q(\ell)) + (1 - \gamma)(\Theta' + q(\ell)), u \rangle \\
&\leq \mathbb{E}_{\ell \sim \mathcal{N}(0, I_{XA})} ( \max_{u \in \Delta(\mathcal{B})} \langle \gamma(\Theta + q(\ell)), u \rangle + \max_{u \in \Delta(\mathcal{B})} \langle (1 - \gamma)(\Theta' + q(\ell)), u \rangle \\
&= \gamma \mathbb{E}_{\ell \sim \mathcal{N}(0, I_{XA})} \max_{u \in \Delta(\mathcal{B})} \langle \Theta + q(\ell), u \rangle + (1 - \gamma)\mathbb{E}_{\ell \sim \mathcal{N}(0, I_{XA})} \max_{u \in \Delta(\mathcal{B})} \langle \Theta' + q(\ell).u \rangle \\
&= \gamma \Phi_{\mathcal{N}}(\Theta) + (1 - \gamma)\Phi_{\mathcal{N}}(\Theta').
\end{aligned}$$

The inequality is by the fact that $\forall u \in \Delta(\mathcal{B}), \forall a, b \in \mathbb{R}^B$ , $\langle a + b, u \rangle \leq \max_{u \in \Delta(\mathcal{B})} \langle a, u \rangle + \max_{u \in \Delta(\mathcal{B})} \langle b, u \rangle$.

Secondly, to show $\Phi_{\mathcal{N}}$ is closed, by [12, Section A.3.3], since $\mathbb{R}^B$ is closed, it suffice to show $\Phi_{\mathcal{N}}$ continuous on $\mathbb{R}^B$. Since $\Phi_{\mathcal{N}}$ is differentiable on $\mathbb{R}^B$ by Lemma 16, we have that $\Phi_{\mathcal{N}}$ is continuous, which concludes that $\Phi_{\mathcal{N}}$ is closed. $\qquad\square$

The following two properties of $\Phi_{\mathcal{N}}$ are largely inspired by [66].

**Lemma 46.** $\Phi_{\mathcal{N}}(0) \leq \sqrt{2X \ln(B)}$.

*Proof.* By the definition of $\Phi_{\mathcal{N}}$, we have $\Phi_{\mathcal{N}}(0) = \mathbb{E}_{\ell \sim \mathcal{N}(0, I_{XA})} \left[ \max_{u \in \Delta(\mathcal{B})} \langle q(\ell), u \rangle \right]$, where $q(\ell) = (\sum_{x \in \mathcal{X}} \ell_x(h(x)))_{h \in B}$. For the remainder of the proof, we use $\mathbb{E}$ as an abbreviation for $\mathbb{E}_{\ell \sim \mathcal{N}(0, I_{XA})}$. Recall that $q[h] = \sum_{x \in \mathcal{X}} \ell_x(h(x))$, we can write $\Phi_{\mathcal{N}}(0) = \mathbb{E} \left[ \max_{h \in \mathcal{B}} q[h] \right]$. For any $b > 0$,

$$\exp\left( b \cdot \mathbb{E} \left[ \max_{h \in \mathcal{B}} q[h] \right] \right) \leq \mathbb{E} \left[ \exp(b \cdot \max_{h \in \mathcal{B}} q[h]) \right] = \mathbb{E} \left[ \max_{h \in \mathcal{B}} \exp\left( b \cdot q[h] \right) \right] \leq \sum_{h \in \mathcal{B}} \mathbb{E} \left[ \exp\left( b \cdot q[h] \right) \right],$$

where the first inequality is from the convexity of exponential function, while the last inequality is form $\max_{h \in \mathcal{B}} \exp(bq[h]) \leq \sum_{h \in \mathcal{B}} \exp(bq[h])$.

By the property of the sum of independent Gaussian variables, since $\ell \sim \mathcal{N}(0, I_{XA})$, we have that for any $h \in \mathcal{B}$, $q[h] = (\sum_{x \in \mathcal{X}} \ell_x(h(x)))_{h \in B}$ follows Gaussian distribution $\mathcal{N}(0, X)$. Then, $\forall h \in \mathcal{B}$, by a standard fact on the moment generating function of Gaussian random variables, we have that $\forall h \in \mathcal{B}$,

$$\mathbb{E} \exp(b \cdot q[h]) = \exp\left( \frac{b^2 X}{2} \right),$$

which implies

$$\exp\left( b \cdot \mathbb{E} \max_{h \in \mathcal{B}} q[h] \right) \leq B \exp\left( \frac{b^2 X}{2} \right).$$

Hence, by taking the natural logarithm and dividing by $b > 0$ on both sides, we get

$$\mathbb{E} \left[ \max_{h \in \mathcal{B}} q[h] \right] \leq \frac{\ln(B)}{b} + \frac{bX}{2}. \tag{17}$$

Since Equation (17) holds for any $b > 0$, by choosing $b = \sqrt{\frac{2 \ln(B)}{X}}$, we obtain $\mathbb{E} \left[ \max_{h \in \mathcal{B}} q[h] \right] \leq \sqrt{2X \ln(B)}$, which concludes $\Phi_{\mathcal{N}}(0) \leq \sqrt{2X \ln(B)}$. $\qquad \square$

**Lemma 47.** $\Phi_{\mathcal{N}}$ *is $\beta$-strongly smooth with respect to* $\| \cdot \|_{\infty}$ *with* $\beta = \sqrt{\frac{8}{\pi}} AX$.

*Proof.* To prove that $\Phi_{\mathcal{N}}$ is $\beta$-strongly smooth with respect to $\| \cdot \|_{\infty}$, by Definition 4.18 in [43], it suffices to show that $\forall \Theta, \Theta' \in \mathbb{R}^B$,

$$\| \nabla \Phi_{\mathcal{N}}(\Theta) - \nabla \Phi_{\mathcal{N}}(\Theta') \|_1 \leq \beta \| \Theta - \Theta' \|_{\infty}.$$

By Lemma 16, $\nabla \Phi_{\mathcal{N}}(\Theta) = \mathbb{E}_{\ell \sim \mathcal{N}(0, I_{XA})} \left[ \operatorname{argmax}_{u \in \Delta(\mathcal{B})} \langle \Theta + q(\ell), u \rangle \right]$, where $q(\ell) = (\sum_{x \in \mathcal{X}} \ell_x(h(x)))_{h \in B}$. Given $\ell$, we introduce shorthands

$$u_q = \operatorname*{argmax}_{u \in \Delta(\mathcal{B})} \langle u, \Theta + q(\ell) \rangle, u_q' = \operatorname*{argmax}_{u \in \Delta(\mathcal{B})} \langle u, \Theta' + q(\ell) \rangle.$$

We also introduce the short hand $h_q$ and $h_q'$ to represent the policy in class $\mathcal{B}$ selected by $u_q$ and $u_q'$. More explicitly,

$$h_q = \operatorname*{argmax}_{h \in B} \Theta[h] + q[h], \quad h_q' = \operatorname*{argmax}_{h \in B} \Theta'[h] + q[h].$$

By Lemma 16, $u_q, u_q', h_q, h_q'$ are well-defined with probability 1 over the randomness of $\ell$. With this notation, we have $\| u_q - u_q' \|_1 = 0$ when $h_q = h_q'$, and $\| u_q - u_q' \|_1 = 2$ when $h_q \neq h_q'$, which means $\| u_q - u_q' \|_1 = 2I(h_q \neq h_q')$. From now on, we use $P(\ell)$ to denote the probability density

function of $\ell$. By this, we have

$$
\begin{aligned}
\|\nabla\Phi_{\mathcal{N}}(\Theta) - \nabla\Phi_{\mathcal{N}}(\Theta')\|_1 &= \|\mathbb{E}_{\ell\sim\mathcal{N}(0, I_{XA})} u_q - \mathbb{E}_{\ell\sim\mathcal{N}(0, I_{XA})} u'_q\|_1 \\
&= \sum_{h\in\mathcal{B}} \left| \int_\ell (I(h = h_q) - I(h = h'_q)) P(\ell)\, d\ell \right| \\
&\leq \sum_{h\in\mathcal{B}} \int_\ell \left| (I(h = h_q) - I(h = h'_q)) \right| P(\ell)\, d\ell \\
&= \int_\ell \sum_{h\in\mathcal{B}} \left| (I(h = h_q) - I(h = h'_q)) \right| P(\ell)\, d\ell \\
&= \int_\ell 2 I(h_q \neq h'_q) P(\ell)\, d\ell \\
&= 2\Pr(h_q \neq h'_q),
\end{aligned}
$$

where $\Pr(h_q \neq h'_q)$ denotes the probability of $h_q \neq h'_q$ under the distribution of $\ell$. By the definition of separator set $\mathcal{X}$, $h \neq h'$ if and only if $\exists x \in \mathcal{X}$ s.t. $h(x) \neq h'(x)$. Then, by bringing in $h_q, h'_q$ and apply the union bound, we have

$$
\Pr(h_q \neq h'_q) \leq \sum_{x\in\mathcal{X}} \Pr(h_q(x) \neq h'_q(x)) = \sum_{x\in\mathcal{X}} \sum_{a\in\mathcal{A}} \Pr(a = h_q(x) \neq h'_q(x)).
$$

Then, given any $x$ and $a$, we denote $\ell_{-xa}$ as all other Gaussian variables in set $\{\ell_x(a)\}_{x\in X, a\in\mathcal{A}}$ except $\ell_x(a)$ and $\Pr(a = h_q(x) \neq h'_q(x))$ as the probability of $a = h_q(x) \neq h'_q(x)$ under the distribution of $\ell_{-xa}$. Then, $\forall x \in X, a \in \mathcal{A}$,

$$
\Pr(a = h_q(x) \neq h'_q(x)) = \int_{\ell_{-xa}} \Pr(a = h_q(x) \neq h'_q(x) | \ell_{-xa}) P(\ell_{-xa})\, d(\ell_{-xa}).
$$

Conditioned on $\ell_{-xa}$, we denote $\tilde{\ell} = \{\tilde{\ell}_x(a)\}_{x\in X, a\in\mathcal{A}}$ as the corresponding perturbation vector that share the same value with $\ell$ on all other entries and set $\tilde{\ell}_x(a) = 0$. Define

$$
\Theta_{xa} = \Theta + \left( \sum_{x\in\mathcal{X}} \tilde{\ell}_x(h(x)) \right)_{h\in\mathcal{B}}, \quad \Theta'_{xa} = \Theta' + \left( \sum_{x\in\mathcal{X}} \tilde{\ell}_x(h(x)) \right)_{h\in\mathcal{B}}.
$$

By algebra, $\Theta_{xa} - \Theta'_{xa} = \Theta - \Theta'$. By the definition of $u_q$ and $u'_q$, with the new notation, we can rewrite

$$
u_q = \operatorname*{argmax}_{u\in\Delta(\mathcal{B})} \langle u, \Theta_{xa} + (I(h(x) = a)\ell_x(a))_{h\in\mathcal{B}} \rangle, \quad u'(q) = \operatorname*{argmax}_{u\in\Delta(\mathcal{B})} \langle u, \Theta'_{xa} + (I(h(x) = a)\ell_x(a))_{h\in\mathcal{B}} \rangle.
$$

By using $(\Theta[h])_{h\in\mathcal{B}} := \Theta$, we can write

$$
h_q = \operatorname*{argmax}_{h\in\mathcal{B}} \Theta_{xa}[h] + I(h(x) = a)\ell_x(a), \quad h'_q = \operatorname*{argmax}_{h\in\mathcal{B}} \Theta'_{xa}[h] + I(h(x) = a)\ell_x(a).
$$

Then, by dividing the set $\mathcal{B}$ into disjoint subsets $\mathcal{B}_{xa} = \{h | h(x) = a, h \in \mathcal{B}\}$ and $\mathcal{B} \setminus \mathcal{B}_{xa}$. If $\mathcal{B}_{xa} = \varnothing$ or $\mathcal{B} \setminus \mathcal{B}_{xa} = \varnothing$, we have that $\Pr(a = h_q(x) \neq h'_q(x)) = 0$. Otherwise, we can view $h_q$ as the $h$ that corresponds to $\max\{ \max_{h\in\mathcal{B}_{xa}} \Theta_{xa}[h] + \ell_x(a), \max_{h\in\mathcal{B}\setminus\mathcal{B}_{xa}} \Theta_{xa}[h] \}$, and $h'_q$ as the $h$ that corresponds to $\max\{ \max_{h\in\mathcal{B}_{xa}} \Theta'_{xa}[h] + \ell_x(a), \max_{h\in\mathcal{B}\setminus\mathcal{B}_{xa}} \Theta'_{xa}[h] \}$. With this insight, it can be seen that

$$
\begin{cases}
\delta := \max_{h\in\mathcal{B}\setminus\mathcal{B}_{xa}} \Theta_{xa}[h] - \max_{h\in\mathcal{B}_{xa}} \Theta_{xa}[h] > \ell_x(a) \to a \neq h_q(x), \\
\delta' := \max_{h\in\mathcal{B}\setminus\mathcal{B}_{xa}} \Theta'_{xa}[h] - \max_{h\in\mathcal{B}_{xa}} \Theta'_{xa}[h] < \ell_x(a) \to a = h'_q(x).
\end{cases}
$$

Therefore, both $\delta \leq \ell_x(a)$ and $\ell_x(a) \leq \delta'$ are necessary for $a = h_q(x) \neq h'_q(x)$ to happen, which implies

$$
\Pr(a = h_q(x) \neq h'_q(x) | \ell_{-xa}) \leq \Pr(\delta \leq \ell_x(a) \leq \delta').
$$

If $\delta' < \delta$, then $\Pr(a = h_q(x) \neq h'_q(x)|\ell_{-xa}) = 0$. Otherwise ($\delta \leq \delta'$), conditioned on $\ell_{-xa}$, by $\ell_x(a) \sim \mathcal{N}(0,1)$, we have

$$\Pr(a = h_q(x) \neq h'_q(x)|\ell_{-xa}, \delta' \geq \delta) \leq \int_\delta^{\delta'} \frac{1}{\sqrt{2\pi}} \exp(-\frac{\ell_x(a)^2}{2}) \, \mathrm{d}(\ell_x(a)) \leq \frac{1}{\sqrt{2\pi}}(\delta' - \delta),$$

in which case,

$$
\begin{aligned}
\delta' - \delta &\leq \left| \max_{h \in \mathcal{B} \setminus \mathcal{B}_{xa}} \Theta_{xa}[h] - \max_{h \in \mathcal{B} \setminus \mathcal{B}_{xa}} \Theta'_{xa}[h] \right| + \left| \max_{h \in \mathcal{B}_{xa}} \Theta_{xa}[h] - \max_{h \in \mathcal{B}_{xa}} \Theta'_{xa}[h] \right| \\
&\leq \left| \max_{h \in \mathcal{B}} \Theta_{xa}[h] - \max_{h \in \mathcal{B}} \Theta'_{xa}[h] \right| + \left| \max_{h \in \mathcal{B}} \Theta_{xa}[h] - \max_{h \in \mathcal{B}} \Theta'_{xa}[h] \right| \\
&\leq 2 \left| \max_{h \in \mathcal{B}} (\Theta_{xa}[h] - \Theta'_{xa}[h]) \right| \\
&\leq 2 \max_{h \in \mathcal{B}} \left| \Theta_{xa}[h] - \Theta'_{xa}[h] \right| \\
&= 2 \| \Theta_{xa} - \Theta'_{xa} \|_\infty = 2 \| \Theta - \Theta' \|_\infty.
\end{aligned}
$$

By this we conclude $\forall x \in \mathcal{X}, \forall a \in \mathcal{A}, \Pr(a = h_q(x) \neq h'_q(x)|\ell_{-xa}) \leq \sqrt{\frac{2}{\pi}} \| \Theta - \Theta' \|_\infty$, and

$$\| \nabla \Phi_{\mathcal{N}}(\Theta) - \nabla \Phi_{\mathcal{N}}(\Theta') \|_1 \leq 2 \sum_{x \in \mathcal{X}} \sum_{a \in \mathcal{A}} \Pr(a = h_q(x) \neq h'_q(x)) \leq \sqrt{\frac{8}{\pi}} AX \| \Theta - \Theta' \|_\infty. \quad \square$$

Secondly, we prove useful properties of $R_{\mathcal{N}} : \mathbb{R}^B \to \mathbb{R} \cup \{+\infty\}$, where $R_{\mathcal{N}} = \Phi_{\mathcal{N}}^*$.

**Lemma 48** (Restatement of Lemma 17). $R_{\mathcal{N}}(u) = \Phi_{\mathcal{N}}^*(u)$ is closed and $\sqrt{\frac{\pi}{8}} \frac{1}{AX}$-strongly convex with respect to $\| \cdot \|_1$.

*Proof.* Since $\Phi_{\mathcal{N}}$ is closed and convex by Lemma 45, by item 1 of Fact 40, $R_{\mathcal{N}}(u) = \Phi_{\mathcal{N}}^*(u)$ is closed and convex. Also, by Lemma 47, $\Phi_{\mathcal{N}}$ is $\sqrt{\frac{8}{\pi}} AX$-strongly smooth with respect to $\| \cdot \|_\infty$. Then, we apply item 4 of Fact 40 and conclude $R_{\mathcal{N}}(u)$ is $\sqrt{\frac{\pi}{8}} \frac{1}{AX}$-strongly convex with respect to $\| \cdot \|_1$. $\square$

**Lemma 49.** $\forall u \in \Delta(\mathcal{B}), R_{\mathcal{N}}(u) \leq 0$.

*Proof.* Recall that $R_{\mathcal{N}}(u) = \sup_{\Theta \in \mathbb{R}^B} \langle \Theta, u \rangle - \Phi_{\mathcal{N}}(\Theta)$, where $\Phi_{\mathcal{N}}(\Theta) = \mathbb{E}_{\ell \sim \mathcal{N}(0, I_{XA})} \left[ \max_{v \in \Delta(\mathcal{B})} \langle \Theta + q(\ell), v \rangle \right]$, it suffices to show $\forall u \in \Delta(\mathcal{B}), \forall \Theta \in \mathbb{R}^B$, $\langle \Theta, u \rangle - \Phi_{\mathcal{N}}(\Theta) \leq 0$. For the remainder of the proof, we use $\mathbb{E}$ as an abbreviation for $\mathbb{E}_{\ell \sim \mathcal{N}(0, I_{XA})}$. Then, $\forall u \in \Delta(\mathcal{B}), \forall \Theta \in \mathbb{R}^B$,

$$\Phi_{\mathcal{N}}(\Theta) = \mathbb{E} \left[ \max_{v \in \Delta(\mathcal{B})} \langle \Theta + q(\ell), v \rangle \right] \geq \max_{v \in \Delta(\mathcal{B})} \mathbb{E} \left[ \langle \Theta + q(\ell), v \rangle \right] = \max_{v \in \Delta(\mathcal{B})} \langle \Theta, v \rangle \geq \langle \Theta, u \rangle,$$

where the first inequality is from Jensen's inequality, while the second equality is by the fact that

$$\mathbb{E} \left[ q(\ell) \right] = \mathbb{E} \left[ (\sum_{x \in \mathcal{X}} \ell_x(h(x)))_{h \in \mathcal{B}} \right] = \sum_{x \in \mathcal{X}} \mathbb{E} \left[ \ell_x(h(x))_{h \in \mathcal{B}} \right] = 0. \quad \square$$

**Lemma 50.** $\mathrm{dom}(R_{\mathcal{N}}) = \Delta(\mathcal{B})$.

*Proof.* We show the lemma in two steps. First, by Lemma 49, $\forall u \in \Delta(\mathcal{B}), R_{\mathcal{N}}(u) \leq 0$, which is finite. Secondly we show $\forall u \in \mathbb{R}^B \setminus \Delta(\mathcal{B}), R_{\mathcal{N}}(u) = +\infty$.

For the second step, $\forall u \in \mathbb{R}^B \setminus \Delta(\mathcal{B})$, by using $\mathbb{E}$ as an abbreviation for $\mathbb{E}_{\ell \sim \mathcal{N}(0, I_{X A})}$, we have

$$R_{\mathcal{N}}(u) = \sup_{\Theta \in \mathbb{R}^B} \langle \Theta, u \rangle - \Phi_{\mathcal{N}}(\Theta)$$

$$= \sup_{\Theta \in \mathbb{R}^B} \langle \Theta, u \rangle - \mathbb{E}\left[ \max_{v \in \Delta(\mathcal{B})} \langle \Theta + q(\ell), v \rangle \right]$$

$$\geq \sup_{\Theta \in \mathbb{R}^B} \left( \langle \Theta, u \rangle - \max_{v \in \Delta(\mathcal{B})} \langle \Theta, v \rangle \right) - \mathbb{E}\left[ \max_{v' \in \Delta(\mathcal{B})} \langle q(\ell), v' \rangle \right],$$

where the first inequality is by the convexity of $\max$ and inner product functions.

By Lemma 46, $\Phi_{\mathcal{N}}(0) = \mathbb{E}\left[ \max_{v' \in \Delta(\mathcal{B})} \langle q(\ell), v' \rangle \right] \leq \sqrt{2X \ln(B)}$, which is a constant. Now, it

suffices to show that $\forall u \in \mathbb{R}^B \setminus \Delta(\mathcal{B})$, $\sup_{\Theta \in \mathbb{R}^B} \left( \langle \Theta, u \rangle - \max_{v \in \Delta(\mathcal{B})} \langle \Theta, v \rangle \right) = +\infty$.

We divide $\mathbb{R}^B \setminus \Delta(\mathcal{B})$ into two disjoint sets:

$$\begin{cases} U^- := \{ u \in \mathbb{R}^B \mid u[i] < 0 \text{ for some } i \in \{1, 2, \cdots, B\} \}, \\ U^+ := \{ u \in \mathbb{R}^B \mid \|u\|_1 > 1, u \succeq 0 \}, \end{cases}$$

where it can be verified that $\mathbb{R} \setminus \Delta(\mathcal{B}) = U^- \cup U^+$.

For any $u \in U^-$, where $u[i] < 0$ for some $i \in \{1, 2, \cdots, B\}$, we have that $\forall C \in \mathbb{R}$, by setting $\theta(u, i, C) = \frac{|C|+1}{u[i]} \operatorname{Onehot}(i, \mathcal{B}) \in \mathbb{R}^B$,

$$\max_{v \in \Delta(\mathcal{B})} \langle \theta(u, i, C), u - v \rangle = \left\langle \frac{|C|+1}{u[i]} \operatorname{Onehot}(i, \mathcal{B}), u \right\rangle - \max_{v \in \Delta(\mathcal{B})} \left\langle \frac{|C|+1}{u[i]} \operatorname{Onehot}(i, \mathcal{B}), v \right\rangle$$

$$\geq \frac{|C|+1}{u[i]} \cdot u[i] = |C| + 1 > C,$$

where the first inequality is by $\left\langle \frac{|C|+1}{u[i]} \operatorname{Onehot}(i, \mathcal{B}), v \right\rangle \leq 0, \forall u[i] < 0, \forall v \in \Delta(\mathcal{B})$. This implies

$\forall u \in U^-$, $\sup_{\Theta \in \mathbb{R}^B} \left( \max_{v \in \Delta(\mathcal{B})} \langle \Theta, u - v \rangle \right) = +\infty$. Thus, $\forall u \in U^-$, $R_{\mathcal{N}}(u) = +\infty$.

Similarly, for any $u \in U^+$, we have that $\forall C \in \mathbb{R}$, by setting $\theta(u, C) = \frac{|C|+1}{\|u\|_1 - 1} \cdot u \in \mathbb{R}^B$,

$$\max_{v \in \Delta(\mathcal{B})} \langle \theta(u, C), u - v \rangle = \max_{v \in \Delta(\mathcal{B})} \left\langle \frac{|C|+1}{\|u\|_1 - 1} \cdot u, u - v \right\rangle$$

$$= \left\langle \frac{|C|+1}{\|u\|_1 - 1} \cdot u, u - \frac{u}{\|u\|_1} \right\rangle$$

$$= \frac{|C|+1}{\|u\|_1 - 1} \cdot \|u\|_1 \cdot (\|u\|_1 - 1) > C,$$

which is by basic algebra. This implies $\forall u \in U^+$, $R_{\mathcal{N}}(u) = +\infty$.

In conclusion, we have that $\forall u \in \mathbb{R}^B \setminus \Delta(\mathcal{B}) = U^- \cup U^+$, $R_{\mathcal{N}}(u) = +\infty$, which concludes the the proof. $\qquad \square$

**Lemma 51** (Restatement of Lemma 18). *For any $\Theta \in \mathbb{R}^d$,*

$$\operatorname*{argmin}_{u \in \Delta(\mathcal{B})} \left( \langle \Theta, u \rangle + R_{\mathcal{N}}(u) \right) = \nabla \Phi_{\mathcal{N}}(-\Theta).$$

*Proof.* As shown by Lemma 48 and Lemma 50, $R_{\mathcal{N}}$ is closed and strongly convex with $\operatorname{dom}(R_{\mathcal{N}}) = \Delta(\mathcal{B})$. By applying Proposition 41 on $R_{\mathcal{N}}$, we have

$$\nabla R_{\mathcal{N}}^*(-\Theta) = \operatorname*{argmax}_{u \in \mathbb{R}^B} \langle -\Theta, u \rangle - R_{\mathcal{N}}(u) = \operatorname*{argmin}_{u \in \Delta(\mathcal{B})} \langle \Theta, u \rangle + R_{\mathcal{N}}(u),$$

where the second equality is by $\text{dom}(R_\mathcal{N}) = \Delta(\mathcal{B})$ shown in Lemma 50.

As shown by Lemma 45, $\Phi_\mathcal{N}$ is closed and convex. By item 2 of Fact 40, $R_\mathcal{N}^* = \Phi_\mathcal{N}^{**} = \Phi_\mathcal{N}$, which concludes the proof. $\qquad\square$

**Lemma 52.** $\sup_{u \in \Delta(\mathcal{B})} R_\mathcal{N}(u) - \inf_{u \in \Delta(\mathcal{B})} R_\mathcal{N}(u) \leq \sqrt{2X \ln(B)}$.

*Proof.* First, by Lemma 49, since $\forall u \in \Delta(\mathcal{B})$, $R_\mathcal{N}(u) \leq 0$. we have that $\sup_{u \in \Delta(\mathcal{B})} R_\mathcal{N}(u) \leq 0$. Next, we show $-\inf_{u \in \Delta(\mathcal{B})} R_\mathcal{N}(u) \leq \sqrt{2X \ln(B)}$. Since $-\inf_{u \in \Delta(\mathcal{B})} R_\mathcal{N}(u) = \sup_{u \in \Omega} \langle 0, u \rangle - R_\mathcal{N}(u) = \Phi_\mathcal{N}(0)$, it suffices to show $\Phi_\mathcal{N}(0) \leq \sqrt{2X \ln(B)}$, which we already shown in Lemma 46.

Together, we conclude $\sup_{u \in \Delta(\mathcal{B})} R_\mathcal{N}(u) - \inf_{u \in \Delta(\mathcal{B})} R_\mathcal{N}(u) \leq \sqrt{2X \ln(B)}$. $\qquad\square$

Now, combining the above lemmas with the general optimistic FTRL lemma, we get the following central regret theorem for optimistic FTRL with separator perturbation-based regularizers for our results:

**Theorem 53.** *Suppose $\mathcal{X}$ is a separator set for $\mathcal{B}$, Optimistic FTRL (Algorithm 6) with $R = R_\mathcal{N}$ achieves regret*

$$\text{LReg}_N \leq \frac{\sqrt{2X \ln(B)}}{\eta} + \sum_{n=1}^{N} (\eta X A \|g_n - \hat{g}_n\|_\infty^2 - \frac{1}{4\eta X A} \|\nabla R_\mathcal{N}^*(-\hat{\Theta}_n) - \nabla R_\mathcal{N}^*(-\Theta_{n-1})\|_1^2).$$

*Furthermore, if $\hat{g}_n = 0$ for all $n$,*

$$\text{LReg}_N \leq \frac{\sqrt{2X \ln(B)}}{\eta} + \eta X A \sum_{n=1}^{N} \|g_n\|_\infty^2.$$

*Proof.* Since $R_\mathcal{N}$ is $\sqrt{\frac{\pi}{8}} \frac{1}{AX}$-strongly convex by Lemma 48, by the regret guarantee of Optimistic FTRL in Theorem 42,

$$\text{LReg}_N \leq \frac{\sup_{w \in \Omega} R(w) - \inf_{w \in \Omega} R(w)}{\eta}$$
$$+ \sum_{n=1}^{N} (\sqrt{\frac{2}{\pi}} \eta X A \|g_n - \hat{g}_n\|_\infty^2 - \sqrt{\frac{\pi}{32}} \frac{1}{\eta X A} \|\nabla R_\mathcal{N}^*(-\hat{\Theta}_n) - \nabla R_\mathcal{N}^*(-\Theta_{n-1})\|_1^2).$$

By bringing in $\sup_{u \in \Delta(\mathcal{B})} R_\mathcal{N}(u) - \inf_{u \in \Delta(\mathcal{B})} R_\mathcal{N}(u) \leq \sqrt{2X \ln(B)}$ proved by Lemma 52 and using the simple facts that $\sqrt{\frac{2}{\pi}} \leq 1$ and $\sqrt{\frac{\pi}{32}} \geq \frac{1}{4}$, we conclude the proof of the first inequality. Specifically, when $\hat{g}_n = 0$ for all $n$, by Algorithm 6 we have that $\hat{\Theta}_n = \Theta_{n-1}$ and

$$\text{LReg}_N \leq \frac{\sqrt{2X \ln(B)}}{\eta} + \eta X A \sum_{n=1}^{N} \|g_n\|_\infty^2. \qquad\square$$

**Lemma 54.** *Let $R : \mathbb{R}^d \to \mathbb{R} \cup \{+\infty\}$ be a closed and $\alpha$-strongly convex function with respect to $\|\cdot\|$, then, for $\Theta, \Theta' \in \mathbb{R}^d$,*

$$D_{R^*}(\Theta, \Theta') \geq \frac{\alpha}{2} \|\nabla R^*(\Theta) - \nabla R^*(\Theta)\|^2.$$

*Proof.* Since $R$ is closed and $\alpha$-strongly convex, by item 1,4 of Fact 40 and Proposition 41, $R^*$ is closed, convex, differentiable, and $\text{dom}(R^*) = \mathbb{R}^d$. By item 3 of Fact 40, $\forall \Theta, \Theta' \in \mathbb{R}^B$,

$$\begin{cases} R^*(\Theta) = \langle \Theta, \nabla R^*(\Theta) \rangle - R(\nabla R^*(\Theta)), \\ R^*(\Theta') = \langle \Theta', \nabla R^*(\Theta') \rangle - R(\nabla R^*(\Theta')). \end{cases}$$

As a consequence, both $\nabla R^*(\Theta)$ and $\nabla R^*(\Theta')$ are in $\mathrm{dom}(R)$. Furthermore, by item 3 of Fact 40, $\Theta \in \partial R(\nabla R^*(\Theta))$ and the definition of Bregman divergence, we have

$$
\begin{aligned}
D_{R^*}(\Theta', \Theta) =& R^*(\Theta') - R^*(\Theta) - \langle \Theta' - \Theta, \nabla R^*(\Theta') \rangle \\
=& \langle \Theta, \nabla R^*(\Theta) \rangle - R(\nabla R^*(\Theta)) - \langle \Theta', \nabla R^*(\Theta') \rangle + R(\nabla R^*(\Theta')) \\
& - \langle \Theta' - \Theta, \nabla R^*(\Theta') \rangle \\
=& R(\nabla R^*(\Theta')) - R(\nabla R^*(\Theta)) - \langle \Theta, \nabla R^*(\Theta') - \nabla R^*(\Theta) \rangle \\
\geq& \frac{\alpha}{2} \| \nabla R^*(\Theta) - \nabla R^*(\Theta) \|^2,
\end{aligned}
$$

where the last inequality uses the $\alpha$-strong convexity of $R$, as well as $\Theta \in \partial R(\nabla R^*(\Theta))$. $\qquad \square$

## H  Auxiliary Lemmas

**Lemma 55.** *For two stationary policies $\pi$ and $\pi^E : \mathcal{S} \to \Delta(\mathcal{A})$, we have*

$$
J(\pi) - J(\pi^E) = H \cdot \mathbb{E}_{s \sim d_\pi} \mathbb{E}_{a \sim \pi(\cdot|s)} \left[ A^E(s, a) \right],
$$

*where $A^E(s, a) := Q_{\pi^E}(s, a) - V_{\pi^E}(s)$, $V_{\pi^E}(s) := \mathbb{E}\left[ \sum_{t=\mathrm{Step}(s)}^{H} c(s_t, a_t) \mid s, \pi^E \right]$, and $Q_{\pi^E}(s, a) := c(s, a) + \mathbb{E}\left[ \sum_{t=\mathrm{Step}(s)+1}^{H} c(s_t, a_t) \mid s, a, \pi^E \right]$.*

The proof can be found at e.g. [55, Lemma 4.3] .

**Lemma 56.** *For benchmark policy class $\mathcal{B}$ that contains $B$ deterministic policies $h : \mathcal{S} \to \mathcal{A}$, consider separator set $\mathcal{X}$ 1 with $X = |\mathcal{X}|$, $A = |\mathcal{A}|$. Then,*

$$
X \geq \log_A(B).
$$

*Proof.* Define $\mathcal{B}_{\mathcal{X}} = \left\{ (h(x_1), \ldots, h(x_X)) \right\}_{h \in \mathcal{B}}$, where $(h(x_1), \ldots, h(x_X)) \in \mathcal{A}^X$. First, note that $\mathcal{B}_{\mathcal{X}} \subset \mathcal{A}^{\mathcal{X}}$, which implies that $|\mathcal{B}_{\mathcal{X}}| \leq |\mathcal{A}^{\mathcal{X}}| = A^X$.

Secondly, by the definition of separator set $\mathcal{X}$, $\forall h, h' \in \mathcal{B}$, $\exists x \in \mathcal{X} = \{ x_1, \cdots, x_X \}$, s.t. $h(x) \neq h'(x)$. This implies $\forall h, h' \in \mathcal{B}$, $(h(x_1), \ldots, h(x_X)) \neq (h'(x_1), \ldots, h'(x_X))$, and every $h$ in $\mathcal{B}$ induces unique $(h(x_1), \ldots, h(x_X))$; this implies that $|\mathcal{B}_X| = |\mathcal{B}| = B$.

Combining the above two observations, we conclude $B = |\mathcal{B}_{\mathcal{X}}| \leq A^X$, thus $X \geq \log_A(B)$. $\qquad \square$