# OpenReview forum: "On Efficient Online Imitation Learning via Classification"
_NeurIPS.cc/2022/Conference — NeurIPS 2022 Accept_

### Official Review · Reviewer_yqpJ · 2022-07-12

**Rating:** 7
**Confidence:** 2
**Soundness:** 4 excellent
**Presentation:** 4 excellent
**Contribution:** 3 good

**Summary:**

This paper investigates the fundamental feasibility and statistical limits of designing oracle-efficient regret minimization algorithms in the setting of classification-based online imitation learning. Specifically, the paper shows that any proper algorithm is unlikely to achieve a sublinear regret (similar to the Cover's impossibility theory). Inspired by this observation, the paper propose an improper online learning framework that is equivalent to online linear optimization with the help of a mixed policy class. This framework allows us to design oracle0efficient algorithms with different sample/interaction round complexities.

**Questions:**

I note that theorem 4 is based on the assumption that the expert's feedback is of the form $\zeta_E(s,a) = A^E(s,a)$. How does this choice affect the results and can we use other choices like direct expert annotation (zero-one loss, which seems more practical or easier to obtain in practice)?

**Limitations:**

Limitations and potential negative societal impact of their work have been properly discussed.

**Strengths And Weaknesses:**

Strengths
- The paper is a solid theoretical work that studies the fundamental limitations of classification-based online imitation learning and propose a new algorithmic framework to bypass the fundamental challenges.
- Good presentation quality with nice discussions about the connection and differences to previous works.

Weaknesses
- I do not find any technical issues. It would be very helpful if there are some experimental results (at least on some toy settings) to demonstrate the practical value as well as to make the key insights easier to understand.

---

> ### Author Response · Authors · 2022-08-02
> **Thank you**
>
> Thank you for your reviews.
>
> **Theorem 4 with expert action feedback.** Thank you for your question. For expert action feedback, the same MDP, policy class, and expert policy in the proof of Theorem 4 continue to yield an $\Omega(N)$ regret lower bound for proper learning when $H \geq 4$.
>
> To recap, in the proof Theorem 4, we construct an MDP with state space $\mathcal{S} = \\{S_0,S_L,S_R\\}$ and action space $\mathcal{A} = \\{L,R\\}$, where playing $L$ and $R$ at $S_0$ transit to $S_L$ and $S_R$ with probability $1$, respectively. The deterministic expert plays action $L$ at $S_0, S_R$ and $R$ at $S_L$, while two base policies $h_L$ and $h_R$ play $L$ and $R$ at all states, respectively.
>
> First, it can be seen from the advantage function values calculated from the proof of Theorem 4 that $(\mathcal{M},\pi^E)$ is 1-recoverable. Therefore, the direct expert annotation feedback is of the form $\zeta_E(s,a) = I(a \neq \pi^E(s))$. Recall the definition of $F_n$ in our line 137. We follow a similar calculation as in the proof of Theorem 4 and obtain:
> * if $\pi_n = h_L$, then $F_n(h_L) = \frac{H-1}{H}$, $F_n(h_R) = \frac{1}{H}$,
> * if $\pi_n = h_R$, then $F_n(h_R) = 1$, $F_n(h_L) = 0$.
>
> With this, we conclude that, for any policy sequence $\pi_1, \ldots, \pi_N \in \\{ h_L, h_R \\}$ output by a proper learner, $\sum_{n=1}^N F_n(\pi_n) \geq  \frac{ N (H-1)}{H}$.
>
> On the other hand, $\min_{\pi \in \mathcal{B}} \sum_{n=1}^N F_n(\pi) \leq \frac{N(H+1)}{2H}$, by choosing $\pi$ to be the less frequently played policy in $\mathcal{B}$, formally, $\pi = h_L$ if $|\\{ n: \pi_n = h_L \\}| \leq N/2$, and $\pi = h_R$ otherwise.
>
> Combining the above two inequalities, we obtain
> $SReg_N(\mathcal{B}) = \sum_{n=1}^N F_n(\pi_n) - \min_{\pi \in \mathcal{B}}  \sum_{n=1}^N F_n(\pi) \geq
> \frac{N (H-1)}{H}-\frac{N(H+1)}{2H}
> =
> \frac{N(H-3)}{2H}$,
> which is linear in $N$ when $H \geq 4$.
>
> We will also add this example (the failure of proper learning with expert providing direct action feedback) in the final version.
>
> **Experiment Plan**: See our post "experimental plan" to all reviewers for more explanations.

---

### Official Review · Reviewer_7TV7 · 2022-07-13

**Rating:** 7
**Confidence:** 3
**Soundness:** 3 good
**Presentation:** 3 good
**Contribution:** 3 good

**Summary:**

This paper studies the fundamental problems of classification-based online imitation learning. A framework called COIL is developed, and several contributions are achieved. First, like Cover's impossibility result, this paper shows there exists a hard instance, such that any proper online learning algorithm must suffer a sublinear regret. Second, an algorithmic framework called COIL is developed, which reduces COIL to online linear optimization. Third, two algorithms, LOGGER-M and LOGGER-ME, are proposed with provably efficient sample and interaction round complexities. This paper argues that these algorithms can be better than BC. Finally, this paper discusses why efficient dynamic regret minimization is inapplicable in the LOGGER framework.


====

After discussion, the authors have addressed my concerns. I am glad to see that this paper makes a remarkable step in the non-realizable cases. I believe such results could help us better apply imitation learning algorithms in practice. Accordingly, I raise my score from 5 to 7.

**Questions:**

- Can the authors help explain the computation benefits of LOGGER-M and LOGGER-ME?

In Section 3.2, this paper argues that regret-minimization algorithms like Hedge has a computation issue, i.e., O(B) complexity. I believe this is not a big issue. Furthermore, I do not find the discussion about the computation efficiency of LOGGER-M and LOGGER-ME in Section 4. I hope the authors can help explain this point.

- What are new messages compared with the prior work [2]?

I find this paper lacks a discussion with the prior work [2], in which it is shown that (with realizability assumption), an online IL algorithm can solve the COIL studied in this paper. Furthermore, a separation between the online and offline settings is established to help the readers understand the fundamental limit. I wonder whether there are new messages in this paper.

- Is the theoretical guarantee for BC tight?

[1] proved that the expert annotation complexity for BC should be $\widetilde{O}(H^2 S/\epsilon^2)$, if there is no approximation error. I believe this result is true even though there is an approximation error. The analysis in the appendix may be loose. For the approximation error of BC, due to the $\mu$-recoverability assumption, I guess the tight dependence should be $\mu H$ rather than $H^2$.

---


[2] Rajaraman, Nived, et al. "On the Value of Interaction and Function Approximation in Imitation Learning." *Advances in Neural Information Processing Systems* 34 (2021): 1325-1336.

**Limitations:**

This paper studies classification-based online imitation learning (COIL). By the powerful techniques from the online optimization literature, new theoretical results are obtained in the context of imitation learning. Some theoretical results can corroborate some discussions in [3]. Though this paper has fruitful results, I am worried about the implications of these results.

What are new messages? (compared with the old paper like DAGGER and the new paper [2]) How can the theoretical results of this paper guide practice? This question is related to the remark in the conclusion part. What kind of experiments do you want to conduct to empirically evaluate algorithms?

This paper is well written. A minor issue is a claim in the introduction (Line 30 and 31). In general, interactive IL can not be better than its offline counterpart; see [1]. Only under suitable assumptions, interactive IL can have superior performance; see [2]. A side message is that for tasks similar to the lower bound in [1], online algorithms cannot help. This message is important for practitioners. Please consider clarifying this misleading claim.

I am willing to recommend accepting this paper if the above concerns can be well-addressed.

---

[3] Spencer, Jonathan, et al. "Feedback in imitation learning: The three regimes of covariate shift." *arXiv preprint arXiv:2102.02872* (2021).

**Strengths And Weaknesses:**

**Strengths**

- Several new theoretical contributions to the online imitation learning field. Like Cover's impossibility result, Theorem 4 shows that without reachability or randomization, it is impossible to achieve a sublinear regret.
- By considering mixed policy classes, new algorithms LOGGER-M and LOGGER-ME are developed.

**Weakness**

- Motivation is weak. This paper claims that the prior frameworks like DAgger have two issues: 1) applying rule-based policies is hard; 2) optimization of such policies is tricky (because of the convex realization). These issues depend on specific applications. It seems that they are not fundamental issues.
- The BC's guarantee is weaker, and the improvements over BC are questionable. The direct reduction of BC to supervised learning yields a loose bound, as shown in [1]. The expert annotation complexity of BC should be \widetilde{O}(H^2/\epsilon) rather than \widetilde{O}(H^4/\epsilon^2).

---

[1] Rajaraman, Nived, et al. "Toward the fundamental limits of imitation learning." *Advances in Neural Information Processing Systems* 33 (2020): 2914-2924.

---

> ### Author Response · Authors · 2022-08-02
> **Thank you**
>
> Thank you for your review. We address your questions below.
>
> **Motivation & Main messages:** As in our general response to all reviewers, this work aims at providing an end-to-end computational and statistical analysis of imitation learning in the *general function approximation* and *nonrealizable* setting, putting provably efficient imitation learning into a firmer ground.
>
> In contrast, prior works on provably-efficient imitation learning mainly fall into two categories:
>
> (1) reduction from imitation learning to online convex optimization, taken by DAgger and subsequent works: these works provide policy suboptimality guarantees in terms of convex-relaxed losses, which are known to be loose even in the special case of supervised learning;
>
> (2) the minimax analysis of [1,2]: although these works provide tight sample complexity analysis, they only cover the realizable setting and tabular/linear function approximation setting to date.
>
> **Practical guidance:** Our work enables the design of imitation learning algorithms for broader policy classes; specifically we show that, for any offline-optimizable policy class with separators, we can perform imitation learning with this policy class using our algorithms, and achieve provable guarantees.
>
> **Guarantee of Behavior Cloning:** Recall that we study imitation learning in the *nonrealizable* setting (where $\pi^E$ is not necessarily in $\mathcal{B}$); in this setting, the $1/\epsilon^2$ dependence in the sample complexity bound is, in general, unavoidable. To see this, note that behavior cloning with episode length $H=1$, action space size $A=2$ is the classical agnostic supervised binary classification setting. In this case, even when the expert policy $\pi^E$ is deterministic, the sample complexity for learning a policy that disagree with $\pi^E$ with probability $\epsilon$ is $\Omega(1/\epsilon^2)$ (Theorem 7 of Urner and Ben-David, the sample complexity of agnostic learning under deterministic labels, COLT 2014). This result can be readily transformed to a sample complexity lower bound $\Omega(1/\epsilon^2)$ for imitation learning.
>
> In realizable or near-realizable settings, we agree with the reviewer that tighter sample complexity bounds can be proved; we will add a discussion on this and leave refined sample complexity analysis in these settings for future work.
>
> **Computational benefits of LOGGER-M and LOGGER-ME:** First, the size of the benchmark policy class $\mathcal{B}$ can well be exponentially large - consider the policy class of linear classifiers or fixed-architecture neural networks; their sizes are exponential in the number of free parameters therein; in these settings, applying Hedge on $\mathcal{B}$ has a per-iteration running time of $O(B)$, which is impractical.
>
> The main computational benefit of LOGGER-M and LOGGER-ME is that, they run in polynomial time, and only make a polynomial number of calls (more concretely, $O(\mathrm{poly}(N, \log S, \log 1/\delta))$ as in Theorems 10 and 12) to the offline cost-sensitive classification oracle. Assuming that each underlying oracle call has a running time polynomial with respect to the number of free parameters of $\mathcal{B}$ (i.e. $\mathrm{polylog}(B)$), the total time complexities of our LOGGER-M and LOGGER-ME are also $\mathrm{polylog}(B)$, which is practical.
>
> **Experiment plan**: See our post "experimental plan" to all reviewers for more explanations.
>
> **Claim in the introduction (line 30-31):** thank you for your suggestion - we agree with it and will revise it in the final version.

---

> > ### Comment · Reviewer_7TV7 · 2022-08-05
> > **Thanks for Your Answer**
> >
> > Thanks for your detailed answers. I have no questions about the main results. Just a few comments.
> >
> > I can understand the separation between realizable and non-realizable cases. This could be a motivation in the introduction, if the authors agree. Furthermore, the separation seems to suggest that we should seek a powerful function approximator (e.g, neural networks) to get a better sample complexity. Of course, this implication is not this paper's main focus/motivation.  In some cases where the realizable assumptions cannot be satisfied, the algorithms in this paper are helpful. Please let me know if I misunderstood such side messages. Thanks!

---

> > > ### Author Response · Authors · 2022-08-07
> > > **Again thanks for your comments.**
> > >
> > >
> > > Thank you for your thoughtful feedback!
> > >
> > >
> > >
> > > > I can understand the separation between realizable and non-realizable cases. This could be a motivation in the introduction, if the authors agree.
> > >
> > > > In some cases where the realizable assumptions cannot be satisfied, the algorithms in this paper are helpful.
> > >
> > >
> > >
> > > Thank you for your suggestion; we agree that addressing the non-realizable case is one of the key features of this paper. The improvement of our results compared to the original DAgger-style convex relaxation & online convex optimization approaches is most significant in the nonrealizable case; in the realizable case, the approximation error (benchmark) terms are zero, with and without convex relaxation. In our appendix (lines 694-698), we compared our setting and recent minimax sample complexity analysis ([1,2] in your review); in the final version, we will move that comparison to the introduction section, and incorporate our discussions above, to provide a stronger motivation.
> > >
> > >
> > >
> > > > the separation seems to suggest that we should seek a powerful function approximator (e.g, neural networks) to get a better sample complexity.
> > >
> > >
> > >
> > > We agree that this separation may encourage learning the expert policy by a richer policy class. However, similar to supervised learning, it is unclear if increasing the complexity of the policy class always reduces sample complexity, as suggested by Occam’s Razor (see e.g. Shalev-Shwartz and Ben-David, Understanding Machine Learning: From Theory to Algorithms, Chapter 7). We believe that there is an interesting problem of model selection here.
> > >
> > >
> > >
> > >
> > >
> > > Again, thank you for the helpful discussions. If our answers addressed your questions, we would appreciate it if you could reconsider the score of our work.

---

> > > > ### Comment · Reviewer_7TV7 · 2022-08-07
> > > > **Increase My Score to 7**
> > > >
> > > > I agree with your answer about increasing the capacity of function class. My concerns are addressed and I have increased my score to 7. Please involve the above discussions in the main text in the final version.

---

### Official Review · Reviewer_vCUw · 2022-07-14

**Rating:** 6
**Confidence:** 3
**Soundness:** 3 good
**Presentation:** 2 fair
**Contribution:** 3 good

**Summary:**

The authors provide (1) an impossibility result for obtaining sublinear regret in general online learning settings, (2) present an online linear optimization algorithm (LOGGER) which uses policy mixtures to improve sample / complexity interaction guarantees over behavioral cloning algorithms, and (3) they show that achieving sublinear dynamic regret is still impossible in the LOGGER framework.





**Questions:**

## Abstract
1. Please fix the spacing in the abstract.

## Background
1. The assumption about labeling of the expert examples seems unrealistic. In most settings checking if something is an expert action requires the same effort as just providing the binary value provided in your framework (unless you are using human in the loop RL for language tasks possibly). It makes sense why this is a much more difficult form of feedback to optimize with, but making the distinction between this and more common assumptions should be included.
2. Theorem 4 seems similar to impossibility results from the reinforcement learning theory literature. This impossibility result is also somewhat confusing considering your previous result which uses effectively they same type of feedback, with what amounts to less information (e.g. binary feedback telling you if you were wrong, versus feedback telling you if you explicitly how much worse you were then the expert). Can you clarify intuitively why your result makes sense.
5. Can you clarify the statement made in line 206 "equivalent to drawing a freshly new policy ...", are you sampling without replacement from the set B? If so does this not enforce additional requirements on the policy class not included in your list of assumptions?
6. Proposition 6 seems strictly worse then the results presented in [1]
7. Does the assumption regarding mu recoverability implicitly impose constraints on the space of MDPs that are not likely to be realistic in practice.

## Efficient algorithms with static regret guarantees
1. Text error on line 261

## Potentially relevant citations
[1] https://arxiv.org/abs/2007.02520

[2] https://arxiv.org/abs/1809.02864

[3] https://arxiv.org/abs/2006.06835

**Limitations:**

The most important limitation of this work, is that it proposes a framework which creates a trade off between sample efficiency and oracle query efficiency, but provides no experimentation of such an algorithm. If the multiple variants of LOGGER are not possible to implement in practice, this should be stated explicitly. If they are, a simple toy experiment would be useful to illistrate to the reader how such an algorithm function, and whether or not there are any hidden constants in the bound or the algorithm definition which make it intractable in practice.

**Strengths And Weaknesses:**

## Strengths
1. The authors combined analysis which composes sample complexity and interaction round complexity is extremely interesting, and to my knowledge novel in the area of online imitation learning.
2. The list of assumptions required in proving the results given by the authors is clear and precise.
3. The use of "separator sets" by the authors seemingly opens up a couple fun algorithmic questions that can take advantage of existing research from support vector machines and active sets.
4. Proposition 2 creates an interesting link between exploration in online imitation learning and recent results in over-parameterization in online convex optimization~[1,2].

## Weaknesses
2. Proposition 6 seems like a restatement of one of the theorems in [1], or perhaps a special case.
3. The assumption regarding mu recoverability seems to implicitly impose some constraints on the space of MDPs that are not likely to be realistic in practice.
5. All analysis are completed in a a discrete finite state-action Markov decision process, making the utility of such analysis unlikely to be directly applicable to practical applications and studies.
6. No experimentation or code is included with the work, making it difficult to determine if the algorithms proposed are actually possible to implement.

---

> ### Author Response · Authors · 2022-08-02
> **Thank you**
>
> Thank you for your review. We address your questions below.
>
> **Novelty of Proposition 6**: Our Proposition 6 is implicit in the general reduction framework of DAgger (and subsequent works), and is thus not technically new. It mainly serves as a conceptual helper to facilitate subsequent development: it enables us to use the LOGGER framework to design computationally-efficient algorithms LOGGER-M, LOGGER-ME with online linear optimization regret guarantees, which yields policy optimization guarantees.
>
> We double checked the paper “Explaining Fast Improvement in Online Imitation Learning” - thanks for the reference. Their equations (7) and (10) follow the same derivation as our proposition. On the other hand, neither their Theorem 1 nor Theorem 2 are applicable to our linear functions, as their Lemma 1 requires the  loss function to be nonnegative over $\mathbb{R}^B$. In case we miss something, please feel free to follow up.
>
> **The $\mu$-recoverability assumption:** First, any expert policy $\pi^E$ and episodic MDP $\mathcal{M}$ always satisfy $\mu$-recoverability with $\mu = H$.
>
> Second, it is known that in the tabular imitation learning setting (Rajaraman, Yang, Jiao, Ramchandran,Toward the fundamental limits of imitation learning, NeurIPS 2020), when $\mu = \Omega(H)$, behavior cloning and interactive imitation learning have the same minimax-optimal sample complexities; under the $\mu$-recoverability assumption with $\mu \ll H$, there is a separation between the minimax optimal sample complexities in behavior cloning and interactive imitation learning. (See also Reviewer 7TV7’s comments that “In general, interactive IL can not be better than its offline counterpart .. cannot help.”) So we view the condition $\mu \ll H$ as natural in establishing the benefit of imitation learning in the more general function approximation setting as well.
>
> Third, we agree that our notion of $\mu$ is a bit pessimistic, as it is requires $|A^E(s,a)|$ to be bounded by $\mu$ for the worst-case $(s,a)$ pair. It would be interesting to study an average-case notion of recoverability - we leave it for future work.
>
> **Discrete finite state-action settings:** We would like to point out that our sample complexity guarantees only have a polylogarithmic dependence on $S$, the number of the states of the MDP (note that $\log(S)$ is the bit complexity of the state space), and a polylogarithmic dependence on $B$, the size of the benchmark policy class (see Table 1 and Appendix E.3), which makes our results relevant in large state space, large policy class settings.
>
> Our sample complexity results have a polynomial dependence on the size of the action space, because of the classification nature of our problem; we leave the challenge of dealing with large action spaces to future work.
>
> **Expert action feedback:** If we understand correctly, you are referring to the discussion on the expert feedback $\zeta^E(s,a) = I(a \neq \pi^E(s))$ in line 121. The practical scenario we had in mind was identical to e.g. (Ross, Gordon, Bagnell, 2011), i.e. given state $s$, expert provides demonstration $\pi^E(s)$, and we use it to construct an $A$-dimensional cost vector $\zeta^E(s,a) = (I(a \neq \pi^E(s)) )_{a \in \mathcal{A}}$. We apologize for the confusion and will clarify it in the final version.
>
> **Theorem 4:** Theorem 4's key intuition is similar to Cover’s impossibility result in online classification: because of the nonrealizability of the benchmark policy class and the construction of the MDP, any proper learning algorithm must suffer a loss of at least $(H-1)$ for every episode, making its cumulative loss at least $N(H-1)$; however, for at least one $s \in {S_L, S_R}$, in at least $N/2$ episodes, the learner's policy enters state $s$ and stays there deterministically, making the best policy in hindsight suffering a cumulative loss at most $N(H-1)/2$.
>
> Theorem 4 also holds for the direct expert annotation feedback; see our response to Reviewer yqpJ: the same MDP, benchmark policy class and expert policy in the proof of Theorem 4 still forces any proper learner to have a linear regret, in the direct expert annotation feedback setting. We will include this to the final version.
>
> **Statement in line 206 “fresh new policy”:** no, the sampling here is with replacement; sorry for the confusion, we will clarify this in the final version.
>
> **Experiment plan:** See our post "experimental plan" to all reviewers for more explanations.

---

> > ### Comment · Reviewer_vCUw · 2022-08-09
> > **Response**
> >
> > Thank you for addressing my comments. I think that this paper as mentioned above addresses problems that are important to the community, while producing non-trivial theoretical results. Without some really basic experimentation added to the paper, I wont be increasing my score to a full accept. I appreciate the time-constraints of the rebuttal period, but stand by the fact that without these checks, it can be very difficult to truly determine if the algorithms and results presented by the authors actually make sense for even simple sets of problems.

---

### Official Review · Reviewer_cDuK · 2022-07-20

**Rating:** 6
**Confidence:** 3
**Soundness:** 4 excellent
**Presentation:** 3 good
**Contribution:** 3 good

**Summary:**

This paper studies the problem of classification-based online imitation learning, and shows a negative result that any proper learning fails to achieve sublinear regret.
The main contributions include:
  - Theoretical guarantee for drawbacks of proper online learning
  - An improper online learning algorithmic framework
  - Efficient algorithms to deal with computational hardness


**Questions:**

N/A

**Strengths And Weaknesses:**

- Strengths
  - The paper proves that proper learning fails to achieve sublinear regret in COIL setting.
  - The paper proposes a clear convex combination to construct mixed policy class so that online regret minimization in IL becomes an online linear optimization problem.
  - The paper proposes Theorem 13 to show that all problems in PPAD are solvable in randomized polynomial time under some specific circumstances in COIL setting.

- Weaknesses
  - [12] already gives a general online convex optimization formulation for online IL, so it seems that linear optimization is covered.

---

> ### Author Response · Authors · 2022-08-02
> **Thank you**
>
> Thank you for your review.
>
> **Comparison with [12] ‘Online learning with continuous variations: dynamic regret and reductions’:** Indeed, interactive imitation learning by reduction to online learning / online convex optimization is well-known, since the seminal work of (Ross, Gordon, and Bagnell, A reduction of imitation learning and structured prediction to no-regret online learning, AISTATS 2011), and online classification can be viewed as an online linear optimization problem (Shalev-Shwartz, Online learning and online convex optimization, Section 2.1.1). Our LOGGER framework also falls into the broad umbrella.
>
> However, despite the practical relevance of the classification-based loss functions, all prior works do not provide end-to-end regret and computational analyses of imitation learning with respect to classification-based loss functions. Our work fills this gap by designing imitation learning algorithms that have provable regret guarantees against classification losses, and enjoy computational efficiency given an offline classification oracle.

---

### Author Response · Authors · 2022-08-02
**Thanks to all reviewers**

We thank all reviewers for their thoughtful comments and encouraging remarks.

To recap, our main contribution is an end-to-end computational and statistical analysis of classification-based imitation learning in the *general function approximation* and *non-realizable* settings. Here, “end-to-end” refers to establishing regret on the original cost-sensitive classification losses ($\mathrm{SReg}_N$ and $\mathrm{DReg}_N$ defined in line 141), which is more relevant in practical imitation learning scenarios. In contrast, most prior works on online imitation learning such as DAgger directly convexify the online losses, which is well-known to distort the original classification loss objectives (see also the discussion in line 55-56).

In the replies below, we address each reviewer's comments. Please feel free to follow up.

---

> ### Author Response · Authors · 2022-08-02
> **Experimental plan in future work**
>
> We plan to evaluate our algorithms in future work. Here is an example experimental setting we are envisioning, similar to the experiments in MoBIL (Cheng, Yan, Boots, Accelerating Imitation Learning with Predictive Models, AISTATS 2019):
>
> Environment: CartPole in Openai Gym. The state consists of 4 real numbers that measure the positions of the cart and the pole, and action space is binary or $0/1$ value indicating the force of pulling left or right. The goal is to keep the positions within a range for $H=500$ time steps, where at each time step the agent gets reward $1$ if it remains in the range.
>
> Expert: A 2-layer neural network (trained using reinforcement learning) as the expert that takes state as input and gives $0/1$ action feedback.
>
> Base policy class: The base policy class $\mathcal{B}$ is the linear class $\\{ \mathrm{sign}(\langle u, \phi(x) \rangle +b): w \in \mathbb{R}^d, b \in \mathbb{R} \\}$ for classification, where $\phi$ is a random Fourier feature map that maps a state to a $d$-dimensional feature vector, for $d=40$.
>
> Separator set and perturbation: Following the separator set construction in (Syrgkanis, Krishnamurthy, Schapire, Efficient Algorithms for Adversarial Contextual Learning, ICML 2016), the separator set is defined to be vectors in $\mathbb{R}^{40}$ that has value from set $\\{-20.5, 19.5...,-0.5, 0.5,...,20.5\\}$ in one dimension and others set to $0$, such as $(0.5, 0, 0, ..., 0)$. The size of separator set is $40 \times 40$. The perturbation set $Z$ is constructed by applying random $0/1$ labels on each element in the separator set with weight that follows the half-normal distribution with mean $0$ and standard deviation $0.1$.
>
> Classification oracle: We plan to use the Ramp-loss SVM, a non-convex SVM learner, using the UniverSVM package (https://github.com/fabiansinz/UniverSVM)
>
> Learning procedure for LOGGER-M (the procedure for LOGGER-ME can be designed similarly):
>
> First, draw a set of $40$ Fourier features, obtaining $\phi: \mathbb{R}^4 \rightarrow \mathbb{R}^{40}$ to transform states to features.
>
> At each learning round $n$, the learner:
> 1. draws $K=50$ i.i.d states by rolling out its current policy $\pi_n$ in the environment.
> 2. queries the experts' action $a$ on the states $s$ and adds those $(\phi(s),a)$ to dataset $D$ with weight $1$.
> 3. Set the sparsification parameter $T = 30$. For $j = 1, \ldots, T$, generates i.i.d. perturbation set $Z_j$ and calls the classification oracle with set $D \cup Z_j$ to obtain linear model weights $(u_j, b_j)$. Obtain the set of models $\mathcal{H} = \\{(u_1,b_1), …, (u_{T}, b_{T})\\}$.
> 4. $\mathcal{H}$ induces policy $\pi_{n+1}$, whose execution is as follows: for any input state $s$, draw $j \in {1,2, ..., T}$ uniformly at random at fresh, and chose action $\mathrm{sign}(\langle{u_j},{\phi(s)} \rangle +b_j)$.

---

### Meta-Review · Area_Chair_SdAz · 2022-08-26

**Recommendation:** Accept
**Confidence:** Certain

**Metareview:**

This paper studied imitation learning in the classification setting. The paper shows that using proper online learning algorithms is not sufficient to obtain sublinear regret, and devises an improper learning framework that relies on online linear optimization resulting in provably efficient algorithms.

All the reviewers appreciated the theoretical novelty and are unanimous in their decision to accept the paper. Please incorporate the reviewers' feedback and the resulting discussion. Adding in some basic experimental results (outlined in the "Experimental plan" comment) would strengthen the paper.

**Award:**

No

---

### Decision · Program_Chairs · 2022-09-14

Accept